# Boosting Medical Visual Understanding from Multi-Granular Language Learning

**Zihan Li**[*]
University of Washington
Seattle, WA 98195, USA
zhanli@uw.edu

**Yiqing Wang**[*]
Duke University
Durham, NC 27708, USA
yq.wang@duke.edu

**Sina Farsiu**[†]
Duke University
Durham, NC 27708, USA
sina.farsiu@duke.edu

**Paul Kinahan**[†]
University of Washington
Seattle, WA 98195, USA
kinahan@uw.edu

## Abstract

Recent advances in image-text pretraining have significantly enhanced visual understanding by aligning visual and textual representations. Contrastive Language-Image Pretraining (CLIP) has played a pivotal role in multimodal learning. However, its focus on single-label, single-granularity alignment limits its effectiveness in complex domains such as medical imaging, where images often correspond to multiple high-level labels (e.g., disease categories) across different annotation granularities (e.g., diagnostic description, clinical explanation). To address this, we propose Multi-Granular Language Learning (MGLL), a contrastive learning framework designed to improve both multi-label and cross-granularity alignment. MGLL leverages structured multi-label supervision, integrates textual descriptions across granularities, and introduces soft-label supervision with pointwise constraints to enhance alignment. MGLL employs smooth Kullback–Leibler (KL) divergence to ensure cross-granularity consistency while maintaining computational efficiency as a plug-and-play module for vision-language models. Pretrained on our constructed large-scale multi-granular datasets and evaluated across multiple datasets, MGLL outperforms other state-of-the-art methods in downstream tasks. The code is available at https://github.com/HUANGLIZI/MGLL.

## 1 Introduction

In recent years, the large-scale image-text pretraining has significantly improved the performance of downstream computer vision tasks. Among these approaches, Contrastive Language-Image Pretraining (CLIP) Radford et al. (2021) has gained widespread popularity for its ability to learn aligned visual and textual representations from paired data. CLIP has been extensively utilized in multimodal learning, ensuring that representations from different modalities remain semantically consistent. Consequently, CLIP has been employed for pretraining vision foundation models and fine-tuning on various downstream tasks, such as classification, image segmentation, and object detection. Despite the success of CLIP and related pretraining methods in aligning images with textual categories, simple image-text pair matching remains inadequate in medical domains such as imaging, biosignal analysis, and genomics. A single medical image or signal often maps to multiple target categories, requiring both multi-label and multi-granularity alignment. As shown in Fig. 1, a retinal fundus image may present both Diabetic Macular Edema and Diabetic Retinopathy, along with finer-grained labels like Severe Diabetic Macular Edema and Moderate Non-Proliferative Diabetic Retinopathy. This calls for alignment across multiple semantic levels. Existing multi-label contrastive methods Wang et al. (2022b); Saporta et al. (2024); Naeem et al. (2024) focus on instance-label correlations but struggle with cross-granular semantics and generalization. Compared to natural images, medical images encode more complex and hierarchical information that spans diagnoses, anatomical structures, lesions, and textures, yet they suffer from data scarcity due to privacy concerns and the high cost of annotation, which further compounds the challenge.

---

[*]Equal contribution.
[†]Co-corresponding author.

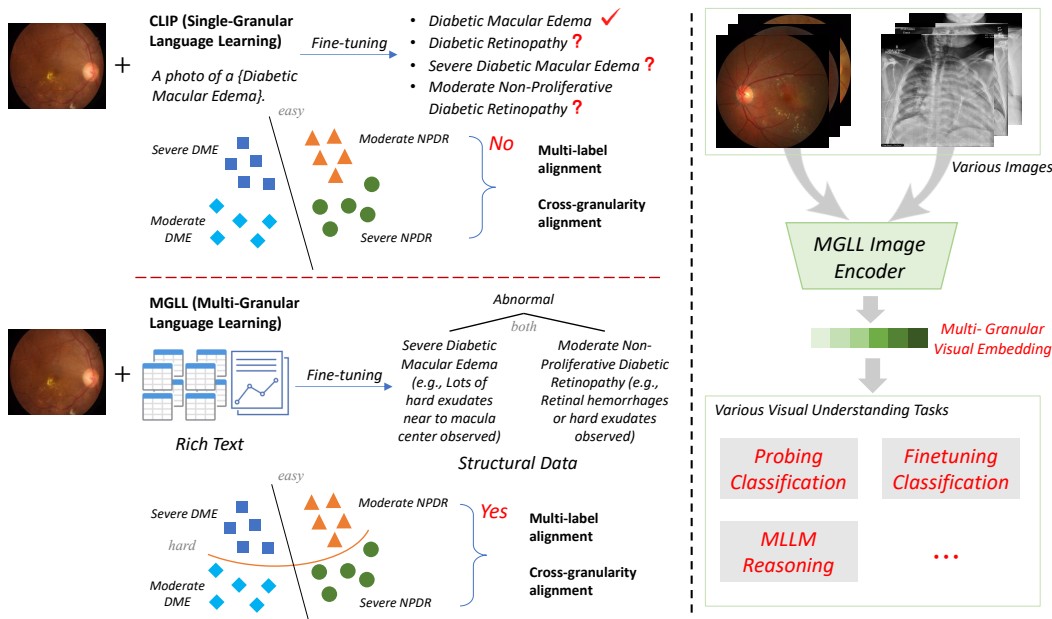

Figure 1: The illustrative comparison of input and outcome between CLIP and MGLL.

In this study, we aim to address the challenges of **multi-label alignment** and **cross-granularity alignment** through a generalizable image-text contrastive learning framework simultaneously. Here, we define "label" as a high-level disease category that an image belongs to, whereas "granularity" represents different levels or aspects of medical annotations, such as diagnostic attributes or clinical explanations. Unlike previous image-text contrastive pretraining approaches, which rely on single-granular, single-label supervision, we construct a multi-granular, multi-label datasets by collecting rich textual descriptions associated with the labels. Furthermore, we extend the original CLIP image and text loss Radford et al. (2021) to incorporate soft-label supervision and introduce point-wise constraints to enhance multi-label alignment. At the same time, we define contrastive learning objectives for each granularity level and employ smooth Kullback–Leibler (KL) divergence to achieve cross-granularity alignment. By jointly optimizing these learning objectives, our proposed MGLL (Multi-Granular Language Learning) effectively aligns image-text pairs across both multiple labels and multiple granularities. Notably, our method does not introduce any granularity-sensitive encoders, ensuring no additional computational cost. This allows MGLL to function as a plug-and-play module that can be integrated into any vision foundation model or large vision-language model Touvron et al. (2023). We hope our method and experiments can provide new insights into medical vision-language pretraining and facilitate more effective visual representation learning. Our contributions are as follows:

- We propose MGLL, a novel contrastive learning framework using multi-granular language that enables simultaneous multi-label and cross-granularity alignment.

- We provide a set of architecture-agnostic, multi-label, multi-granularity learning objectives that can be seamlessly integrated into vision-language models and foundation models to enhance medical visual understanding.

- We design a structured multi-granular, multi-label system and construct large-scale multi-granular retinal and X-ray image-text datasets. Extensive experiments on over ten downstream datasets demonstrate that MGLL consistently outperforms other state-of-the-art (SOTA) methods, exhibiting superior generalization ability.

## 2 RELATED WORK

### 2.1 IMAGE-TEXT CONTRASTIVE LEARNING

Large-scale image-text pretraining underpins modern multimodal learning. Contrastive methods like CLIP Radford et al. (2021) align visual and textual features via paired data. Variants such as SILC Naeem et al. (2024) use local-to-global pairwise learning, Symile Saporta et al. (2024) models higher-order multimodal relations, and Long-CLIP Zhang et al. (2024) handles extended text via

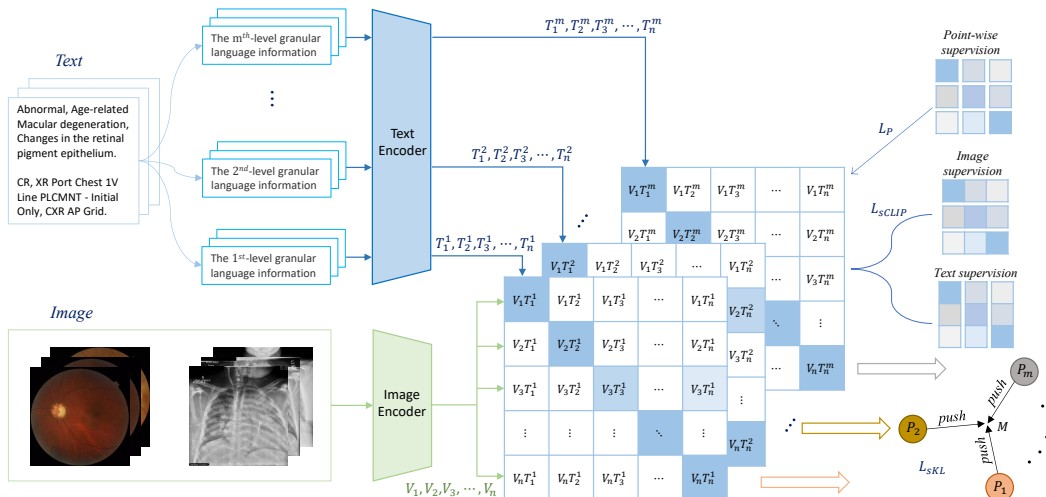

Figure 2: The overview of MGLL (Multi-Granular Language Learning) pretraining pipeline.

stretched embeddings. MedCLIP Wang et al. (2022b) addresses false negatives in medical data by semantic matching. Recent foundation models Silva-Rodriguez et al. (2025); Du et al. (2024); Li et al. (2025); Zhang et al. (2023); He et al. (2025; 2024) tailor contrastive learning to specific domains. Still, standard frameworks often underperform in medical settings due to data scarcity and complex semantics.

## 2.2 MULTI-LABEL LEARNING

While conventional deep networks perform well in single-label classification, real-world objects often carry multiple labels across entities, actions, and attributes Zhang et al. (2022); Khattak et al. (2024); Dai et al. (2024); Wang et al. (2023). Early work by Wang et al. Wang et al. (2016) learned joint image-label embeddings, and later introduced a recurrent attention module for interpretability Wang et al. (2017b). SupCon Khosla et al. (2020) extended contrastive learning to supervised settings using label structures, while Zhang et al. Zhang et al. (2022) proposed a hierarchy-preserving loss. Yet, these methods remain limited in vision-language integration, with fixed label spaces constraining semantic flexibility.

## 2.3 MULTI-GRANULARITY LEARNING

Visual and textual data convey semantics across multiple granularities. Recent work explores this via diverse frameworks Zhao et al. (2024); Li et al. (2023); Liu et al. (2023). Wang et al. Wang et al. (2022a) use bidirectional cross-attention for fine-grained alignment, Zhao et al. Zhao et al. (2024) propose a multi-granularity vision flow, and Xiong et al. Xiong et al. (2022) align inter-/intra-modal features with decision fusion. Du et al. Du et al. (2022) capture cross-modal multi-granular semantics for retrieval. Most of the these approaches rely on fixed training pipelines that limit their ability to incorporate heterogeneous or hierarchical annotations. Our MGLL provides a more flexible and generalizable framework for learning both multi-label and cross-granularity visual–language representations. MGLL can effectively utilize different types of granularity information across diverse datasets without requiring specific annotation formats or model architectures. MGLL also displays robust performance under complex scenarios such as mixed granularity and noised annotations.

## 3 MULTI-GRANULAR LANGUAGE LEARNING

### 3.1 OVERVIEW

To achieve multi-label and cross-granularity matching between images and text, we propose MGLL, a multi-granularity language-based contrastive learning framework. As shown in Fig. 2, our framework consists of an image encoder and a text encoder, where we use Vision Transformer Dosovitskiy et al. (2020) and BERT Devlin et al. (2019) as the default choices. First, we collect rich textual descriptions for both fundus and X-ray images, constructing two multi-granularity datasets: MGLL-Fundus and MGLL-Xray. Next, we leverage the encoded image representations and multi-granularity text representations for multi-label contrastive learning, employing smoothed KL di-

vergence to align cross-granularity representations. We then describe how to transform multi-granularity text into hierarchical representations and detail the MGLL objective. Furthermore, we provide both empirical and theoretical analyses, demonstrating that MGLL captures richer image-text correlations than CLIP without additional parameters. This enables the learning of more discriminative visual features, improving downstream vision tasks. Finally, we introduce the construction of large-scale multi-granular datasets: MGLL-Fundus and MGLL-Xray.

## 3.2 THE MGLL OBJECTIVES

Most contrastive learning methods rely on the traditional CLIP loss, but our primary goal is to achieve simultaneous multi-label and cross-granularity alignment between image-text pairs. To this end, we improve the standard CLIP loss by introducing the **soft CLIP loss**, the **point-wise loss**, and the **smooth KL (Kullback–Leibler) divergence loss** in our proposed multi-granularity language learning objective. The soft CLIP loss $\mathcal{L}_{\text{sCLIP}}$ enhances the visual encoder by enabling better alignment with multi-label features. The point-wise loss optimizes the alignment of visual features with specific text features at a given granularity, further improving multi-label alignment. The smooth KL divergence loss helps different granularity features converge toward a unified feature space, facilitating cross-granularity alignment of visual representations. To quantify the similarity between image and text features, we adopt a soft alignment strategy, allowing an image $V_i$ to align not only with a single label $T_i$ but also with multiple related labels $T_{ik}, k \in \{1, 2, \ldots, M_i\}$ as Eqs. (1) and (2), where $N$ is the total number of images, $M_i$ is the number of text labels associated with the $i$-th image, $V_i$ and $T_{ik}$ represent the encoded features of the $i$-th image and its corresponding $k$-th text label, respectively. $\text{sim}(V_i, T_{ik})$ is the similarity function measuring their alignment. The temperature parameter $\tau$ controls the sharpness of the probability distribution, while the weight factor $w_{ik}$ determines the contribution of the $k$-th text label to the alignment of the $i$-th image. The text-to-image loss $l_{ki}$ can be obtained simply by swapping the roles of the image and text terms of the image-to-text loss $l_{ik}$ in Eq. (2).

$$l_{ik} = -w_{ik} \log \frac{\exp(\text{sim}(V_i, T_{ik})/\tau)}{\sum_{n=1}^{N} \sum_{m=1}^{M_n} \exp(\text{sim}(V_i, T_{nm})/\tau)} \tag{1}$$

$$\mathcal{L}_{\text{sCLIP}} = \frac{1}{2 \sum_{i=1}^{N} M_i} \sum_{i=1}^{N} \sum_{k=1}^{M_i} (l_{ik} + l_{ki}) \tag{2}$$

Each image feature $V_i$ is treated as multiple pairs $(V_i, T_{i1}), (V_i, T_{i2}), ..., (V_i, T_{iM_i})$, and $w_{ik}$ is the probability of selecting $T_{ik}$ as a valid label for $V_i$. $w_{ik}$ is derived from the co-occurrence matrix normalization as Eq. (3). Instead of forcing the model to align strictly with one label like CLIP, MGLL allows multi-label optimization and prevents the model from being biased toward a single label.

$$w_{ik} = \frac{\text{coocurrence}(V_i, T_{ik})}{\sum_k \text{coocurrence}(V_i, T_{ik})} \tag{3}$$

To further optimize the alignment between visual and textual features, we employ binary cross entropy as point-wise loss $\mathcal{L}_{\text{P}}$ to refine multi-label alignment as Eq. (4), where $x'_{ij} = \sigma(x_{ij})$, $x_{ij} = \text{sim}(V_i, T_j)$ represents the similarity logits between the encoded image feature $V_i$ and text feature $T_j$ before applying activation. The binary label $y_{ij} \in \{0, 1\}$ indicates whether the image-text pair is a valid match. $\sigma(x)$ is the Sigmoid activation function, defined as $\sigma(x) = \frac{1}{1+e^{-x}}$, which normalizes the logits into a probability range. $T_j$ denotes the annotation corresponding to a single label at a specific granularity level. $M$ denotes the total number of annotations, and $N$ represents the total number of images. These annotations are consistent with those defined in Eq. (1). Since the point-wise loss does not explicitly model the relationships among annotations, we omit the label subscripts of $M$ and $T_j$ for simplicity. By explicitly supervising individual image-text pairs, this loss enhances fine-grained multi-label alignment and improves the discriminability of visual representations.

$$\mathcal{L}_{\text{P}} = -\sum_{i=1}^{N} \sum_{j=1}^{M} \frac{y_{ij} \log x'_{ij} + (1 - y_{ij}) \log(1 - x'_{ij})}{N} \tag{4}$$

To achieve cross-granularity alignment, we employ the smooth Kullback–Leibler (KL) divergence loss $\mathcal{L}_{\text{sKL}}$, formulated as follows. Given $m$ similarity logits between the encoded image feature and the text feature $\{P_i\}_{i=1}^{m}$, we define the mean distribution as the average of all predicted distributions:

$M = \frac{1}{m} \sum_{i=1}^{m} P_i$. Then, we compute the KL divergence between each predicted distribution and the mean distribution as Eqs. (5) and (6), where $P_i^{(j)}$ represents the predicted probability of the $i$-th model for category $j$. This loss encourages consistency across different granularity levels by aligning their predicted distributions toward the mean distribution $M$, which achieves cross-granularity alignment.

$$D_{\text{KL}}(P_i \| M) = \sum_j P_i^{(j)} \log \frac{P_i^{(j)}}{M^{(j)}} \tag{5}$$

$$\mathcal{L}_{\text{sKL}} = \sum_{i=1}^{m} D_{\text{KL}}(P_i \| M) \tag{6}$$

The final loss with weight factors is as Eq. (7), where $\alpha_1$ is 0.5, $\alpha_2$ is 1, and $\alpha_3$ is 1 by experimental setting.

$$\mathcal{L}_{\text{MGLL}} = \alpha_1 \mathcal{L}_{\text{sCLIP}} + \alpha_2 \mathcal{L}_{\text{p}} + \alpha_3 \mathcal{L}_{\text{sKL}} \tag{7}$$

### 3.3 EMPIRICAL AND THEORETICAL ANALYSIS OF MGLL

#### 3.3.1 EMPIRICAL ANALYSIS

CLIP aligns each image with a single text label, limiting its effectiveness in multi-label scenarios. MGLL addresses this by using soft CLIP loss and point-wise loss to align visual features with multiple correlated text labels on a shared manifold. For multi-granularity alignment, MGLL encodes each granularity in separate spaces and aligns image features accordingly. A smooth KL divergence loss further promotes consistency by aligning features across granularities with their mean distribution, preventing overfitting to any single level. This enables MGLL to distinguish both coarse and fine-grained categories (e.g., Glaucoma vs. Diabetic Macular Edema, and Severe vs. Moderate Diabetic Macular Edema), where CLIP typically fails.

#### 3.3.2 THEORETICAL ANALYSIS

We provide a theoretical comparison between MGLL and CLIP. CLIP maximizes similarity between image and corresponding text features while minimizing contrastive loss for mismatched pairs, as defined in Eq. (8), where $V_i$ and $T_i$ are image and text features, $\text{sim}(I, T) = \frac{I \cdot T}{\|I\| \|T\|}$ is cosine similarity, and $\tau$ is a temperature parameter.

$$\mathcal{L}_{\text{CLIP}} = -\frac{1}{N} \sum_{i=1}^{N} \log \frac{\exp(\text{sim}(V_i, T_i)/\tau)}{\sum_{j=1}^{N} \exp(\text{sim}(V_i, T_j)/\tau)} \tag{8}$$

However, CLIP only aligns an image $V_i$ with a single text label $T_i$, limiting its effectiveness in multi-label settings. It also projects text features of different granularities into the same space, which is suboptimal when finer semantic distinctions are needed. MGLL overcomes these issues by introducing Soft CLIP Loss, Point-wise Loss, and Smooth KL Divergence Loss to support multi-label and cross-granularity alignment in appropriate feature subspaces.

**(1) Soft CLIP Loss:** MGLL allows an image feature $V_i$ to align with multiple text features $\{T_{i1}, T_{i2}, ..., T_{iM_i}\}$. At optimality, this leads to the condition in Eq. (9), implying Eq. (10), where $V_i$ converges to the weighted center of its associated text features. This contrasts with CLIP, which aligns each image to a single text feature, highlighting MGLL's advantage in multi-label learning.

$$\sum_{k=1}^{M_i} w_{ik} \nabla_{V_i} \text{sim}(V_i, T_{ik}) = 0 \tag{9}$$

$$\sum_{k=1}^{M_i} w_{ik} \frac{T_{ik}}{\|T_{ik}\|} = \frac{V_i}{\|V_i\|} \tag{10}$$

**(2) Point-wise Loss:** To enhance image-text alignment, we introduce a point-wise binary cross-entropy loss $\mathcal{L}_{\text{p}}$ with its gradient shown in Eq. (11). If $y_{ij} = 1$, the objective is to maximize $\sigma(x_{ij})$, strengthening similarity between $V_i$ and $T_j$. If $y_{ij} = 0$, it minimizes $\sigma(x_{ij})$, suppressing similarity with irrelevant text. This encourages alignment with all valid labels while filtering out noise, improving over CLIP's single-label constraint.

$$\frac{\partial \mathcal{L}_{\mathrm{p}}}{\partial x_{ij}} = \sigma(x_{ij}) - y_{ij} \tag{11}$$

**(3) Smooth KL Divergence Loss:** To enforce cross-granularity consistency, we introduce Smooth KL Divergence Loss. The mean distribution is defined as $M = \frac{1}{m}\sum_{i=1}^{m} P_i$ where $\{P_i\}_{i=1}^{m}$ are predicted distributions. By the non-negativity of KL divergence, Eq. (12) achieves equality only when $P_i = M$. Minimizing $\mathcal{L}_{\mathrm{KL}}$ thus enforces $P_1 = P_2 = \cdots = P_m = M$, encouraging consistent representations across granularities and improving alignment in feature space.

$$D_{\mathrm{KL}}(P_i \| M) \geq 0, \quad \forall i \tag{12}$$

While CLIP optimizes image-text alignment, it overlooks feature variability across granularities and lacks consistency in visual alignment. By aligning each image feature $V_i$ with a single text feature $T_i$, it risks biased representations. In contrast, MGLL drives text features of different granularities toward a shared mean distribution $M$, promoting common semantic grounding and aligning visual features with all granularity levels, not just one.

### 3.4 LARGE-SCALE MULTI-GRANULAR DATASETS

#### 3.4.1 MGLL-FUNDUS DATASET

In this study, we construct a large-scale multi-granularity fundus image-text dataset, MGLL-Fundus, consisting of 246,389 pairs of fundus images and corresponding multi-granularity textual descriptions. The image data in MGLL-Fundus originates from 49 public datasets, covering more than 50 disease categories (details are provided in the supplementary material). The multi-granularity textual descriptions mainly include two levels of granularity: disease category and clinical explanation. The disease-level granularity comprises normal/abnormal labels along with specific disease categories. The clinical explanation granularity provides detailed textual descriptions derived from label explanations in datasets and EyeWiki EyeWiki (2024). As shown in Fig. 2, the disease-level description is "Abnormal, Age-related Macular Degeneration", while its corresponding clinical explanation is "Changes in the retinal pigment epithelium." By incorporating multi-granularity textual descriptions, we establish a hierarchical labeling system for fundus images, including normal/abnormal classification, disease categorization, and detailed clinical descriptions, which enable cross-granularity image-text alignment and enhance performance across different granularity levels. Our multi-granularity approach can also be adopted to other modalities.

#### 3.4.2 MGLL-XRAY DATASET

In radiology research, the heterogeneity of study descriptions in DICOM (Digital Imaging and Communications in Medicine) headers complicates patient cohort selection, especially with manual methods. MIDRC (Medical Imaging and Data Resource Center) MIDRC (2024) highlight this challenge, where over 138,000 studies are categorized into only 97 unique descriptions, while the rest are described by 1,300 different descriptions. Therefore, we need LOINC (Logical Observation Identifiers Names and Codes), which provides a standardized coding system to enhance data sharing and analysis. To facilitate data coordination with, we collect 190,882 X-ray images from the MIDRC repository MIDRC (2024). We convert the images from DICOM to PNG format while extracting key metadata. The extracted multi-granularity textual information includes modality, study description, and series description. Modality includes CR (Computed Radiography), which has a lower resolution and signal-to-noise ratio (SNR), and DX (Digital Radiography), which uses flat-panel detectors for higher-quality imaging. Study Description provides an exam-level overview, such as "Chest X-ray", while Series Description details specific imaging sequences like "PA View" (posteroanterior) or "Lateral View". These multi-granularity textual features serve as the textual component of MGLL-Xray dataset.

## 4 EXPERIMENTS

### 4.1 SETUP

We construct a large-scale multi-granularity fundus image-text dataset for pre-training, with further details provided in the supplementary material. To evaluate our model's performance, we conduct experiments on eleven downstream datasets: FIVES Jin et al. (2022), IDRiD Porwal et al. (2018), OIA-DDR Li et al. (2019b), ADAM Fang et al. (2022), PALM Fang et al. (2024), REFUGE Orlando et al. (2020), RIM-ONE Batista et al. (2020), RFMiD Pachade et al. (2021), MIDRC-XR MIDRC (2024), MIDRC-XR-Portable MIDRC (2024), ChestX-ray14 Wang et al. (2017a) under both linear

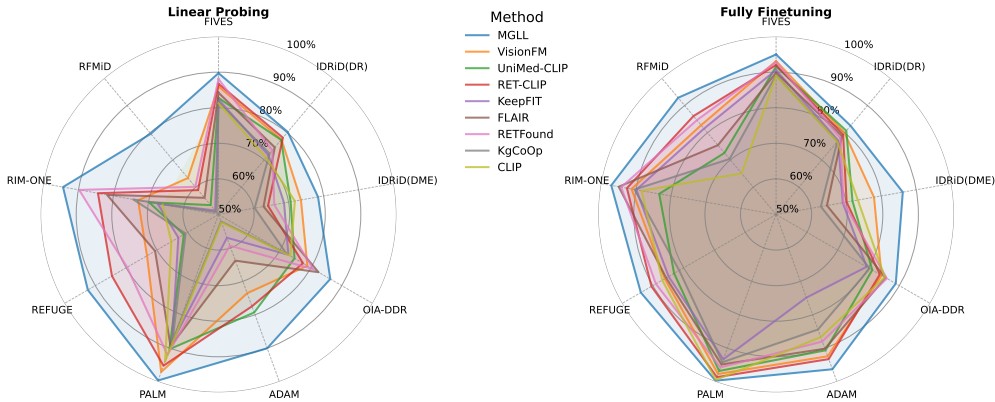

Figure 3: The quantitative comparison (AUC) between baseline methods and proposed MGLL on nine fundus downstream datasets.

probing and full fine-tuning settings. In our quantitative evaluation, we employ AUC (Area Under the receiver operating characteristic Curve), mAP (mean Average Precision), and ACC (Accuracy) as assessment metrics. As for the multi-label setting, we report the category-wise average accuracy as ACC. We adopt ViT-L/14 Dosovitskiy et al. (2020) as the image encoder and BiomedicalBERT Alsentzer et al. (2019) as the text encoder by default. All experiments were conducted under identical settings, with baselines pre-trained on our self-constructed multi-granularity datasets to ensure fair comparison. We strictly followed the official data splits of all downstream datasets. During pretraining, we only used the training sets for model training and the validation sets for pretraining evaluation, while the test sets were never accessed.

## 4.2 COMPARISON WITH STATE-OF-THE-ART METHODS

### 4.2.1 EVALUATION ON RETINAL FUNDUS DATASETS

Utilizing the multi-granularity image-text fundus dataset we constructed, we pretrain our model within the Multi-Granular Language Learning (MGLL) framework to enhance its capability in feature representation for retinal fundus images. We conduct comprehensive experiments to compare the performance of MGLL against several state-of-the-art (SOTA) baseline methods across nine downstream datasets, covering a wide range of retinal diseases. The AUC results for both linear probing and full fine-tuning are presented in Fig. 3, while more detailed results can be found in the supplementary material (Tables 15 to 23). MGLL consistently achieves significant performance improvements across all nine datasets, with particularly strong gains in the linear probing setting. Notably, on the multi-label dataset RFMiD Pachade et al. (2021), MGLL outperforms other methods by at least 16.6% in linear probing and 6.7% in full fine-tuning, demonstrating its superior capability in handling imbalanced data distributions. Fig. 4 visualizes class activation maps (CAMs) from CLIP and MGLL on two cases with different retinal diseases. It is evident that CLIP fails to extract meaningful features, instead assigning nearly uniform attention weights across the entire fundus image. In contrast, MGLL effectively localizes key regions of interest (ROIs) for different diseases. Specifically, MGLL accurately highlights hard exudates for chorioretinitis and the retinal pigment epithelium for age-related macular degeneration. These quantitative and qualitative evaluations collectively indicate that MGLL possesses extraordinary capability in effective feature extraction and performance enhancement across diverse retinal diseases.

### 4.2.2 EVALUATION ON X-RAY DATASETS

We pretrain MGLL on the MGLL-Xray dataset and conduct experiments on MGLL and other SOTA baseline methods (Radford et al. (2021); Tiu et al. (2022); Zhang et al. (2023); Zhou et al. (2023a); Dai et al. (2024); Khattak et al. (2024); Lai et al. (2024); Xie et al. (2025)) on the MIDRC-XR and MIDRC-XR-Portable datasets, which are shown in the Table 1. In the linear probe setting, MGLL achieves significant advancements over the second-best method (UniChest Dai et al. (2024) on MIDRC-XR and UniMed-CLIP Khattak et al. (2024) on MIDRC-XR-Portable), with improvements of 2.23% and 3.81% in AUC, respectively, indicating superior representation learning capabilities. The performance gap becomes even more significant in the fully fine-tuned setting. To demonstrate its generalization capability, we conduct additional experiments using multi-granular

Table 1: The performance evaluation on MIDRC-XR, MIDRC-XR-Portable, and ChestX-ray14. **Bold** indicates best performance and underline shows second-best.

| Method | MIDRC-XR | | | | | | MIDRC-XR-Portable | | | | | | ChestX-ray14 | | | | | |
|---|---|---|---|---|---|---|---|---|---|---|---|---|---|---|---|---|---|---|
| | Linear Probe (%) | | | Fully Fine-tune (%) | | | Linear Probe (%) | | | Fully Fine-tune (%) | | | Linear Probe (%) | | | Fully Fine-tune (%) | | |
| | AUC | ACC | mAP | AUC | ACC | mAP | AUC | ACC | mAP | AUC | ACC | mAP | AUC | ACC | mAP | AUC | ACC | mAP |
| CLIP (ICML-21) | 54.72 | 51.18 | 16.62 | 88.52 | 80.83 | 62.04 | 71.43 | 78.22 | 22.31 | 91.83 | 90.08 | 83.94 | 69.75 | 78.09 | 18.33 | 82.05 | 87.58 | 31.79 |
| CheXzero (Nat. BME-22) | 51.31 | 43.85 | 12.71 | 80.11 | 73.26 | 55.46 | 72.84 | 80.13 | 23.56 | 92.47 | 92.42 | 85.23 | 68.72 | 76.98 | 15.97 | 81.81 | 87.39 | 31.52 |
| KAD (Nat. Com-23) | 53.44 | 47.13 | 14.86 | 85.74 | 78.39 | 60.12 | 73.53 | 80.71 | 23.88 | 93.41 | 92.98 | 85.96 | 73.72 | 78.95 | 21.87 | 83.80 | 89.13 | 34.01 |
| MRM (ICLR-23) | 56.23 | 53.61 | 17.73 | 90.67 | 83.95 | 64.76 | 79.38 | 86.05 | 27.72 | 96.52 | 95.07 | 86.95 | 74.63 | 79.87 | 23.23 | 84.28 | 89.57 | 35.62 |
| UniChest (TMI-24) | 59.02 | 54.78 | 19.32 | 92.51 | 86.32 | 66.93 | 78.49 | 85.28 | 27.37 | 95.44 | 94.32 | 86.38 | 76.15 | 81.72 | 25.52 | 85.84 | 89.99 | 37.97 |
| UniMed-CLIP (arXiv-24) | 57.33 | 54.07 | 18.06 | 94.15 | 87.47 | 68.49 | 80.05 | 86.63 | 28.16 | 94.31 | 93.55 | 86.19 | 75.54 | 81.21 | 24.96 | 82.59 | 88.36 | 32.19 |
| CARZero (CVPR-24) | 57.92 | 54.43 | 18.79 | 93.48 | 86.94 | 67.62 | 75.24 | 82.67 | 25.65 | 92.94 | 92.66 | 85.67 | 77.32 | 83.94 | 26.88 | 82.95 | 88.65 | 32.86 |
| FG-CLIP (ICML-25) | 58.31 | 54.59 | 19.03 | 93.29 | 86.71 | 67.44 | 80.31 | 86.77 | 28.27 | 96.93 | 95.74 | 87.42 | 76.62 | 82.35 | 25.98 | 85.10 | 89.73 | 37.02 |
| MGLL | **61.25** | **56.57** | **21.19** | **99.08** | **90.06** | **73.33** | **83.86** | **89.06** | **30.62** | **99.75** | **98.80** | **89.87** | **82.94** | **90.41** | **28.53** | **87.37** | **92.71** | **39.17** |

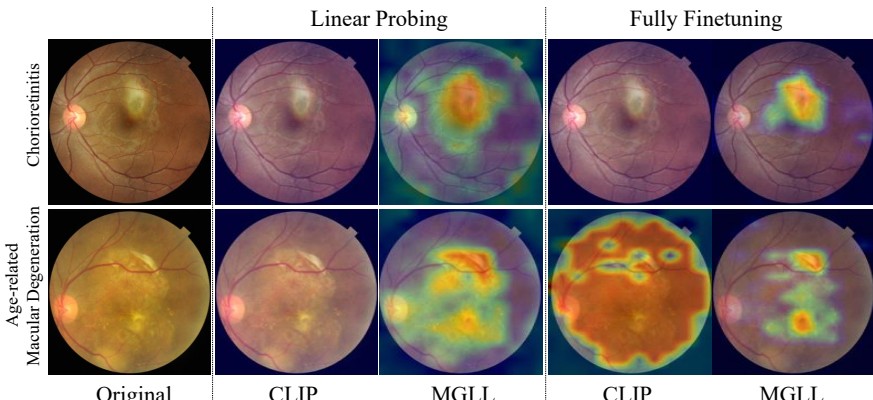

Figure 4: The Class Activation Maps of different diseases from CLIP and MGLL.

labels constructed from the MIMIC-CXR dataset Johnson et al. (2019), evaluating performance on the ChestX-ray14 benchmark Wang et al. (2017a). The results reveal its exceptional transferability, with substantial performance advantages across all other baseline methods. In the linear probe setting, MGLL achieves 82.94% AUC, 90.41% accuracy, and 28.53% mAP, surpassing the second-best method (CARZero Lai et al. (2024)) by 5.62%, 6.47%, and 1.65% respectively. The improvement highlights its superior representation learning capacity, suggesting that its multi-granular approach captures more generalizable features that transfer effectively across datasets. These consistent and substantial improvements also demonstrate that MGLL enables more robust feature extraction.

### 4.3 PERFORMANCE WITH MGLL IN MLLMS

To evaluate MGLL's impact as a specialized vision encoder within multimodal large language models (MLLMs) for ophthalmological diagnostics, we design a multiple-choice benchmark involving 2,233 clinical cases over ten ophthalmological conditions, where each fundus image prompted models to select the correct diagnosis from four options (one correct, three random alternatives). We replace the standard vision encoders in seven advanced MLLMs with our pretrained MGLL: InstructBLIP Dai et al. (2023), Mini-Gemini Li et al. (2024), Qwen-VL Bai et al. (2023), InternVL Chen et al. (2024a), LLaVA Liu et al. (2023), LLaVA-Med Li et al. (2023), Med-Flamingo Moor et al. (2023), and Janus-Pro Chen et al. (2025). All MLLMs were fine-tuned on the target dataset to ensure a fair comparison. Results demonstrate consistent and substantial improvements across all tested MLLMs, with average accuracy gains ranging from 4.6% (InternVL) to 34.1% (LLaVA-Med) as shown in Table 2. Notably, medically-specialized models exhibited the most dramatic enhancements, with Med-Flamingo and LLaVA-Med showing 31.7% and 34.1% increases respectively. This dramatic improvement can be attributed to the alignment between MGLL's ophthalmology-specific visual feature extraction capabilities and the medical reasoning frameworks already embedded within these models. Even the high-performing general-purpose MLLMs like LLaVA (72.73% to 79.98%) also achieve significant gains with MGLL. The improvements across challenging conditions like Tessellation and Retinitis underscore MGLL's capacity to extract clinically relevant visual features from fundus images, highlighting its robust adaptability across models.

Table 2: Comparison of multiple-choice accuracy with MGLL in multimodal large language models on selected ten representative diseases.

| Method | AMD | Cataract | CSR | DR | Glaucoma | Media Haze | Myopia | Retinitis | DME | Tessellation | Average ↑ |
|---|---|---|---|---|---|---|---|---|---|---|---|
| InstructBLIP Dai et al. (2023) | 80.17% | 80.00% | 0.00% | 76.51% | 59.30% | 16.13% | 44.25% | 11.11% | 63.79% | 41.67% | 47.29% |
| + MGLL | **83.63%** | **85.00%** | **28.57%** | **82.55%** | **65.43%** | **45.16%** | **52.65%** | **44.44%** | **74.14%** | **58.33%** | **61.99% (14.7% ↑)** |
| Mini-Gemini Li et al. (2024) | 76.61% | 85.00% | 14.29% | 79.87% | 67.90% | 38.71% | 58.41% | 33.33% | 60.34% | 33.33% | 54.78% |
| + MGLL | **82.46%** | **85.00%** | **42.86%** | **84.56%** | **72.22%** | **58.06%** | **64.16%** | **55.56%** | **65.52%** | **41.67%** | **65.21% (10.4% ↑)** |
| Qwen-VL Bai et al. (2023) | 81.87% | 75.00% | 28.57% | 80.54% | 78.40% | 54.84% | 76.55% | 22.22% | 84.48% | 25.00% | 60.75% |
| + MGLL | **85.96%** | **80.00%** | **42.86%** | **89.93%** | **87.04%** | **70.97%** | **80.97%** | **33.33%** | **89.66%** | **41.67%** | **70.24% (9.5% ↑)** |
| InternVL Chen et al. (2024a) | 81.29% | 85.00% | 71.43% | 94.63% | 89.51% | 64.52% | 88.05% | 44.44% | 87.93% | 66.67% | 77.35% |
| + MGLL | **86.55%** | **90.00%** | **71.43%** | **96.64%** | **90.74%** | **67.74%** | **91.15%** | **55.56%** | **94.83%** | **75.00%** | **81.96% (4.6% ↑)** |
| LLaVA Liu et al. (2023) | 83.04% | 90.00% | 42.86% | 87.25% | 91.36% | 48.39% | 88.50% | 44.44% | 93.10% | 58.33% | 72.73% |
| + MGLL | **84.80%** | **90.00%** | **57.14%** | **93.96%** | **91.98%** | **61.29%** | **90.71%** | **66.67%** | **96.55%** | **66.67%** | **79.98% (7.3% ↑)** |
| LLaVA-Med Li et al. (2023) | 16.37% | 15.00% | 42.86% | 26.85% | 25.31% | 25.81% | 23.89% | 33.33% | 16.67% | 16.67% | 24.28% |
| + MGLL | **58.48%** | **65.00%** | **57.14%** | **77.18%** | **59.26%** | **51.61%** | **57.08%** | **44.44%** | **55.17%** | **58.33%** | **58.37% (34.1% ↑)** |
| Med-Flamingo Moor et al. (2023) | 25.73% | 30.00% | 57.14% | 36.91% | 24.07% | 22.58% | 18.58% | 22.22% | 24.14% | 8.33% | 26.97% |
| + MGLL | **69.01%** | **75.00%** | **71.43%** | **80.54%** | **61.11%** | **54.84%** | **45.58%** | **44.44%** | **51.72%** | **33.33%** | **58.70% (31.7% ↑)** |
| Janus-Pro Chen et al. (2025) | 88.30% | 75.00% | 42.86% | 93.29% | 90.74% | 58.06% | 87.17% | 33.33% | 62.07% | 58.33% | 68.92% |
| + MGLL | **90.64%** | **85.00%** | **71.43%** | **96.64%** | **95.06%** | **67.74%** | **90.27%** | **55.56%** | **70.69%** | **75.00%** | **79.80% (10.88% ↑)** |

## 4.4 ABLATION STUDIES

### 4.4.1 ABLATION STUDY ON MGLL OBJECTIVES

We conduct an ablation study to analyze the effectiveness of each objective in MGLL on the RFMiD dataset, as shown in Table 3. The standard CLIP model performs the worst, highlighting its limitations in medical image understanding. Incorporating the point-wise Loss $\mathcal{L}_P$ significantly improves performance, demonstrating its ability to enhance feature extraction. The soft CLIP loss $\mathcal{L}_{sCLIP}$ also improves over CLIP, which enables soft alignment with multiple labels. Combining both losses ($\mathcal{L}_{sCLIP} + \mathcal{L}_P$) further boosts performance, indicating their complementary effects. Finally, adding the soft-KL loss $\mathcal{L}_{sKL}$ leads to the best performance, demonstrating its role in refining feature consistency across different learning objectives. These results validate the effectiveness of each objective.

Table 3: Ablations of different MGLL objectives on RFMiD.

| Method | Linear Probe (%) | | | Fully Fine-tune (%) | | |
|---|---|---|---|---|---|---|
| | AUC | ACC | mAP | AUC | ACC | mAP |
| CLIP | 44.66 | 92.53 | 7.28 | 65.10 | 92.86 | 17.31 |
| $\mathcal{L}_P$ | 70.34 | 92.69 | 23.83 | 88.25 | 94.31 | 56.46 |
| $\mathcal{L}_{sCLIP}$ | 67.86 | 92.63 | 20.52 | 85.13 | 93.58 | 50.67 |
| $\mathcal{L}_{sCLIP} + \mathcal{L}_P$ | 75.73 | 92.77 | 30.16 | 90.31 | 94.87 | 62.27 |
| $\mathcal{L}_{sCLIP} + \mathcal{L}_P + \mathcal{L}_{sKL}$ | **79.62** | **92.84** | **34.08** | **92.83** | **95.48** | **64.99** |

Table 4: Ablations of granularity count on MIDRC-XR-Portable.

| Method | Linear Probe (%) | | | Fully Fine-tune (%) | | |
|---|---|---|---|---|---|---|
| | AUC | ACC | mAP | AUC | ACC | mAP |
| CLIP | 71.43 | 78.22 | 22.31 | 91.83 | 90.08 | 83.94 |
| $MGLL_1$ | 80.54 | 86.97 | 28.32 | 95.96 | 94.66 | 86.54 |
| $MGLL_2$ | 82.92 | 88.35 | 29.43 | 97.26 | 96.84 | 87.68 |
| $MGLL_3$ | **83.86** | **89.06** | **30.62** | **99.75** | **98.80** | **89.87** |

Table 5: Ablations of image encoder on RFMiD.

| Method | Linear Probe (%) | | | Fully Fine-tune (%) | | |
|---|---|---|---|---|---|---|
| | AUC | ACC | mAP | AUC | ACC | mAP |
| CLIP | 44.66 | 92.53 | 7.28 | 65.10 | 92.86 | 17.31 |
| ConvNext-Base | 74.45 | 92.73 | 27.94 | 88.55 | 94.57 | 57.62 |
| ConvNext-Large | 78.34 | 92.80 | 32.03 | 91.29 | 94.95 | 62.93 |
| ViT-B/16 | 75.53 | 92.76 | 30.11 | 89.46 | 94.83 | 61.84 |
| ViT-L/14 | **79.62** | **92.84** | **34.08** | **92.83** | **95.48** | **64.99** |
| ViT-H/14 | 79.18 | 92.81 | 33.42 | 92.07 | 95.29 | 63.85 |

Table 6: Ablations of text encoder on RFMiD.

| Method | Linear Probe (%) | | | Fully Fine-tune (%) | | |
|---|---|---|---|---|---|---|
| | AUC | ACC | mAP | AUC | ACC | mAP |
| CLIP | 44.66 | 92.53 | 7.28 | 65.10 | 92.86 | 17.31 |
| $CLIP_{text}$ | 68.93 | 92.66 | 22.14 | 88.76 | 94.58 | 58.14 |
| BERT | **79.62** | **92.84** | **34.08** | **92.83** | **95.48** | **64.99** |
| LLaMA | 74.89 | 92.75 | 29.38 | 90.97 | 95.04 | 62.86 |

### 4.4.2 ABLATION STUDY ON GRANULARITY COUNT

The ablation study on granularity count demonstrates the significant impact of multi-granular language supervision on model performance. As shown in Table 4, incrementally increasing the number of granularity levels consistently improves performance across all evaluation metrics. $MGLL_3$, which utilizes three distinct granularity levels (modality, study description, and series description), achieves superior results compared to both $MGLL_1$ (all textual information combined into a single granularity) and $MGLL_2$ (two granularity levels). Specifically, under linear probe evaluation, $MGLL_3$ outperforms the baseline CLIP by substantial margins (+12.43% AUC, +10.84% ACC, +8.31% mAP) and shows marked improvement over $MGLL_1$ (+3.32% AUC, +2.09% ACC, +2.30% mAP). The similar trend persists in fully fine-tuned scenarios. These results confirm that preserving the hierarchical structure of medical imaging information enables more comprehensive vision-language alignment than flattened representations, validating the core idea of the Multi-Granular Language Learning framework.

### 4.4.3 SELECTION OF IMAGE ENCODER AND TEXT ENCODER

The ablation study on image encoders reveals significant performance variations across different architectural choices when evaluated on the RFMiD dataset as shown in Table 5. Vision Transformer (ViT) Dosovitskiy et al. (2020) generally outperforms CNN counterparts (ConvNeXt Liu et al. (2022b)), with ViT-L/14 achieving optimal results across all metrics. Interestingly, the larger ViT-H/14 model shows slightly diminished performance compared to ViT-L/14, suggesting a potential overfitting scenario or diminishing returns with increased model complexity in this domain.

The comparative analysis of text encoders demonstrates that the choice of language model significantly impacts the model's ability to align textual and visual representations. BERT emerges as the optimal text encoder, achieving the highest performance across all evaluation metrics as shown in Table 6. The standard CLIP text encoder (denoted as $CLIP_{text}$) shows the limited performance among the tested alternatives, though it still substantially improves upon the baseline CLIP model. These findings suggest that the bidirectional attention mechanisms are suitable for the structured, hierarchical medical terminology utilized in MGLL.

### 4.4.4 ABLATION STUDY ON IMAGE QUALITY AND TEXT QUALITY

Image resolution is a critical factor in the performance of MGLL as evidenced by the ablation experiments on the MIDRC-XR-Portable dataset. The performance exhibits a clear monotonic relationship with image resolution, with Standard-Resolution (512×512) significantly outperforming both Low-Resolution (128×128) and Ultra Low-Resolution (64×64) configurations across all evaluation metrics as shown in Table 7. These findings underscore the importance of preserving fine-grained visual details in medical imaging applications, as higher resolution allows the model to capture subtle radiological features. However, MGLL substantially outperforms the baseline CLIP even at Ultra Low-Resolution, suggesting that MGLL provides robust improvements regardless of image quality.

The integrity of textual information significantly impacts model performance, as demonstrated through controlled degradation experiments on the MIDRC-XR-Portable dataset. The analysis contrasts standard textual descriptions against two degraded conditions: "Error" (20% partial errors in modality, study, or series descriptions) and "Missing" (20% partial omissions in these same fields). Standard textual descriptions yield superior performance across all metrics as show in Table 8. The "Missing" condition demonstrates intermediate performance, while the "Error" condition shows more performance degradation, suggesting that incorrect information is more detrimental than incomplete information. Nevertheless, both degraded conditions still significantly outperform the baseline CLIP model, indicating the robustness of MGLL to textual noise. These findings have important practical implications for clinical deployment scenarios, where reporting systems may contain documentation gaps or transcription errors, and suggest that MGLL maintains considerable diagnostic utility even under suboptimal documentation conditions.

Table 7: Ablations of image quality on MIDRC-XR-Portable.

| Method | Linear Probe (%) | | | Fully Fine-tune (%) | | |
|---|---|---|---|---|---|---|
| | AUC | ACC | mAP | AUC | ACC | mAP |
| CLIP | 71.43 | 78.22 | 22.31 | 91.83 | 90.08 | 83.94 |
| Ultra Low-Res | 78.82 | 85.68 | 27.49 | 94.48 | 93.69 | 86.22 |
| Low-Res | 80.66 | 87.02 | 28.53 | 98.18 | 97.65 | 88.46 |
| Standard-Res | **83.86** | **89.06** | **30.62** | **99.75** | **98.80** | **89.87** |

Table 8: Ablations of text quality on MIDRC-XR-Portable.

| Method | Linear Probe (%) | | | Fully Fine-tune (%) | | |
|---|---|---|---|---|---|---|
| | AUC | ACC | mAP | AUC | ACC | mAP |
| CLIP | 71.43 | 78.22 | 22.31 | 91.83 | 90.08 | 83.94 |
| Error | 80.02 | 86.48 | 28.01 | 97.71 | 97.22 | 87.96 |
| Missing | 81.14 | 87.25 | 28.86 | 98.62 | 97.95 | 88.74 |
| Standard | **83.86** | **89.06** | **30.62** | **99.75** | **98.80** | **89.87** |

## 5 CONCLUSION

This study introduces Multi-Granular Language Learning (MGLL), a novel contrastive learning framework that addresses limitations in existing vision-language pretraining methods. MGLL utilizes textual information on different granular levels while employing soft-label supervision with point-wise constraints to enhance representation quality, which advances multi-label and cross-granularity alignment capabilities simultaneously. The implementation of smooth Kullback–Leibler divergence also ensures cross-granularity consistency. Our evaluations across multiple downstream datasets demonstrate that MGLL consistently outperforms state-of-the-art methods in downstream tasks, particularly in domains requiring multi-label understanding at various granular levels. The results validate its ability to capture complex semantics in visual data, establishing MGLL as an advancement in developing future vision-language models.

**Ethics statement** This research uses exclusively publicly available medical imaging datasets and does not require Institutional Review Board (IRB) approval. All datasets utilized in this study, including the 49 public fundus imaging datasets comprising MGLL-Fundus, the MIDRC repository for X-ray images, MIMIC-CXR, and other evaluation datasets, have been previously released for research purposes with appropriate ethical clearances and patient consent procedures handled by the original data providers. All patient identifiers have been removed from the datasets prior to our access, ensuring full de-identification in compliance with HIPAA and other relevant privacy regulations. We have strictly adhered to the usage terms and conditions specified by each dataset provider and have not attempted to re-identify any individuals. Our multi-granularity learning approach is designed to improve automated medical image analysis, particularly for diabetic retinopathy, glaucoma, age-related macular degeneration, and chest X-ray interpretation. The MIDRC dataset applications focus on enhancing radiological assessment capabilities, which could potentially assist healthcare providers in resource-limited settings and improve diagnostic consistency. However, we emphasize that our models are intended as diagnostic support tools and should not replace clinical judgment. We commit to responsible AI development by ensuring transparent reporting of model limitations, encouraging rigorous clinical validation before any real-world deployment, and advocating for appropriate human oversight in all clinical applications. We do not claim our models are ready for direct clinical use without further validation and regulatory approval.

**Reproducibility statement** We have made extensive efforts to ensure the reproducibility of our work. The complete implementation, including the original source code and a README file with detailed instructions, is provided in the supplementary materials. Training configurations and hyperparameters are fully documented in both the source code and Section 4.1. Step-by-step mathematical derivations of the proposed methodology are presented in Section 3.3 and Appendix A. All datasets employed in this study are publicly available, with references and preprocessing procedures described in Section 3.4 and Appendix B. The experimental setup is described in Section 4.1 and more details are described in Appendix C. To ensure fair comparison and reproducibility, all experiments used identical settings with baseline models pre-trained on our custom multi-granularity datasets.

**Acknowledgments** Research reported is part of the MIDRC (Medical Imaging and Data Resource Center) project. MIDRC is funded by the National Institute of Biomedical Imaging and Bioengineering (NIBIB) of the National Institutes of Health and ARPA-H under contract 75N92020D00021.

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

# APPENDIX

# Table of Contents

# A DETAILED THEORETICAL ANALYSIS OF MGLL

## A.1 SOFT CLIP LOSS

*Formulation of the Soft CLIP Loss* In contrast to standard CLIP, which forces an image representation $V_i$ to align with a single text label $T_i$. MGLL uses the Soft CLIP Loss to allow an image to align with multiple text features $\{T_{i1}, T_{i2}, \ldots, T_{iM_i}\}$. In doing so, the loss is designed so that the optimal image feature becomes the weighted "center" of its associated text features. This mitigates bias toward any single label and better captures the multi-label nature of the data. The loss for an image–label pair is defined as the following formula:

$$l_{ik} = -w_{ik} \log \frac{e^{\mathrm{sim}(V_i, T_{ik})/\tau}}{\sum_{n=1}^{N} \sum_{m=1}^{M_n} e^{\mathrm{sim}(V_i, T_{nm})/\tau}} \tag{13}$$

where $w_{ik}$ is the weight assigned to the $k$-th text label for image $i$, derived via the following formula:

$$w_{ik} = \frac{\mathrm{cooccurrence}(V_i, T_{ik})}{\sum_k \mathrm{cooccurrence}(V_i, T_{ik})} \tag{14}$$

where $\mathrm{sim}(V_i, T_{ik})$ measures the similarity between image and text features (typically cosine similarity). $\tau$ is the temperature parameter controlling the sharpness of the resulting probability distribution. The overall loss is given by

$$\mathcal{L}_{\mathrm{sCLIP}} = \frac{1}{2 \sum_{i=1}^{N} M_i} \sum_{i=1}^{N} \sum_{k=1}^{M_i} \left( l_{ik} + l_{ki} \right) \tag{15}$$

*Derivation of the Optimality Condition*

### (a) The Goal of the Optimization

For the purposes of our analysis, we focus on how the loss aligns $V_i$ with its multiple text features $T_{ik}$. At an optimum, the gradient of the loss with respect to the image feature $V_i$ must vanish. That is, we require

$$\nabla_{V_i} \mathcal{L}_{\mathrm{sCLIP}} = 0. \tag{16}$$

Focusing on the part of the loss involving the alignment between $V_i$ and its labels, we can write a simplified optimality condition (ignoring symmetric contributions from text-to-image terms):

$$\sum_{k=1}^{M_i} w_{ik} \nabla_{V_i} \mathrm{sim}(V_i, T_{ik}) = 0. \tag{17}$$

### (b) Computing the Gradient

Assume for simplicity that both image and text features are normalized to unit norm:

$$\|V_i\| = 1 \quad \text{and} \quad \|T_{ik}\| = 1. \tag{18}$$

Under this assumption, the cosine similarity reduces to a dot product:

$$\mathrm{sim}(V_i, T_{ik}) = V_i \cdot T_{ik}. \tag{19}$$

The derivative with respect to $V_i$ is then straightforward:

$$\nabla_{V_i}(V_i \cdot T_{ik}) = T_{ik}. \tag{20}$$

Thus, the optimality condition becomes

$$\sum_{k=1}^{M_i} w_{ik} \, T_{ik} = \lambda \, V_i \,, \tag{21}$$

where $\lambda$ is a scalar (a Lagrange multiplier that arises from the normalization constraint on $V_i$). Because $V_i$ is unit norm, this equation can be re-arranged to yield

$$V_i = \frac{\sum_{k=1}^{M_i} w_{ik} \, T_{ik}}{\left\| \sum_{k=1}^{M_i} w_{ik} \, T_{ik} \right\|} \,. \tag{22}$$

This result shows that at the optimal solution, the visual feature $V_i$ is aligned with the weighted average (or the "center") of its associated text features.

**(c) Rewriting in Normalized Form**

More generally, even when the features are not explicitly normalized in the network, one can express the optimality condition in terms of normalized vectors:

$$\sum_{k=1}^{M_i} w_{ik} \frac{T_{ik}}{\|T_{ik}\|} = \frac{V_i}{\|V_i\|} \,. \tag{23}$$

This is equivalent to the expression provided earlier, emphasizing that $V_i$ converges to the weighted centroid of the normalized text features.

*Interpretation and Discussion* Unlike CLIP, which uses a one-to-one image–text matching—the Soft CLIP Loss allows each image to be simultaneously aligned with multiple text descriptions. The weighting $w_{ik}$ (derived from the co-occurrence matrix) ensures that each text label contributes to the final image representation in proportion to its relevance. The optimality condition guarantees that the image representation $V_i$ is not overly biased by any single text feature but is instead the "center" of all its semantic descriptors. This is crucial in multi-label settings where an image may contain several objects or concepts. In standard CLIP, the loss encourages $V_i$ to align closely with a single text label $T_i$. Here, the Soft CLIP Loss's optimality condition shows that the ideal $V_i$ is instead a weighted aggregate of multiple text labels, overcoming the limitation of forcing one-to-one alignment in a multi-label context. The detailed derivation shows that under the Soft CLIP Loss, the following gradient condition

$$\sum_{k=1}^{M_i} w_{ik} \, \nabla_{V_i} \mathrm{sim}(V_i, T_{ik}) = 0 \tag{24}$$

leads directly to the interpretation that the optimal image feature $V_i$ is the normalized weighted sum of its associated text features. This theoretical result underpins MGLL's capability to perform multi-label alignment, ensuring that image representations capture the combined semantic information provided by multiple labels.

A.2   POINT-WISE LOSS

*Formulation of the Point-wise Loss* The point-wise loss $\mathcal{L}_\mathrm{P}$ is designed to refine the alignment between visual and textual features on a per-pair basis. Unlike global or batch-level losses, this loss explicitly supervises each image–text pair, ensuring that every valid pair (where $y_{ij} = 1$) is pulled closer together in the feature space while non-matching pairs (where $y_{ij} = 0$) are pushed apart. This detailed supervision is achieved via a binary cross-entropy formulation applied to the similarity logits between an image feature $V_i$ and a text feature $T_j$. The point-wise loss is defined as

$$\mathcal{L}_\mathrm{P} = \sum_{i=1}^{N} \sum_{j=1}^{M} \frac{y_{ij} \log x'_{ij} + (1 - y_{ij}) \log(1 - x'_{ij})}{-1 \times N} \tag{25}$$

where $x_{ij} = \mathrm{sim}(V_i, T_j)$ are the similarity logits (before activation) between the $i$-th image and the $j$-th text. $x'_{ij} = \sigma(x_{ij}) = \frac{1}{1+e^{-x_{ij}}}$ is the probability computed by the Sigmoid activation. $y_{ij} \in$

$\{0, 1\}$ is the binary label indicating whether the image-text pair is a valid match. This formulation is typical for binary classification tasks and allows the network to learn to distinguish relevant from irrelevant image–text pairs.

*Derivation of the Gradient* To understand how the loss drives the optimization, we derive the gradient with respect to the logits $x_{ij}$.

**(a) Loss for a Single Pair**

Consider the binary cross-entropy loss for a single image–text pair $(i, j)$:

$$\ell_{ij} = - \left[ y_{ij} \log \sigma(x_{ij}) + (1 - y_{ij}) \log(1 - \sigma(x_{ij})) \right] \tag{26}$$

**(b) Derivative with Respect to $x_{ij}$**

We can let $p_{ij} = \sigma(x_{ij})$. Using the chain rule, the derivative of $\ell_{ij}$ with respect to $x_{ij}$ is:

$$\frac{\partial \ell_{ij}}{\partial x_{ij}} = \frac{\partial \ell_{ij}}{\partial p_{ij}} \cdot \frac{dp_{ij}}{dx_{ij}} \tag{27}$$

Step 1. Compute $\frac{\partial \ell_{ij}}{\partial p_{ij}}$. We can differentiate $\ell_{ij}$ with respect to $p_{ij}$ as follows:

$$\frac{\partial \ell_{ij}}{\partial p_{ij}} = - \left( \frac{y_{ij}}{p_{ij}} - \frac{1 - y_{ij}}{1 - p_{ij}} \right) \tag{28}$$

Step 2. Compute $\frac{dp_{ij}}{dx_{ij}}$. Since $p_{ij} = \sigma(x_{ij})$ and the derivative of the sigmoid function is as follows:

$$\frac{d\sigma(x_{ij})}{dx_{ij}} = \sigma(x_{ij})(1 - \sigma(x_{ij})) = p_{ij}(1 - p_{ij}) \tag{29}$$

So we have

$$\frac{dp_{ij}}{dx_{ij}} = p_{ij}(1 - p_{ij}) \tag{30}$$

Step 3. Combine the Derivatives. We can multiply the two derivatives gives as follow:

$$\frac{\partial \ell_{ij}}{\partial x_{ij}} = - \left( \frac{y_{ij}}{p_{ij}} - \frac{1 - y_{ij}}{1 - p_{ij}} \right) \cdot p_{ij}(1 - p_{ij}) \tag{31}$$

Expanding and simplifying, we have:

$$\frac{\partial \ell_{ij}}{\partial x_{ij}} = - \left( y_{ij}(1 - p_{ij}) - (1 - y_{ij})p_{ij} \right) \tag{32}$$

Distributing the negative sign results in:

$$\frac{\partial \ell_{ij}}{\partial x_{ij}} = p_{ij} - y_{ij} \tag{33}$$

Thus, for each image–text pair, the gradient is:

$$\frac{\partial \ell_{ij}}{\partial x_{ij}} = \sigma(x_{ij}) - y_{ij} \tag{34}$$

Since the overall point-wise loss $\mathcal{L}_{\mathrm{P}}$ is the average over all pairs, the gradient with respect to each $x_{ij}$ remains the same:

$$\frac{\partial \mathcal{L}_{\mathrm{P}}}{\partial x_{ij}} = \sigma(x_{ij}) - y_{ij} \tag{35}$$

*Interpretation and Discussion* This gradient expression, $\sigma(x_{ij}) - y_{ij}$, provides clear insights into the optimization dynamics:

**For a Positive Pair** ($y_{ij} = 1$): The gradient becomes $\sigma(x_{ij}) - 1$. If the predicted probability $\sigma(x_{ij})$ is less than 1, the gradient is negative, prompting an increase in $x_{ij}$ (and hence $\sigma(x_{ij})$). This drives the features $V_i$ and $T_j$ closer together.

**For a Negative Pair** ($y_{ij} = 0$): The gradient simplifies to $\sigma(x_{ij})$. If $\sigma(x_{ij})$ is positive (which it always is, since $\sigma(x) \in (0,1)$), the gradient is positive, pushing $x_{ij}$ downward. This reduces the similarity, ensuring that irrelevant image–text pairs are further separated.

Thus, the gradient directs the model to increase similarity for valid pairs (driving $\sigma(x_{ij})$ towards 1). And it also decreases similarity for invalid pairs (driving $\sigma(x_{ij})$ towards 0). In contrast to the standard CLIP framework, which optimizes a global alignment between an image and a single text description, the point-wise loss enables the model to adjust each image–text pair individually, leading to a more discriminative and robust feature space. Every potential image–text pair is evaluated, allowing the model to learn subtle distinctions. By penalizing both false positives and false negatives at the individual pair level, the loss helps to create a more separable and robust embedding space. The derivation of its gradient, $\frac{\partial \mathcal{L}_P}{\partial x_{ij}} = \sigma(x_{ij}) - y_{ij}$, clearly shows that the optimization encourages high similarity for valid pairs and low similarity for invalid pairs. Point-wise Loss not only enhances the discriminability of the visual representations but also supports the multi-label learning methods.

## A.3 SMOOTH KL DIVERGENCE LOSS

*Overview and Motivation* The Smooth KL Divergence Loss is introduced to enforce consistency across predictions obtained at different granularities. In scenarios where multiple predicted logits $\{\mathbf{z}_i\}_{i=1}^m$ are available (each corresponding to a different granularity or viewpoint), we wish to align their probability distributions so that they all "agree" on the prediction. This is achieved by first converting the logits into probability distributions via the Softmax function:

$$P_i = \text{Softmax}(\mathbf{z}_i) \tag{36}$$

and then encouraging each $P_i$ to be close to the average (mean) distribution:

$$M = \frac{1}{m} \sum_{i=1}^m P_i \tag{37}$$

The overall loss is given by

$$\mathcal{L}_{\text{sKL}} = \sum_{i=1}^m D_{\text{KL}}(P_i \| M) \tag{38}$$

where for each $i$ the KL divergence is defined as

$$D_{\text{KL}}(P_i \| M) = \sum_j P_i^{(j)} \log \frac{P_i^{(j)}}{M^{(j)}} \tag{39}$$

*Properties of KL Divergence* There are two key properties of the KL divergence:

1. Non-negativity (Gibbs' Inequality):

$$D_{\text{KL}}(P \| Q) \geq 0 \tag{40}$$

for any two probability distributions $P$ and $Q$.

2. Zero if and Only if Equality:

$$D_{\text{KL}}(P \| Q) = 0 \quad \text{if and only if} \quad P = Q \tag{41}$$

These properties imply that the divergence is minimized (and equals zero) when the two distributions are identical.

*Detailed Derivation and Proof*

**(a) Expressing the Loss**

Given the predicted distributions $\{P_i\}_{i=1}^m$ and the mean distribution $M$, the loss is

$$\mathcal{L}_{\text{sKL}} = \sum_{i=1}^m \underbrace{\left[ \sum_j P_i^{(j)} \log \frac{P_i^{(j)}}{M^{(j)}} \right]}_{D_{\text{KL}}(P_i \| M)} \tag{42}$$

Because each $D_{\text{KL}}(P_i \| M) \geq 0$, it follows that

$$\mathcal{L}_{\text{sKL}} \geq 0 \tag{43}$$

**(b) Conditions for Zero Loss**

The loss for a single term, $D_{\text{KL}}(P_i \| M)$, equals zero if and only if

$$P_i^{(j)} = M^{(j)} \quad \text{for every category } j \tag{44}$$

Since this must hold for every $i$, we have:

$$P_1 = P_2 = \cdots = P_m = M \tag{45}$$

This is the necessary and sufficient condition for minimizing the loss:

$$\mathcal{L}_{\text{sKL}} = 0 \quad \Longleftrightarrow \quad P_1 = P_2 = \cdots = P_m \tag{46}$$

Thus, by minimizing $\mathcal{L}_{\text{sKL}}$, the model is encouraged to produce consistent predictions across different granularities. So the mean distribution is defined as

$$M = \frac{1}{m} \sum_{i=1}^m P_i \tag{47}$$

If all $P_i$ are equal, then it is trivial to see that

$$M = P_i \quad \forall i \tag{48}$$

Therefore, the minimization process pushes each individual $P_i$ toward the common distribution $M$, ensuring consistency across different predictions.

**(c) Gradient Considerations**

While the explicit gradient derivation is more involved due to the dependency of $M$ on every $P_i$, we can outline the intuition of individual KL divergence and the intuition of coupled optimization.

1. Individual KL Divergence: For each $i$, consider the derivative of

$$D_{\text{KL}}(P_i \| M) = \sum_j P_i^{(j)} \log \frac{P_i^{(j)}}{M^{(j)}} \tag{49}$$

with respect to $P_i^{(j)}$. If we ignore the dependence of $M$ on $P_i$ (as an approximation), the derivative is

$$\frac{\partial D_{\text{KL}}(P_i \| M)}{\partial P_i^{(j)}} \approx \log \frac{P_i^{(j)}}{M^{(j)}} + 1 \tag{50}$$

The condition for a minimum (when this derivative is zero for all $j$) is then

$$\log \frac{P_i^{(j)}}{M^{(j)}} + 1 = 0 \quad \Longrightarrow \quad \frac{P_i^{(j)}}{M^{(j)}} = e^{-1} \tag{51}$$

which alone does not yield $P_i = M$; however, when accounting for the normalization constraint $\sum_j P_i^{(j)} = 1$ and the simultaneous optimization across all $P_i$, the equilibrium is reached only when $P_i^{(j)} = M^{(j)}$ for every $i$ and $j$.

2. Coupled Optimization: Since $M$ is the average of all $P_i$, any deviation in one $P_i$ from the others will increase its corresponding KL divergence. Thus, the overall optimization drives all predictions to align with each other.

*Interpretation and Discussion* By minimizing $\mathcal{L}_{\text{sKL}}$, we force the different predictions (from various granularities) to become consistent. Every $P_i$ is toward the common mean $M$, ensuring that the predictions from different parts of the model or from different feature granularities agree with each other. This consistency also contributes to a more stable and robust feature space, as the model learns to reconcile variations in prediction across granularities. The Smooth KL Divergence Loss $\mathcal{L}_{\text{sKL}} = \sum_{i=1}^{m} D_{\text{KL}}(P_i \| M)$ is fundamentally designed to enforce that all predicted probability distributions $P_i$ (across different granularities) become identical by driving them toward the mean distribution $M$.

# B  DATASET DETAILS

## B.1  PRETRAIN DATASETS

**MGLL-Fundus:** We develop MGLL-Fundus, a comprehensive multi-granularity fundus image-text dataset comprising 246,389 image-text pairs. This dataset integrates fundus images from 49 public datasets, encompassing more than 50 disease categories. The data distribution of the MGLL-Fundus dataset is presented in Table 9. The textual descriptions in MGLL-Fundus are structured across two distinct granularity levels: disease category and clinical explanation. The disease-level granularity includes normal/abnormal classification and specific disease categorization, while the clinical explanation granularity provides detailed textual descriptions derived from dataset label explanations and EyeWiki EyeWiki (2024). Some multi-granularity textual description examples from the dataset are illustrated in Table 10.

Table 9: The data distribution of MGLL-Fundus dataset.

| Dataset | Num | Dataset | Num | Dataset | Num |
|---|---|---|---|---|---|
| HRF Budai et al. (2013) | 45 | INSPIRE-AVR Niemeijer et al. (2011) | 40 | IOSTAR Zhang et al. (2016) | 30 |
| RITE Hu et al. (2013) | 40 | G1020 Bajwa et al. (2020) | 1020 | GAMMA Wu et al. (2023) | 100 |
| ORIGA Zhang et al. (2010) | 650 | REFUGE Orlando et al. (2020) | 1200 | ODIR Larxel (2021) | 7000 |
| PALM Fang et al. (2024) | 1200 | RFMiD Pachade et al. (2021) | 3200 | RFMiDv2 Panchal et al. (2023) | 860 |
| APTOS Karthik et al. (2019) | 3662 | DeepDRiD Liu et al. (2022a) | 1600 | EyePACS Dugas et al. (2015) | 35126 |
| IDRID Porwal et al. (2018) | 516 | ADAM Fang et al. (2022) | 1200 | ACRIMA Diaz-Pinto et al. (2019) | 705 |
| MESSIDOR-2 Abràmoff et al. (2013) | 1748 | JSIEC Cen et al. (2021) | 1000 | AIROGS De Vente et al. (2023) | 101442 |
| LAG Li et al. (2019a) | 4854 | PARAGUAY Benítez et al. (2021) | 757 | PAPILA Kovalyk et al. (2022) | 488 |
| BiDR Darabi (2024) | 2838 | FIVES Jin et al. (2022) | 800 | FUND Hassan et al. (2022) | 179 |
| E-ophta Decenciere et al. (2013) | 463 | BRSET Nakayama et al. (2023) | 16266 | MuReD Rodríguez et al. (2022) | 2208 |
| OIA-DDR Li et al. (2019b) | 12522 | SUSTech-SYSU Lin et al. (2020) | 1219 | Cataract 202 (2020) | 601 |
| DGOCF Takahashi et al. (2017) | 9939 | BoVW Pires et al. (2014) | 2013 | HarvardGlaucoma Kim (2018a) | 1544 |
| RIM-ONE Batista et al. (2020) | 485 | CHAKSU Kumar et al. (2023) | 1345 | DiaRetDB Kauppi et al. (2007) | 89 |
| LSD Wei et al. (2019) | 175 | GNG Nandi (2022) | 400 | AOD 202 (2021) | 14813 |
| DHRF 202 (2022) | 2757 | VietAI vie (2020) | 3435 | ToxoFundus Alam et al. (2024) | 411 |
| Papilledema Kim (2018b) | 1369 | BEH Islam et al. (2021) | 634 | ROD Binu (2023) | 281 |
| ROI Adal et al. (2015) | 1120 | | | | |
| Summary | | 246,389 images | | | |

**MGLL-Xray:** To enhance data compatibility, we assembled 190,882 X-ray images from the MIDRC repository MIDRC (2024). We transformed these images from DICOM to PNG format while preserving essential metadata. The extracted multi-granularity textual information encompasses three levels: modality, study description, and series description. The modality category distinguishes between CR (Computed Radiography), characterized by relatively lower resolution and signal-to-noise ratio (SNR), and DX (Digital Radiography), which employs flat-panel detectors to achieve superior image quality. Study Description provides examination-level context (e.g., "Chest X-ray"), while Series Description specifies imaging protocols such as "PA View" (posteroanterior) or "Lateral View." These hierarchical textual elements constitute the textual component of our MGLL-Xray dataset. Some multi-granularity textual description examples from this dataset are presented in Table 11.

**MIMIC-CXR:** To further evaluate the generalization capability of our approach, we conducted supplementary experiments using multi-granular labels from MIMIC-CXR dataset Johnson et al. (2019), with performance assessed on the ChestX-ray14 benchmark Wang et al. (2017a). The MIMIC-CXR dataset represents one of the largest publicly available collections of chest radiographs, comprising 377,110 images associated with 227,835 imaging studies. This dataset encompasses 14 common thoracic pathologies, including atelectasis, cardiomegaly, consolidation, edema,

Table 10: The multi-granularity textual description examples of MGLL-Fundus dataset.

| Disease Category | Clinical Explanation |
|---|---|
| Abnormal, Mild Non-Proliferative Diabetic Retinopathy | Only microaneurysms observed |
| Abnormal, Moderate Non-Proliferative Diabetic Retinopathy | Retinal hemorrhages or hard exudates observed |
| Abnormal, Severe Non-Proliferative Diabetic Retinopathy | Many intraretinal hemorrhages or definite venous beading observed |
| Abnormal, Proliferative Diabetic Retinopathy | Neovascularization or vitreous/preretinal hemorrhage |
| Abnormal, Cataract | Opacification of crystalline lens observed |
| Abnormal, Myopia | Leopard fundus observed |
| Abnormal, Media Haze | Opacity of media observed |
| Abnormal, Branch Retinal Vein Occlusion | Occlusion of the central retinal vein |
| Abnormal, Tessellation | The choroidal vessels are visible due to the reduced density of the pigments |
| Abnormal, Laser Scars | Circular or irregular shaped scars on the retinal surface observed |
| Abnormal, Central Serous Retinopathy | Fluid accumulation under the retina observed |
| Abnormal, Optic Disk Cupping | The thinning of neuroretinal rim such that optic disc appears excavated |
| Abnormal, Central Retinal Vein Occlusion | Occlusion of the central retinal vein and the presence of flame-shaped hemorrhages |
| Abnormal, Tortuous Vessels | Marked tortuosity of the retinal blood vessels |
| Abnormal, Asteroid Hyalosis | Numerous astroid bodies are dispersed in vitreous |
| Abnormal, Optic Disc Pallor | Pale yellow discoloration of the optic disc |
| Abnormal, Optic Disc Edema | Swelling of the optic disc |
| Abnormal, Optociliary Shunt | Presence of prepapillary vascular loops or optociliary shunt vessels |
| Abnormal, Anterior Ischemic Optic Neuropathy | Optic disc swelling and pallor |
| Abnormal, Parafoveal Telangiectasia | Yellow, lipid-rich exudation or parafoveal graying or tortuous blood vessels |
| Abnormal, Retinal Traction | Presence of traction and retinal traction detachment |
| Abnormal, Retinitis | Presence of vitreous inflammation or intraretinal hemorrhage |
| Abnormal, Chorioretinitis | The hard exudates observed |
| Abnormal, Macular Hole | A small retinal break located in the center of the fovea observed |
| Abnormal, Retinitis Pigmentosa | The presence of bone-spicule deposits and arterial narrowing |
| Abnormal, Cotton Wool Spots | The presence of soft exudates |
| Abnormal, Coloboma | The missing of portion of tissue in both the choroid and retina |
| Abnormal, Preretinal Hemorrhage | Boat-shaped hemorrhage which obscures the underlying retina |
| Abnormal, Myelinated Nerve Fibers | Gray-white opaque lesions with feathery edges observed |
| Abnormal, Hemorrhagic Retinopathy | The presence of flame-shaped hemorrhages |
| Abnormal, Central Retinal Artery Occlusion | The presence of pale, whitening, and retinal swelling |
| Abnormal, Tilted Disk | The tilting presence of the oval optic disc |
| Abnormal, Cystoid Macular Edema | The presence of multiple cystoid areas in the macula and causes retinal edema |
| Abnormal, Post-traumatic Choroidal Rupture | The breaks in the choroid, Bruch's membrane, and RPE |
| Abnormal, Choroidal Folds | The presence of folds in the choroid |
| Abnormal, Vitreous Hemorrhage | The presence of extravasated blood in one of the spaces created around the vitreous body |
| Abnormal, Macroaneurysm | Fusiform or round dilation of the retinal arterioles which occur in the temporal retina observed |
| Abnormal, Vasculitis | The presence of inflammation of retinal blood vessels |
| Normal, Healthy | Clear optic disk boundaries, Normal fundus color, No apparent retinopathy |

Table 11: The multi-granularity textual description examples of MGLL-Xray dataset.

| Modality | Study Description | Series Description |
|---|---|---|
| CR | CHEST PORT 1 VIEW (RAD)-CS | AP(shutter) |
| CR | XR CHEST AP PORTABLE | AP |
| CR | XR RIBS RIGHT WITH CHEST AP OR PA - SINGLE VIEW | PA Ribs LOWER |
| CR | XR PORT CHEST 1V | CXR AP GRID |
| CR | XR CHEST 2 VIEWS | W Chest Lat. |
| CR | XR CHEST AP PA LATERAL 2 VW | Lateral |
| CR | XR PORT CHEST 1V | ClearRead Bone Suppression |
| CR | XRAY CHEST ONE VIEW | XRAY CHEST FRONTAL AND LATERAL VIEWS |
| CR | XR RIGHT HIP 2+ VIEWS ORTHOPEDICS PRE OPERATIVE | X HIP X-Table Lat |
| CR | XR CHEST PA AND LATERAL | X Chest a.p. |
| DX | XR CHEST 2 VIEWS, FRONTAL AND LATERAL | PA |
| DX | XR WRIST LEFT (ROUTINE: AP,LAT,OBL) | XR WRIST LEFT (ROUTINE: AP,LAT,OBL) |
| DX | XR CHEST 2 VIEWS PA AND LATERAL | Chest |
| DX | XR THORACOLUMBAR SPINE STANDING 2 OR 3 VIEWS | Thoraco Lumbar |
| DX | XR THORACIC SPINE AP AND LATERAL | Thoracic-spine |
| DX | XR SCOLIOSIS STUDY 2 OR 3 VIEWS (NEURO INTERPRETATION) | DR Long Spine |
| DX | XR RIGHT RIBS 2 VIEWS UNILATERAL | Rib |
| DX | XR RIGHT SHOULDER 1 VIEW | AP Ext Rot(shutter) |
| DX | XR RIGHT KNEE 4+ VIEWS (NON-TRAUMA, PAIN/ARTHRITIS) | Patella |
| DX | XR RIGHT HIP 2-3 VIEWS (UNILATERAL) WITH PELVIS WHEN PERFORMED | Hip-joint |
| DX | XR CHEST 1 VW, FRONTAL | PA CHEST LANDSCAPE |

effusion, emphysema, fibrosis, hernia, infiltration, mass, nodule, pleural thickening, pneumonia, and pneumothorax, along with a "No Finding" category for normal cases. For our multi-granularity framework, we leveraged two distinct levels of textual information available in MIMIC-CXR: the structured disease labels (coarse granularity) and the detailed radiology reports (fine granularity). The disease labels provide categorical classification, while the reports offer comprehensive clinical interpretations with anatomical specificity, disease progression details, and differential diagnoses. This hierarchical representation allows our model to simultaneously process high-level disease categorization and nuanced clinical descriptions. Some examples of these multi-granularity textual descriptions are presented in Table 12.

Table 12: The multi-granularity textual description examples of MIMIC-CXR dataset.

| Disease Labels | Radiology Reports |
|---|---|
| Pleural Effusion | New tracheostomy is midline. The approximate diameter of the tube, 11 mm, compares to the diameter of the trachea, 27 mm. This sizing should be evaluated clinically. Pneumomediastinum outlines the tracheal wall and extends into deep subcutaneous emphysema in the neck, presumably a function of tracheostomy. Followup advised. There is no pneumothorax or pleural effusion. Lungs are clear. Heart size is normal. |
| Edema | The small right apical pneumothorax is stable and unchanged. The right chest tube is in stable position. Unchanged parenchymal opacity at the left lung base. Unchanged size of the cardiac silhouette and stable position of the right internal jugular vein catheter. |
| Atelectasis | Monitoring and support devices are in stable position. Stable left retrocardiac atelectasis and right basal parenchymal opacity. No pulmonary edema. No larger pleural effusions. No pneumothorax. |
| Cardiomegaly | Support lines and tubes are unchanged in position. The left-sided pleural effusion continues to decrease in size. There is improved aeration at the left base. Partially layering right-sided pleural effusion is again seen. There is a new small left-sided apical pneumothorax. |
| Lung Opacity | Slight worsening of cardiomegaly and mild-to-moderate pulmonary edema, accompanied by increasing moderate left pleural effusion and persistent small right pleural effusion. Indwelling support and monitoring devices are unchanged in position, including a proximally located left PICC, terminating at the junction of the left axillary and subclavian veins. |
| Lung Lesion | Single portable upright AP image of the chest. There are low lung volumes. There is an interval increase in the alveolar opacities bilaterally, consistent with moderate to severe new onset pulmonary edema. The cardiomediastinal silhouette appears to be somewhat enlarged from prior exam, particularly in the right mediastinum. There is no large pleural effusion or pneumothorax. A pacer is seen overlying the left anterior chest with intact leads in appropriate position. |
| Pneumonia | Portable AP radiograph of the chest was reviewed with no prior studies available for comparison. Heart size is top normal. Mediastinum is grossly unremarkable. Lungs are essentially clear except for right basal opacity which unclear if represents a true lesion or summation of shadows. Repeated radiograph preferably with full inspiration is required. If finding is persistent, assessment with chest CT would be necessary. |
| No Finding | Tracheostomy tube is in satisfactory position with the tip 4.5 cm above the carina. The right internal jugular central line and nasogastric tube are unchanged in position. The heart remains stably enlarged. Lung volumes are markedly reduced and there is a small layering left effusion with persistent retrocardiac consolidation likely reflecting partial lower lobe atelectasis. No pulmonary edema. No obvious pneumothorax. |

## B.2 DOWNSTREAM DATASETS

We evaluate our proposed MGLL alongside several baseline methods on multiple downstream datasets. The details of these datasets are presented in Table 13. The multiple-choice evaluation benchmark details are in Table 14.

***Fundus Imaging Datasets:***

**FIVES** Jin et al. (2022): A collection of 800 retinal images categorized into four diagnostic classes: normal, age-related macular degeneration, diabetic retinopathy, and glaucoma.

**IDRiD** Porwal et al. (2018): Contains 516 retinal images with annotated severity grades for diabetic retinopathy (DR) and diabetic macular edema (DME).

**OIA-DDR** Li et al. (2019b): Comprises 12,523 fundus images labeled with diabetic retinopathy severity classifications.

**ADAM** Fang et al. (2022): A dataset of 1,200 fundus images specifically designed for age-related macular degeneration detection.

**PALM** Fang et al. (2024): Consists of 1,200 fundus images annotated for pathological myopia diagnosis.

**REFUGE** Orlando et al. (2020): Includes 1,200 retinal images with binary classification for glaucomatous and non-glaucomatous.

**RIM-ONE** Batista et al. (2020): A retinography collection of 485 images developed for glaucoma evaluation.

**RFMiD** Pachade et al. (2021): Encompasses 3,200 retinal images with multi-label annotations across 45 categories. Our evaluation focuses on 12 labels where positive cases exceed 2% prevalence.

***Radiographic Imaging Datasets:***

**MIDRC-XR** MIDRC (2024): A dataset contains 111,816 X-ray images across 14 LOINC-coded categories (including XR Chest AP views).

Table 13: The details of downstream datasets.

| Name | Numer (Train : Val : Test) | Label Categories |
|---|---|---|
| FIVES Jin et al. (2022) | 480 : 120 : 200 | Normal, Age-related Macular Degeneration, Diabetic retinopathy, and Glaucoma |
| IDRiD Porwal et al. (2018) | 319 : 94 : 103 | Severity levels of Diabetic Retinopathy (no apparent, mild non-proliferative, moderate non-proliferate, severe non-proliferate, proliferative) and Diabetic Macular Edema (no apparent, mild, moderate, severe) |
| OIA-DDR Li et al. (2019b) | 6260 : 2503 : 3759 | Severity levels of Diabetic Retinopathy (no apparent, mild non-proliferative, moderate non-proliferate, severe non-proliferate, proliferative) |
| ADAM Fang et al. (2022) | 400 : 400 : 400 | Age-related Macular Degeneration and no Age-relatedd Macular Degeneration |
| PALM Fang et al. (2024) | 400 : 400 : 400 | Pathological myopia and Healthy |
| REFUGE Orlando et al. (2020) | 400 : 400 : 400 | Glaucoma and Healthy |
| RIM-ONE Batista et al. (2020) | 270 : 69 : 146 | Glaucoma and Healthy |
| RFMiD Pachade et al. (2021) | 1920 : 640 : 640 | Diabetic Retinopathy, Age-related Macular Degeneration, Media haze, drusens, Myopia, Branch Retinal Vein Occlusion, Tessellation, Laser scars, Optic disc cupping, Optic disc pallor, Optic disc edema, and Retinitis |
| MIDRC-XR MIDRC (2024) | 89453 : 11182 : 11181 | XR Chest AP, XR Chest 2 Views, XR Unspecified body region Views, XR Chest Single view, XR Chest PA and Lateral, XR Chest Views, XR Chest AP and Lateral, XR Chest and Abdomen Single view, XR Ribs Views, XR Abdomen AP, XR Abdomen Single view, XR Chest View and Abdomen Supine and Upright, XR Abdomen Supine and Upright, XR Ribs Views and Chest PA |
| MIDRC-XR-Portable MIDRC (2024) | 63253 : 7906 : 7907 | Portable XR Chest AP single view, Portable XR Chest Views AP, Portable XR Abdomen AP, Portable XR Chest Views, Portable XR Chest Views W inspiration and expiration, Portable XR Abdomen Supine and Upright |
| ChestX-ray14 Wang et al. (2017a) | 77872 : 8652 : 25596 | Atelectasis, Cardiomegaly, Effusion, Infiltration, Mass, Nodule, Pneumonia, Pneumothorax, Consolidation, Edema, Emphysema, Fibrosis, Pleural Thickening, Hernia |

**MIDRC-XR-Portable** MIDRC (2024): Focuses on portable radiography with 79,066 X-ray images across 6 LOINC-coded categories (including Portable XR Chest AP single views).

**ChestX-ray14** Wang et al. (2017a): A comprehensive medical imaging repository contains 112,120 frontal-view chest radiographs annotated with 14 labels. Labels are extracted from corresponding radiological reports.

***Evaluation Benchmark on MLLMs (Multiple-choice Benchmark):*** A comprehensive ophthalmic evaluation dataset comprising 2,233 images across 25 distinct diagnostic categories. The label distribution of the multiple-choice evaluation benchmark is in Table 14.

Table 14: The label distribution of the multiple-choice evaluation benchmark.

| Label | Num | Label | Num |
|---|---|---|---|
| Health | 890 | Other disease (Other) | 50 |
| Myopia | 226 | Tessellation | 12 |
| Retinitis | 9 | Chorioretinitis | 3 |
| Diabetic Retinopathy (DR) | 149 | Drusen | 30 |
| Media Haze (MH) | 31 | Central Serous Retinopathy (CSR) | 7 |
| Cataract | 20 | Arteriosclerotic Retinopathy (AR) | 2 |
| Optic Disk Cupping (ODC) | 32 | Optic Disc Edema (ODE) | 11 |
| Optic Disc Pallor (ODP) | 2 | Hypertensive Retinopathy (HR) | 3 |
| Branch Retinal Vein Occlusion (BRVO) | 16 | Central Retinal Vein Occlusion (CRVO) | 11 |
| Age-related Macular Degeneration (AMD) | 171 | No Age-related Macular Degeneration (No AMD) | 311 |
| Diabetic Macular Edema (DME) | 58 | No Diabetic Macular Edema (No DME) | 11 |
| Glaucoma | 162 | No Glaucoma | 12 |
| Choroidal Neovascularization (CN) | 4 | **Summary** | **2233 images** |

## C SETUP DETAILS

### C.1 EVALUATION METRICS

In our quantitative evaluation, we employ Area Under the Receiver Operating Characteristic Curve (AUC), mean Average Precision (mAP), and Accuracy (ACC) as assessment metrics. Among these, AUC serves as our primary evaluation metric, as it reflects overall model performance. mAP is particularly useful for handling long-tailed label distributions. To better assess performance on the imbalanced multi-label dataset such as RFMiD Pachade et al. (2021), we report the category-wise average accuracy instead of overall accuracy. As for the accuracy on the multiple-choice benchmark, we implement a four-option forced-choice classification approach utilizing the MGLL-Fundus dataset. For each fundus image presented, the model must select the most probable diagnostic classification from among four distinct disease labels. These options comprise the correct diagnostic label along with three additional labels randomly sampled from the complete disease names available in the dataset. The randomized inclusion of incorrect options helps evaluate model performance in distinguishing the correct diagnosis from plausible alternatives, which also enables quantitative assessment of diagnostic accuracy.

### C.2 IMPLEMENTATION DETAILS

We adopt ViT-L/14 Dosovitskiy et al. (2020) as the image encoder and BiomedicalBERT Alsentzer et al. (2019) as the text encoder. All images are resized to $224 \times 224$, and data preprocessing includes random flipping (probability = 0.5) and color jittering (factor = 0.1). We set the batch size to 32, the feature vector dimension to 768, and the temperature coefficient to 0.07. Optimization is performed using AdamW Loshchilov & Hutter (2017) with a learning rate of 1e-4, weight decay of 0.0001, and hyperparameters $\beta_1 = 0.9$, $\beta_2 = 0.98$, and $\epsilon = 1e-6$. All experiments are conducted with the NVIDIA RTX A6000 GPU.

# D    MORE DETAILED EXPERIMENTAL RESULTS

## D.1    DETAILED RESULTS ON RETINAL FUNDUS DATASETS

We present the complete experimental results for performance comparison with eight baselines (Radford et al. (2021); Yao et al. (2023); Zhou et al. (2023b); Silva-Rodriguez et al. (2025); Wu et al. (2024); Du et al. (2024); Khattak et al. (2024); Qiu et al. (2024)) on downstream retinal fundus datasets in Table 15 to Table 23. The experimental results demonstrate that the MGLL consistently outperforms existing approaches across nine retinal fundus datasets in both linear probing and full fine-tuning evaluation settings. Notably, MGLL demonstrates particularly strong gains in the linear probing setting, where it demonstrates substantial improvements over second-best methods (e.g., achieving 90.02% AUC in ADAM compared to UniMed-CLIP's 79.33%, and 92.42% AUC in REFUGE versus RET-CLIP's 84.59%). To further analyze these results, we observe consistent performance improvements across multiple evaluation metrics. For instance, in the FIVES dataset, MGLL achieved 89.73% AUC, 52.00% ACC, and 75.32% mAP in linear probing, significantly outperforming RETFound (88.09% AUC, 49.00% ACC, 72.55% mAP). When fully fine-tuned on this dataset, MGLL maintained its advantage with 94.98% AUC, 72.00% ACC, and 86.34% mAP.

The robust performance of MGLL can be attributed to its multi-granularity learning approach, which effectively captures both local and global features in retinal fundus images. This architectural advantage enables MGLL to identify subtle pathological patterns that may be overlooked by conventional methods. For example, in the REFUGE dataset (glaucoma detection), MGLL achieved a 7.83% improvement in AUC over the second-best method in linear probing setting.

Additionally, the exceptional performance on the PALM dataset (99.66% AUC, 96.00% ACC, and 99.72% mAP in linear probing) demonstrates MGLL's capacity to achieve near-perfect diagnostic accuracy in certain retinal conditions. When compared to previous state-of-the-art methods such as VisionFM (97.12% AUC) and RET-CLIP (95.25% AUC), MGLL offers clinically significant improvements in detection reliability. This superior performance indicates excellent feature representation quality and transferability of our pretrained MGLL, enabling effective adaptation to diverse diagnostic tasks with fine-tuning.

Table 15: The performance evaluation on FIVES. **Bold** indicates best performance and underline shows second-best.

| Method | Linear Probe (%) | | | Fully Fine-tune (%) | | |
|---|---|---|---|---|---|---|
| | AUC | ACC | mAP | AUC | ACC | mAP |
| CLIP (ICML-21) | 81.25 | 37.00 | 64.21 | 88.96 | 64.00 | 76.22 |
| KgCoOp (CVPR-23) | 81.63 | 39.50 | 64.72 | 89.16 | 64.50 | 76.53 |
| RETFound (Nature-23) | 88.09 | 49.00 | 72.55 | 92.83 | 69.50 | 81.36 |
| FLAIR (MedIA-25) | 84.24 | 43.50 | 67.17 | 89.85 | 66.00 | 77.14 |
| KeepFIT (MICCAI-24) | 82.31 | 41.00 | 64.98 | 90.62 | 67.00 | 77.96 |
| RET-CLIP (MICCAI-24) | 86.74 | 47.50 | 69.35 | 92.04 | 68.50 | 79.89 |
| UniMed-CLIP (arXiv-24) | 82.75 | 41.50 | 65.46 | 91.59 | 68.00 | 79.23 |
| VisionFM (NEJM AI-24) | 85.98 | 45.00 | 68.73 | 93.11 | 70.50 | 82.76 |
| MGLL | **89.73** | **52.00** | **75.32** | **94.98** | **72.00** | **86.34** |

Table 16: The performance evaluation on IDRiD (DR). **Bold** indicates best performance and underline shows second-best.

| Method | Linear Probe (%) | | | Fully Fine-tune (%) | | |
|---|---|---|---|---|---|---|
| | AUC | ACC | mAP | AUC | ACC | mAP |
| CLIP (ICML-21) | 70.83 | 35.92 | 34.44 | 76.74 | 44.66 | 44.01 |
| KgCoOp (CVPR-23) | 72.34 | 40.78 | 36.29 | 76.81 | 46.60 | 44.86 |
| RETFound (Nature-23) | 73.51 | 43.69 | 37.48 | 78.14 | 53.40 | 47.05 |
| FLAIR (MedIA-25) | 74.62 | 45.63 | 39.16 | 78.82 | 55.34 | 48.54 |
| KeepFIT (MICCAI-24) | 71.81 | 37.86 | 35.37 | 77.13 | 48.54 | 45.32 |
| RET-CLIP (MICCAI-24) | 78.18 | 52.43 | 47.42 | 79.32 | 56.31 | 49.42 |
| UniMed-CLIP (arXiv-24) | 77.39 | 49.51 | 45.75 | 80.72 | 58.25 | 52.84 |
| VisionFM (NEJM AI-24) | 77.52 | 51.46 | 46.22 | 79.95 | 57.28 | 51.38 |
| MGLL | **80.28** | **58.25** | **51.19** | **82.57** | **60.19** | **54.30** |

Table 17: The performance evaluation on IDRiD (DME). **Bold** indicates best performance and underline shows second-best.

| Method | Linear Probe (%) | | | Fully Fine-tune (%) | | |
|---|---|---|---|---|---|---|
| | AUC | ACC | mAP | AUC | ACC | mAP |
| CLIP (ICML-21) | 71.81 | 77.67 | 56.67 | 73.34 | 76.70 | 57.87 |
| KgCoOp (CVPR-23) | 60.33 | 65.05 | 48.86 | 62.78 | 66.02 | 50.11 |
| RETFound (Nature-23) | 65.92 | 69.90 | 52.05 | 69.26 | 72.82 | 55.39 |
| FLAIR (MedIA-25) | 62.85 | 66.99 | 50.31 | 64.52 | 68.93 | 51.83 |
| KeepFIT (MICCAI-24) | 68.71 | 70.87 | 54.76 | 69.03 | 71.84 | 54.97 |
| RET-CLIP (MICCAI-24) | 64.13 | 67.96 | 51.28 | 70.14 | 73.79 | 55.75 |
| UniMed-CLIP (arXiv-24) | 70.59 | 74.76 | 56.04 | 71.81 | 75.73 | 56.85 |
| VisionFM (NEJM AI-24) | 73.53 | 78.64 | 59.98 | 77.95 | 79.61 | 62.23 |
| MGLL | **78.41** | **79.61** | **64.72** | **86.17** | **80.58** | **67.80** |

Table 18: The performance evaluation on OIA-DDR. **Bold** indicates best performance and underline shows second-best.

| Method | Linear Probe (%) | | | Fully Fine-tune (%) | | |
|---|---|---|---|---|---|---|
| | AUC | ACC | mAP | AUC | ACC | mAP |
| CLIP (ICML-21) | 73.30 | 55.41 | 39.21 | 85.29 | 71.75 | 49.91 |
| KgCoOp (CVPR-23) | 72.09 | 53.68 | 37.87 | 80.39 | 65.47 | 45.92 |
| RETFound (Nature-23) | 80.77 | 61.13 | 46.94 | 85.96 | 72.12 | 52.21 |
| FLAIR (MedIA-25) | 82.48 | 63.36 | 48.09 | 85.54 | 70.63 | 50.27 |
| KeepFIT (MICCAI-24) | 72.68 | 54.46 | 38.72 | 79.42 | 64.03 | 44.53 |
| RET-CLIP (MICCAI-24) | 77.43 | 58.18 | 43.31 | 83.68 | 68.82 | 48.05 |
| UniMed-CLIP (arXiv-24) | 74.67 | 56.19 | 40.18 | 81.23 | 66.69 | 46.88 |
| VisionFM (NEJM AI-24) | 78.85 | 59.35 | 44.27 | 84.25 | 69.46 | 48.77 |
| MGLL | **86.28** | **72.09** | **50.92** | **88.85** | **73.13** | **56.67** |

Table 19: The performance evaluation on ADAM. **Bold** indicates best performance and underline shows second-best.

| Method | Linear Probe (%) | | | Fully Fine-tune (%) | | |
|---|---|---|---|---|---|---|
| | AUC | ACC | mAP | AUC | ACC | mAP |
| CLIP (ICML-21) | 52.41 | 76.50 | 25.05 | 86.70 | 83.00 | 65.48 |
| KgCoOp (CVPR-23) | 52.13 | 74.75 | 24.36 | 84.42 | 82.75 | 60.54 |
| RETFound (Nature-23) | 59.34 | 78.25 | 33.53 | 88.02 | 83.50 | 65.86 |
| FLAIR (MedIA-25) | 63.82 | 79.50 | 39.27 | 90.16 | 84.75 | 66.91 |
| KeepFIT (MICCAI-24) | 56.97 | 77.75 | 29.88 | 74.88 | 82.00 | 52.26 |
| RET-CLIP (MICCAI-24) | 77.48 | 82.25 | 53.76 | 93.27 | 86.25 | 79.92 |
| UniMed-CLIP (arXiv-24) | 79.33 | 82.50 | 55.63 | 90.64 | 85.25 | 69.22 |
| VisionFM (NEJM AI-24) | 73.56 | 81.75 | 51.42 | 92.43 | 85.50 | 75.89 |
| MGLL | **90.02** | **85.00** | **62.40** | **96.30** | **90.00** | **90.08** |

Table 20: The performance evaluation on PALM. **Bold** indicates best performance and underline shows second-best.

| Method | Linear Probe (%) | | | Fully Fine-tune (%) | | |
|---|---|---|---|---|---|---|
| | AUC | ACC | mAP | AUC | ACC | mAP |
| CLIP (ICML-21) | 93.94 | 91.75 | 94.62 | 99.51 | **96.00** | 99.58 |
| KgCoOp (CVPR-23) | 87.94 | 88.25 | 88.11 | 94.21 | 92.00 | 94.81 |
| RETFound (Nature-23) | 92.02 | 90.50 | 92.87 | 95.75 | 93.00 | 96.39 |
| FLAIR (MedIA-25) | 89.42 | 89.25 | 90.03 | 94.88 | 92.25 | 95.24 |
| KeepFIT (MICCAI-24) | 88.21 | 88.50 | 88.46 | 93.34 | 91.50 | 94.18 |
| RET-CLIP (MICCAI-24) | 95.25 | 92.50 | 95.89 | 98.67 | 95.25 | 98.42 |
| UniMed-CLIP (arXiv-24) | 90.31 | 89.75 | 91.12 | 96.82 | 93.75 | 97.02 |
| VisionFM (NEJM AI-24) | 97.12 | 94.25 | 97.45 | 97.73 | 94.50 | 97.84 |
| MGLL | **99.66** | **96.00** | **99.72** | **99.72** | 95.75 | **99.76** |

Table 21: The performance evaluation on REFUGE. **Bold** indicates best performance and underline shows second-best.

| Method | Linear Probe (%) | | | Fully Fine-tune (%) | | |
|---|---|---|---|---|---|---|
| | AUC | ACC | mAP | AUC | ACC | mAP |
| CLIP (ICML-21) | 65.33 | 88.75 | 15.68 | 86.96 | 93.00 | 58.37 |
| KgCoOp (CVPR-23) | 60.69 | 86.25 | 11.52 | 81.45 | 90.75 | 49.07 |
| RETFound (Nature-23) | 79.67 | 90.50 | 45.39 | 89.02 | 93.50 | 63.84 |
| FLAIR (MedIA-25) | 70.59 | 89.25 | 27.82 | 85.67 | 92.25 | 56.82 |
| KeepFIT (MICCAI-24) | 63.04 | 88.25 | 14.15 | 84.89 | 91.50 | 55.78 |
| RET-CLIP (MICCAI-24) | 84.59 | 91.25 | 55.06 | 90.46 | 93.75 | 70.45 |
| UniMed-CLIP (arXiv-24) | 61.25 | 87.50 | 12.87 | 83.11 | 91.00 | 52.84 |
| VisionFM (NEJM AI-24) | 73.21 | 89.50 | 33.42 | 86.13 | 92.50 | 57.68 |
| MGLL | **92.42** | **94.50** | **75.65** | **93.90** | **94.75** | **80.99** |

Table 22: The performance evaluation on RIM-ONE. **Bold** indicates best performance and underline shows second-best.

| Method | Linear Probe (%) | | | Fully Fine-tune (%) | | |
|---|---|---|---|---|---|---|
| | AUC | ACC | mAP | AUC | ACC | mAP |
| CLIP (ICML-21) | 65.96 | 66.44 | 54.11 | 88.38 | 82.88 | 84.00 |
| KgCoOp (CVPR-23) | 74.34 | 73.97 | 63.45 | 90.39 | 84.25 | 85.88 |
| RETFound (Nature-23) | 89.79 | 83.56 | 83.92 | 94.22 | 86.99 | 90.35 |
| FLAIR (MedIA-25) | 81.83 | 79.45 | 70.14 | 94.93 | 88.36 | 92.41 |
| KeepFIT (MICCAI-24) | 67.35 | 67.81 | 55.24 | 89.91 | 83.56 | 85.22 |
| RET-CLIP (MICCAI-24) | 84.42 | 82.19 | 79.85 | 92.58 | 86.30 | 89.19 |
| UniMed-CLIP (arXiv-24) | 69.87 | 70.55 | 58.43 | 83.37 | 81.51 | 78.14 |
| VisionFM (NEJM AI-24) | 72.97 | 72.60 | 61.86 | 91.27 | 84.93 | 87.21 |
| MGLL | **94.39** | **87.67** | **86.68** | **97.05** | **89.73** | **94.97** |

Table 23: The performance evaluation on RFMiD. **Bold** indicates best performance and underline shows second-best.

| Method | Linear Probe (%) | | | Fully Fine-tune (%) | | |
|---|---|---|---|---|---|---|
| | AUC | ACC | mAP | AUC | ACC | mAP |
| CLIP (ICML-21) | 44.66 | 92.53 | 7.28 | 65.10 | 92.86 | 17.31 |
| KgCoOp (CVPR-23) | 50.82 | 92.21 | 7.96 | 70.36 | 92.28 | 21.49 |
| RETFound (Nature-23) | 60.16 | 92.55 | 16.37 | 84.62 | 93.48 | 50.21 |
| FLAIR (MedIA-25) | 56.11 | 92.38 | 14.85 | 75.43 | 92.62 | 23.24 |
| KeepFIT (MICCAI-24) | 51.48 | 92.19 | 10.24 | 81.52 | 93.09 | 42.39 |
| RET-CLIP (MICCAI-24) | 58.94 | 92.46 | 16.02 | 86.12 | 93.92 | 51.27 |
| UniMed-CLIP (arXiv-24) | 53.44 | 92.25 | 13.68 | 72.59 | 92.57 | 22.54 |
| VisionFM (NEJM AI-24) | 63.38 | 92.59 | 17.84 | 82.78 | 93.17 | 48.92 |
| MGLL | **79.62** | **92.84** | **34.08** | **92.83** | **95.48** | **64.99** |

## D.2 ZERO-SHOT COMPARISONS ACROSS MEDICAL AND NATURAL DOMAINS

As summarized in Table 24, we evaluate the zero-shot classification performance of MGLL across three representative datasets to demonstrate its strong generalization ability across diverse modalities. On the COVIDx dataset Wang et al. (2020), MGLL achieves 39.0% accuracy, outperforming strong medical vision-language baselines such as CheXAgent Chen et al. (2024b) (34.3%) and MedVersa Zhou et al. (2024) (35.5%), as well as recent contrastive methods including FG-CLIP Xie et al. (2025), MGCA Wang et al. (2022a), RetiZero Wang et al. (2025), MAVL Phan et al. (2024), and Ark+ Ma et al. (2025). For anatomical recognition on the CT-based OrganAMNIST dataset Yang et al. (2023), MGLL again surpasses FG-CLIP Xie et al. (2025) with a significant margin (52.7% vs. 47.9%). Moreover, MGLL achieves the best performance on the natural image dataset CC3M Sharma et al. (2018) (evaluated on ImageNet), outperforming FG-CLIP Xie et al. (2025) with an accuracy of 23.5%. These results collectively highlight MGLL's flexibility and universal applicability across both medical and natural image domains in zero-shot settings.

Table 24: Comparisons of Zero-Shot classification on MGLL and others methods. * denotes using published pretrained model.

| Method | Pretrain Data | Downstream Data | ACC (%) |
|---|---|---|---|
| FG-CLIP (ICML-25) | CC3M | ImageNet | 21.4 |
| MGLL | CC3M | ImageNet | **23.5** |
| FG-CLIP (ICML-25) | PMC-OA | OrganAMNIST | 47.9 |
| MGLL | PMC-OA | OrganAMNIST | **52.7** |
| CheXAgent (Arxiv-24) | * | COVIDx | 34.3 |
| MedVersa (Arxiv-24) | * | COVIDx | 35.5 |
| FG-CLIP (ICML-25) | MIMIC-CXR | COVIDx | 36.3 |
| MGCA (NIPS-22) | MIMIC-CXR | COVIDx | 37.3 |
| RetiZero (Nat. Com-25) | MIMIC-CXR | COVIDx | 35.8 |
| MAVL (CVPR-24) | MIMIC-CXR | COVIDx | 37.0 |
| Ark+ (Nature-25) | MIMIC-CXR | COVIDx | 37.8 |
| MGLL | MIMIC-CXR | COVIDx | **39.0** |

## D.3 PERFORMANCE EVALUATION ON REGION SEGMENTATION

We have evaluated MGLL on region segmentation tasks, and the results are reported in Table 25. Medical image segmentation is inherently challenging due to the subtle differences between adjacent pixels of heterogeneous classes. We compared our MGLL with several excellent methods such as GLoRIAHuang et al. (2021), CLIPRadford et al. (2021), LAVTYang et al. (2022), UniLSegLiu et al. (2024), and STPNetShan et al. (2025). By incorporating multi-level semantic alignment, MGLL enhances the model's language-guided spatial understanding and achieves the best performance among compared methods.

Table 25: Comparisons of COVID-19 lesion segmentation between MGLL and others methods on COVID-Xray dataset Degerli et al. (2021).

| Method | Dataset | Dice (%) | IoU(%) |
|---|---|---|---|
| GLoRIA (ICCV-21) | COVID-Xray | 79.94 | 70.68 |
| CLIP (ICML-21) | COVID-Xray | 79.81 | 70.66 |
| LAVT (CVPR-22) | COVID-Xray | 79.28 | 69.89 |
| UniLSeg (CVPR-24) | COVID-Xray | 79.99 | 70.29 |
| STPNet (TIP-25) | COVID-Xray | 80.63 | 71.42 |
| MGLL | COVID-Xray | **81.69** | **73.06** |

### D.4 MORE DETAILED RESULTS OF MGLL IN MLLMS

To evaluate MGLL's impact on multimodal large language models' diagnostic capabilities, we conduct comprehensive multiple-choice evaluations across 25 distinct ophthalmological conditions as shown in Table 27 to Table 34. This expanded analysis provides more insights into MGLL's performance beyond the ten primary conditions highlighted in previous experiments.

Table 26: Brief summary of recent vision-language models.

| Model Name | Key Features and Description |
|---|---|
| InstructBLIPDai et al. (2023) | Combines vision and language with instruction tuning to enable versatile zero-shot performance across tasks. |
| Mini-GeminiLi et al. (2024) | A lightweight and efficient multimodal model designed for fast inference and strong performance. |
| Qwen-VLBai et al. (2023) | Supports multimodal reasoning with a strong focus on Chinese vision-language understanding. |
| InternVLChen et al. (2024a) | Achieves strong cross-modal alignment and generalization across image-text benchmarks. |
| LLaVALiu et al. (2023) | Integrates CLIP and LLaMA for open-ended visual question answering and dialogue. |
| LLaVA-MedLi et al. (2023) | Adapts LLaVA for medical vision-language tasks including medical image question answering. |
| Med-FlamingoMoor et al. (2023) | Extends Flamingo to the medical domain with few-shot learning capabilities. |
| Janus-ProChen et al. (2025) | Uses bidirectional multimodal modeling to enhance multi-turn visual-language interactions. |

Our detailed analysis reveals that MGLL integration yields significant improvements across all seven multimodal architectures. InstructBLIP demonstrates a 9.76% overall accuracy improvement (55.17% to 64.94%), with particularly notable enhancements in challenging conditions such as Retinitis (11.11% to 44.44%) and Media Haze (16.13% to 45.16%). These improvements highlight MGLL's capacity to enhance feature extraction for complex ophthalmological pathologies. MGLL integration showcases substantial performance gains across multiple diagnostic tasks and architectures. Qwen-VL with MGLL integration exhibits a 6.58% overall improvement (76.80% to 83.39%), with remarkable advances in low-prevalence conditions including Astigmatic Refractive Error (0.00% to 50.00%) and Hypertensive Retinopathy (0.00% to 66.67%). Even high-performing models benefit significantly, InternVL achieves a 5.19% improvement with MGLL integration, enhancing diagnostic accuracy particularly for conditions such as Optic Disc Edema (45.45% to 63.64%) and Central Retinal Vein Occlusion (36.36% to 63.64%). LLaVA exhibits similarly robust baseline performance (85.22%), yet MGLL integration yields a 5.55% improvement, achieving near-perfect accuracy in several categories including No Diabetic Macular Edema (90.91% to 100.00%) and miscellaneous conditions (90.00% to 100.00%). The most dramatic improvements occur in medical-specialized models. Med-Flamingo demonstrates a substantial 21.76% improvement (49.17% to 70.94%), with particularly significant gains in Glaucoma (24.07% to 61.11%) and Diabetic Retinopathy (36.91% to 80.54%). Similarly, LLaVA-Med shows a 20.78% improvement (56.47% to 77.25%), with exceptional gains in AMD (16.37% to 58.48%) and Diabetic Retinopathy (26.85% to 77.18%). Across all evaluated models, we observe consistent patterns of improvement that highlight MGLL's particular efficacy with complex retinal conditions, vascular pathologies, and conditions requiring fine-grained feature discrimination. The results demonstrate that MGLL provides substantial benefits regardless of the underlying model architecture. The multiple-choice evaluation framework presented models with standardized diagnostic queries. Some prompt examples of the multiple-choice evaluation benchmark are as follows:

**Question 1**: "What is the most reasonable diagnosis? A. Glaucoma B. Drusen C. Chorioretinitis D. Hypertensive Retinopathy Answer with the option's letter from the given choices directly."

**Answer 1**: A.

**Question 2**: "What diagnosis is most likely? A. Central Serous Retinopathy B. Media Haze C. Diabetic Retinopathy D. Age-related Macular degeneration Answer with the option's letter from the given choices directly."

**Answer 2**: D.

**Question 3**: "What diagnosis is most probable? A. Optic Disk Cupping B. Mild Non-Proliferative Diabetic Retinopathy C. Central Serous Retinopathy D. Central Retinal Vein Occlusion Answer with the option's letter from the given choices directly."

**Answer 3**: B.

Table 27: Comparison of multiple-choice accuracy with MGLL in InstructBLIP on the multiple-choice evaluation benchmark.

| Label Name | AMD | AR | BRVO | Cataract | Chorioretinitis | CN | CRVO | CSR |
|---|---|---|---|---|---|---|---|---|
| InstructBLIP Dai et al. (2023) | 80.17% | 0.00% | 50.00% | 80.00% | 0.00% | 75.00% | 45.45% | 0.00% |
| + MGLL | **83.63%** | **50.00%** | **62.50%** | **85.00%** | **33.33%** | **75.00%** | **63.64%** | **28.57%** |
| Diabetic Retinopathy | Drusen | Glaucoma | Health | HR | DME | MH | Myopia | No AMD |
| 76.51% | 36.67% | 59.30% | 51.57% | 0.00% | 63.79% | 16.13% | 44.25% | 54.34% |
| **82.55%** | **50.00%** | **65.43%** | **60.22%** | **33.33%** | **74.14%** | **45.16%** | **52.65%** | **70.10%** |
| No Glaucoma | No DME | ODC | ODE | ODP | Other | Retinitis | Tessellation | Overall ↑ |
| 41.67% | 63.64% | 65.63% | 27.27% | 50.00% | 58.00% | 11.11% | 41.67% | 55.17% |
| **58.33%** | **72.73%** | **75.00%** | **45.45%** | **50.00%** | **70.00%** | **44.44%** | **58.33%** | **64.94%** (9.76% ↑) |

Table 28: Comparison of multiple-choice accuracy with MGLL in Mini-Gemini on the multiple-choice evaluation benchmark.

| Label Name | AMD | AR | BRVO | Cataract | Chorioretinitis | CN | CRVO | CSR |
|---|---|---|---|---|---|---|---|---|
| Mini-Gemini Li et al. (2024) | 76.61% | 0.00% | 43.75% | 85.00% | 0.00% | 25.00% | 27.27% | 14.29% |
| + MGLL | **82.46%** | **0.00%** | **56.25%** | **85.00%** | **66.67%** | **50.00%** | **54.55%** | **42.86%** |
| Diabetic Retinopathy | Drusen | Glaucoma | Health | HR | DME | MH | Myopia | No AMD |
| 79.87% | 23.33% | 67.90% | 61.46% | 0.00% | 60.34% | 38.71% | 58.41% | 63.02% |
| **84.56%** | **46.67%** | **72.22%** | **68.99%** | **33.33%** | **65.52%** | **58.06%** | **64.16%** | **72.99%** |
| No Glaucoma | No DME | ODC | ODE | ODP | Other | Retinitis | Tessellation | Overall ↑ |
| 66.67% | 72.73% | 62.50% | 36.36% | 50.00% | 66.00% | 33.33% | 33.33% | 62.65% |
| **75.00%** | **81.82%** | **78.13%** | **45.45%** | **50.00%** | **74.00%** | **55.56%** | **41.67%** | **70.58%** (7.93% ↑) |

Table 29: Comparison of multiple-choice accuracy with MGLL in Qwen-VL on the multiple-choice evaluation benchmark.

| Label Name | AMD | AR | BRVO | Cataract | Chorioretinitis | CN | CRVO | CSR |
|---|---|---|---|---|---|---|---|---|
| Qwen-VL Bai et al. (2023) | 81.87% | 0.00% | 43.75% | 75.00% | 0.00% | 25.00% | 9.09% | 28.57% |
| + MGLL | **85.96%** | **50.00%** | **62.50%** | **80.00%** | **33.33%** | **75.00%** | **36.36%** | **42.86%** |
| Diabetic Retinopathy | Drusen | Glaucoma | Health | HR | DME | MH | Myopia | No AMD |
| 80.54% | 26.67% | 78.40% | 79.89% | 0.00% | 84.48% | 54.84% | 76.55% | 82.96% |
| **89.93%** | **43.33%** | **87.04%** | **85.39%** | **66.67%** | **89.66%** | **70.97%** | **80.97%** | **87.14%** |
| No Glaucoma | No DME | ODC | ODE | ODP | Other | Retinitis | Tessellation | Overall ↑ |
| 75.00% | 72.73% | 56.25% | 27.27% | 50.00% | 84.00% | 22.22% | 25.00% | 76.80% |
| **83.33%** | **72.73%** | **68.75%** | **54.55%** | **100.00%** | **86.00%** | **33.33%** | **41.67%** | **83.39%** (6.58% ↑) |

Table 30: Comparison of multiple-choice accuracy with MGLL in InternVL on the multiple-choice evaluation benchmark.

| Label Name | AMD | AR | BRVO | Cataract | Chorioretinitis | CN | CRVO | CSR |
|---|---|---|---|---|---|---|---|---|
| InternVL Chen et al. (2024a) | 81.29% | 0.00% | 37.50% | 85.00% | 0.00% | 25.00% | 36.36% | 71.43% |
| + MGLL | **86.55%** | **50.00%** | **56.25%** | **90.00%** | **0.00%** | **50.00%** | **63.64%** | **71.43%** |
| Diabetic Retinopathy | Drusen | Glaucoma | Health | HR | DME | MH | Myopia | No AMD |
| 94.63% | 43.33% | 89.51% | 85.73% | 33.33% | 87.93% | 64.52% | 88.05% | 85.85% |
| **96.64%** | **53.33%** | **90.74%** | **91.46%** | **66.67%** | **94.83%** | **67.74%** | **91.15%** | **90.68%** |
| No Glaucoma | No DME | ODC | ODE | ODP | Other | Retinitis | Tessellation | Overall ↑ |
| 91.67% | 81.82% | 87.50% | 45.45% | 50.00% | 90.00% | 44.44% | 66.67% | 84.33% |
| **100.00%** | **90.91%** | **93.75%** | **63.64%** | **50.00%** | **96.00%** | **55.56%** | **75.00%** | **89.52%** (5.19% ↑) |

Table 31: Comparison of multiple-choice accuracy with MGLL in LLaVA on the multiple-choice evaluation benchmark.

| Label Name | AMD | AR | BRVO | Cataract | Chorioretinitis | CN | CRVO | CSR |
|---|---|---|---|---|---|---|---|---|
| LLaVA Liu et al. (2023) | 83.04% | 0.00% | 50.00% | 90.00% | 0.00% | 25.00% | 9.09% | 42.86% |
| + MGLL | **84.80%** | **100.00%** | **68.75%** | **90.00%** | **33.33%** | **25.00%** | **45.45%** | **57.14%** |
| Diabetic Retinopathy | Drusen | Glaucoma | Health | HR | DME | MH | Myopia | No AMD |
| 87.25% | 36.67% | 91.36% | 88.65% | 0.00% | 93.10% | 48.39% | 88.50% | 90.68% |
| **93.96%** | **50.00%** | **91.98%** | **94.38%** | **33.33%** | **96.55%** | **61.29%** | **90.71%** | **96.78%** |
| No Glaucoma | No DME | ODC | ODE | ODP | Other | Retinitis | Tessellation | Overall ↑ |
| 100.00% | 90.91% | 62.50% | 18.18% | 50.00% | 90.00% | 44.44% | 58.33% | 85.22% |
| **100.00%** | **100.00%** | **62.50%** | **45.45%** | **100.00%** | **100.00%** | **66.67%** | **66.67%** | **90.77%** (5.55% ↑) |

Table 32: Comparison of multiple-choice accuracy with MGLL in LLaVA-Med on the multiple-choice evaluation benchmark.

| Label Name | AMD | AR | BRVO | Cataract | Chorioretinitis | CN | CRVO | CSR |
|---|---|---|---|---|---|---|---|---|
| LLaVA-Med Li et al. (2023) | 16.37% | 100.00% | 31.25% | 15.00% | 0.00% | 0.00% | 45.45% | 42.86% |
| + MGLL | **58.48%** | **100.00%** | **50.00%** | **65.00%** | **66.67%** | **25.00%** | **63.64%** | **57.14%** |
| Diabetic Retinopathy | Drusen | Glaucoma | Health | HR | DME | MH | Myopia | No AMD |
| 26.85% | 16.67% | 25.31% | 91.46% | 33.33% | 16.67% | 25.81% | 23.89% | 66.56% |
| **77.18%** | **40.00%** | **59.26%** | **97.42%** | **33.33%** | **55.17%** | **51.61%** | **57.08%** | **78.46%** |
| No Glaucoma | No DME | ODC | ODE | ODP | Other | Retinitis | Tessellation | **Overall ↑** |
| 16.67% | 36.36% | 21.88% | 27.27% | 0.00% | 28.00% | 33.33% | 16.67% | 56.47% |
| **41.67%** | **72.73%** | **40.63%** | **63.64%** | **50.00%** | **62.00%** | **44.44%** | **58.33%** | **77.25%** (20.78% ↑) |

Table 33: Comparison of multiple-choice accuracy with MGLL in Med-Flamingo on the multiple-choice evaluation benchmark.

| Label Name | AMD | AR | BRVO | Cataract | Chorioretinitis | CN | CRVO | CSR |
|---|---|---|---|---|---|---|---|---|
| Med-Flamingo Moor et al. (2023) | 25.73% | 100.00% | 31.25% | 30.00% | 0.00% | 25.00% | 63.64% | 57.14% |
| + MGLL | **69.01%** | **100.00%** | **56.25%** | **75.00%** | **33.33%** | **50.00%** | **72.73%** | **71.43%** |
| Diabetic Retinopathy | Drusen | Glaucoma | Health | HR | DME | MH | Myopia | No AMD |
| 36.91% | 20.00% | 24.07% | 74.16% | 33.33% | 24.14% | 22.58% | 18.58% | 52.73% |
| **80.54%** | **43.33%** | **61.11%** | **83.37%** | **66.67%** | **51.72%** | **54.84%** | **45.58%** | **69.13%** |
| No Glaucoma | No DME | ODC | ODE | ODP | Other | Retinitis | Tessellation | **Overall ↑** |
| 16.67% | 36.36% | 43.75% | 36.36% | 0.00% | 28.00% | 22.22% | 8.33% | 49.17% |
| **50.00%** | **63.64%** | **59.38%** | **63.64%** | **50.00%** | **70.00%** | **44.44%** | **33.33%** | **70.94%** (21.76% ↑) |

Table 34: Comparison of multiple-choice accuracy with MGLL in Janus-Pro on the multiple-choice evaluation benchmark.

| Label Name | AMD | AR | BRVO | Cataract | Chorioretinitis | CN | CRVO | CSR |
|---|---|---|---|---|---|---|---|---|
| Janus-Pro Chen et al. (2025) | 88.30% | 50.00% | 56.25% | 75.00% | 33.33% | 25.00% | 36.36% | 42.86% |
| + MGLL | **90.64%** | **100.00%** | **62.50%** | **85.00%** | **66.67%** | **75.00%** | **63.64%** | **71.43%** |
| Diabetic Retinopathy | Drusen | Glaucoma | Health | HR | DME | MH | Myopia | No AMD |
| 93.29% | 40.00% | 90.74% | 96.40% | 33.33% | 62.07% | 58.06% | 87.17% | 92.28% |
| **96.64%** | **53.33%** | **95.06%** | **96.63%** | **66.67%** | **70.69%** | **67.74%** | **90.27%** | **94.21%** |
| No Glaucoma | No DME | ODC | ODE | ODP | Other | Retinitis | Tessellation | **Overall ↑** |
| 58.33% | 72.73% | 81.25% | 36.36% | 50.00% | 82.00% | 33.33% | 58.33% | 88.54% |
| **83.33%** | **90.91%** | **87.50%** | **54.55%** | **50.00%** | **92.00%** | **55.56%** | **75.00%** | **91.85%** (3.31% ↑) |

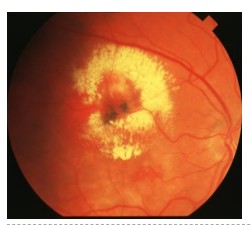

What is the probable diagnosis?
A. Hypertensive Retinopathy
B. Optic Disc Pallor
C. Central Serous Retinopathy
D. Choroidal Neovascularization
Answer with the option's letter from the given choices directly.
**Ground Truth: D**

InstructBLIP: D    Mini-Gemini: C    Qwen-VL: C    InternVL: C
InstructBLIP + MGLL: D    Mini-Gemini + MGLL: D    Qwen-VL + MGLL: D    InternVL + MGLL: D

Med-Flamingo: A    LLaVA-Med: A    LLaVA: C    Janus-Pro: C
Med-Flamingo + MGLL: D    LLaVA-Med + MGLL: D    LLaVA + MGLL: A    Janus-Pro + MGLL: D

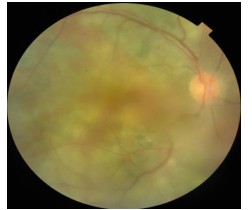

What diagnosis is most probable?
A. Arteriosclerotic Retinopathy
B. Chorioretinitis
C. Retinitis
D. Healthy
Answer with the option's letter from the given choices directly.
**Ground Truth: B**

InstructBLIP: A    Mini-Gemini: A    Qwen-VL: D    InternVL: D
InstructBLIP + MGLL: B    Mini-Gemini + MGLL: B    Qwen-VL + MGLL: B    InternVL + MGLL: C

Med-Flamingo: C    LLaVA-Med: C    LLaVA: D    Janus-Pro: D
Med-Flamingo + MGLL: B    LLaVA-Med + MGLL: B    LLaVA + MGLL: B    Janus-Pro + MGLL: B

Figure 5: Case Studies (Top: Case 1, Bottom: Case 2) Demonstrating MGLL Integration Impact on Diagnostic Accuracy of Different Multimodal Large Langue Models (MLLMs).

The Fig. 5 presents two representative case studies demonstrating the diagnostic impact of MGLL integration across multiple MLLMs.

Case 1 displays a fundus image with characteristic features of choroidal neovascularization (CNV), including a well-defined yellowish lesion with surrounding subretinal hemorrhage in the macula. Only InstructBLIP correctly identifies this pathology in its baseline configuration, whereas five models with MGLL integration provide accurate diagnoses, demonstrating MGLL's capacity to enhance the detection of vascular abnormalities.

Case 2 exhibits subtle inflammatory changes consistent with chorioretinitis, characterized by chorioretinal infiltrates against a background of mild vitreous haze, which is a condition none of the baseline models correctly identify. Following MGLL integration, six models accurately diagnose this inflammatory condition, with responses shifting from incorrect options (Arteriosclerotic Retinopathy, Retinitis, or Healthy) to the correct identification.

### D.5 More CAMs on Retinal Fundus Datasets

We present additional class activation maps (CAMs) from CLIP and MGLL on downstream retinal datasets in Fig. 6. These images include cases of diabetic retinopathy, diabetic macular edema, and glaucoma. Through both linear probing and full fine-tuning approaches, MGLL consistently demonstrates more precise lesion localization than CLIP, specifically highlighting pathological features rather than producing diffuse, non-specific activations. In diabetic retinopathy cases spanning mild to proliferative stages, MGLL accurately identifies microaneurysms, hemorrhages, venous beading, and neovascularization sites. While in diabetic macular edema, it effectively localizes retinal thickening and exudate formation with activation intensity proportional to disease severity. For glaucoma, MGLL appropriately focuses on optic disc abnormalities, cup enlargement, and neural rim thinning—critical diagnostic markers often missed by CLIP, which tends to highlight anatomical landmarks regardless of pathological relevance. These findings demonstrate MGLL's advantages for ophthalmological applications, offering more robust performance for clinical feature detection that supports diagnostic confidence.

### D.6 Ablation Study on Weight Factors of Loss

To investigate the impact of different weight factors in our composite loss function, we conducted an extensive ablation study using the RFMiD dataset as shown in Table 35. Our loss function incorporates three components with corresponding weight factors ($\alpha_1$, $\alpha_2$, and $\alpha_3$), and the results demonstrate that weight selection significantly influences model performance across all metrics. Compared to the baseline CLIP model, all our weight configurations show substantial improvements, with the optimal configuration ($\alpha_1 = 0.5$, $\alpha_2 = 1.0$, and $\alpha_3 = 1.0$) achieving the best performance in both linear probing (79.62% AUC, 92.84% ACC, 34.08% mAP) and full fine-tuning (92.83% AUC, 95.48% ACC, 64.99% mAP) scenarios. Notably, reducing the weight of the first component improved performance, while reducing either the second or third component weights resulted in performance degradation, suggesting that the information captured by these components is particularly valuable for medical image classification tasks and should be emphasized during training. These findings highlight the importance of appropriate loss weighting in multi-component objective functions and provide empirical evidence for the optimal configuration selection.

Table 35: Ablations of Weight Factors on RFMiD. **Bold** indicates best performance and underline shows second-best.

| Method | Linear Probe (%) | | | Fully Fine-tune (%) | | |
|---|---|---|---|---|---|---|
| | AUC | ACC | mAP | AUC | ACC | mAP |
| CLIP Radford et al. (2021) | 44.66 | 92.53 | 7.28 | 65.10 | 92.86 | 17.31 |
| $\alpha_1$: 1.0, $\alpha_2$: 1.0, $\alpha_3$: 1.0 | 79.29 | 92.83 | 33.82 | 92.51 | 95.35 | 64.57 |
| $\alpha_1$: 0.5, $\alpha_2$: 1.0, $\alpha_3$: 1.0 | **79.62** | **92.84** | **34.08** | **92.83** | **95.48** | **64.99** |
| $\alpha_1$: 1.0, $\alpha_2$: 0.5, $\alpha_3$: 1.0 | 78.11 | 92.79 | 32.42 | 91.46 | 94.99 | 63.28 |
| $\alpha_1$: 1.0, $\alpha_2$: 1.0, $\alpha_3$: 0.5 | 78.85 | 92.80 | 33.39 | 92.01 | 95.18 | 63.77 |

## D.7 Ablation studies on the temperature coefficient ($\tau$)

We have conducted the ablation studies of temperature coefficient and observe that the performance first improves and then drops as the temperature coefficient $\tau$ increases, as shown in Table 36. A smaller $\tau$ sharpens the similarity distribution, enhancing discrimination but causing training instability. Conversely, a larger $\tau$ produces smoother gradients but weakens alignment. The best results are achieved when $\tau = 0.07$, which provides a good balance between discriminative alignment and stable optimization.

Table 36: Ablations of temperature coefficient ($\tau$) on MIDRC-XR-Portable.

| Method | Linear Probe (%) | | | Fully Fine-tune (%) | | |
|---|---|---|---|---|---|---|
| | AUC | ACC | mAP | AUC | ACC | mAP |
| CLIP, $\tau = 0.07$ | 71.43 | 78.22 | 22.31 | 91.83 | 90.08 | 83.94 |
| MGLL, $\tau = 0.05$ | 83.55 | 88.91 | 30.49 | 99.60 | 98.67 | 89.71 |
| MGLL, $\tau = 0.20$ | 81.89 | 87.02 | 28.93 | 97.89 | 97.26 | 87.92 |
| MGLL, $\tau = 0.50$ | 79.52 | 86.19 | 27.74 | 95.53 | 94.57 | 85.95 |
| MGLL, $\tau = 0.07$ | **83.86** | **89.06** | **30.62** | **99.75** | **98.80** | **89.87** |

## D.8 Performance on datasets with artificially introduced noise

We evaluated the robustness of MGLL under varying levels of artificially introduced noise, where 10%–30% of granularity labels were randomly removed. As shown in Table 37, even with 30% missing labels, MGLL achieves AUCs of 79.61% (linear probing) and 96.74% (full fine-tuning), which remain substantially higher than CLIP trained with complete labels (71.43% and 91.83%, respectively). These results demonstrate that MGLL maintains strong robustness against incomplete or noisy granularity supervision.

Table 37: Ablations of missing granularity labels on MIDRC-XR-Portable.

| Method | Linear Probe (%) | | | Fully Fine-tune (%) | | |
|---|---|---|---|---|---|---|
| | AUC | ACC | mAP | AUC | ACC | mAP |
| CLIP, No Missing | 71.43 | 78.22 | 22.31 | 91.83 | 90.08 | 83.94 |
| MGLL, 10% Missing | 82.58 | 88.15 | 29.97 | 99.31 | 98.39 | 89.30 |
| MGLL, 20% Missing | 81.14 | 87.25 | 28.86 | 98.62 | 97.95 | 88.74 |
| MGLL, 30% Missing | 79.61 | 86.23 | 27.75 | 96.74 | 96.02 | 87.28 |
| MGLL, No Missing | **83.86** | **89.06** | **30.62** | **99.75** | **98.80** | **89.87** |

## D.9 Performance on datasets with mixing granularity levels

We have evaluated MGLL on datasets with mixed granularity levels, and the results are reported in Table 38. Specifically, the pretraining dataset was randomly divided into two subsets of equal size: Set A with two levels of granularity and Set B with a single level. MGLL achieves comparable performance across both subsets and their combination, demonstrating its robustness to heterogeneous annotation structures and its applicability to real-world scenarios with mixed-granularity data.

Table 38: Ablations of mixing granularity levels on MIDRC-XR-Portable. The dataset is randomly divided into two subsets of equal size for the ablation study, Set A (50% data) and Set B (50% data).

| Study Desc. | Series Desc. | Linear Probe (%) | | | Fully Fine-tune (%) | | |
|---|---|---|---|---|---|---|---|
| | | AUC | ACC | mAP | AUC | ACC | mAP |
| Set A | Set B | 78.93 | 85.47 | 27.04 | 94.81 | 93.58 | 85.14 |
| Set A | Set A + Set B | 80.25 | 86.46 | 27.97 | 96.21 | 94.87 | 86.72 |
| Set A + Set B | Set B | 80.92 | 87.07 | 28.63 | 97.03 | 96.56 | 87.45 |
| MGLL (Ours) | | **83.86** | **89.06** | **30.62** | **99.75** | **98.80** | **89.87** |

# E   DISCUSSION AND FUTURE WORK

Our investigation into Multi-Granular Language Learning (MGLL) reveals several important insights about vision-language alignment in complex domains. The consistent performance improvements across various medical imaging datasets demonstrate that hierarchical textual information substantially enhances visual understanding, particularly when images correspond to multiple clinical findings at different levels of specificity. The ablation studies confirm that performance gains scale with both the number of granularity levels and the quality of input data, suggesting that MGLL effectively leverages the complementary information contained in multi-granular textual descriptions.

While MGLL achieves simultaneous multi-label and cross-granularity alignment without additional computational cost, further optimization could potentially improve its generality. Future work should explore several directions: (1) extending MGLL to incorporate multimodal inputs beyond images and text, such as patient metadata or temporal information; (2) investigating domain adaptation techniques to improve generalization to unseen medical conditions or imaging modalities; and (3) exploring the integration of MGLL with large language models to generate more nuanced textual descriptions at multiple granularities. Additionally, applying MGLL to other domains with inherently hierarchical structures, such as satellite imagery or scientific visualization, could further validate its broader applicability beyond medical imaging. These extensions would strengthen MGLL's position as a generalizable framework for improved multimodal understanding.

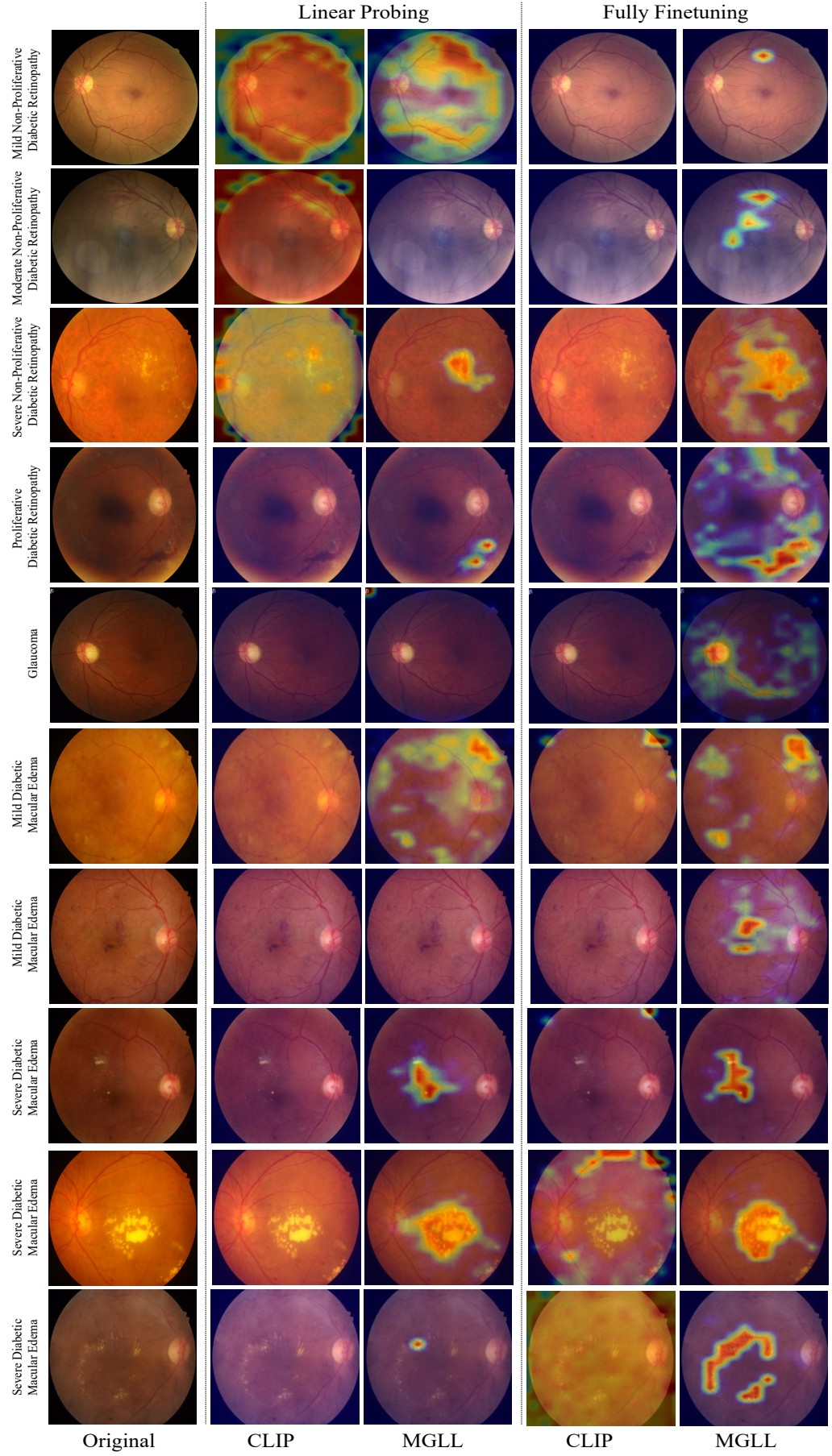

Figure 6: More Class Activation Maps from CLIP and Proposed MGLL.

