# OpenReview forum: "Boosting Medical Visual Understanding From Multi-Granular Language Learning"
_ICLR.cc/2026/Conference — ICLR 2026 Poster_

### Official Review · Reviewer_pAv1 · 2025-10-27

**Soundness:** 3
**Presentation:** 3
**Contribution:** 3
**Rating:** 8
**Confidence:** 4

**Summary:**

The paper proposes MGLL (Multi-Granular Language Learning) which is a training method that extends CLIP to jointly handle multi-label supervision and text granularities in medical imaging. It initroduces three complementary losses: soft-CLIP, point-wise BCE, and smooth KL. The model trained with MGLL outperforms prior CLIP-style baselines across benchmarks and also boost downstream MLLM performance.

**Strengths:**

1) The study introduces a triple loss function to for training medical image-text to capture multi-granular level meaning, which standard CLIP does not handle.

2) The study curates two multi-granular datasets, MGLL-Funds and MGLL-Xray.

3) The paper explain the motivation for each term of the MGLL with empirical analyses.

4) The paper provides strong, consistent grains across benchmarks in both linear-probe and full fine-tune settings.

5) MGLL can be easily adopt in other medical domains that have hierarchical labels or reports.

**Weaknesses:**

1) The study does not provide report on the overlap between MGLL-Fundus (aggregates from 49 sources) and target datasets. Possible risky for data leakage.

2) The paper claims MGLL is computationally efficient due to no additional encoders, but provides no evidence such as training time, memory usage, compared to baselines like CLIP.

3) The paper does not provide clinical relevant metrics such as F1, or sensitive, specificity.

**Questions:**

1) Can the authors provide experimental results showing MGLL's performance on datasets with artificially introduced noise, such as 10%–30% missing granularity labels?

2)  Can the authors provide experimental results showing MGLL's performance on datasets with mixing granularity levels? For example, some samples have 2 level of granularity level, while others have 1 level.

3) Have clinical experts validated MGLL's alignments or its class activation maps for clinical relevance?

4) Can the authors provide specific metrics (e.g., training time, GPU memory usage) comparing MGLL to CLIP or other baselines on the MGLL-Fundus dataset?

---

> ### Author Response · Authors · 2025-11-19
>
> Thank you for raising these concerns.
>
> For **Q1 in Weaknesses**, details of both the pretraining and downstream datasets are provided in Appendix B. We strictly followed the official data splits of all downstream datasets. During pretraining, we only used the training sets for model training and the validation sets for pretraining evaluation, while the test sets were never accessed. This ensures that no data leakage occurred between the pretraining and downstream evaluation stages, and all reported results are based on fair and independent evaluations. We have explicitly clarified this implementation detail in Section 4.1 of the revised paper.
>
> For **Q2 in Weaknesses** and **Q4 in Questions**, we describe MGLL as computationally efficient because it introduces no additional encoders or heavy modules beyond the standard CLIP architecture. All proposed objectives (soft CLIP, point-wise, and smooth KL losses) are applied within the existing embedding space, adding only lightweight similarity and divergence computations. In practice, the computational overhead is negligible compared to CLIP training, since these losses operate on already-encoded features without increasing the number of forward passes or parameters. The total pretraining time of our MGLL is 7 hours 20 minutes on one GPU of NVIDIA RTX 4090, while that of standard CLIP is 7 hours 10 minutes. The peak GPU memory usage of our MGLL is 16.21 GB, while that of standard CLIP is 16.14 GB. These outcomes indicate that MGLL adds less than 3\% overhead compared with CLIP, as it reuses the same encoders and only introduces lightweight loss computations.
>
> For **Q3 in Weaknesses**, thank you for the valuable suggestion. In our paper, we primarily reported AUC, mAP, and ACC, which are strongly correlated with clinically relevant metrics such as F1-score, sensitivity, and specificity. For completeness, we also computed these metrics on our downstream datasets. For example, on OIA-DDR, MGLL achieves an F1-score of 40.83\% / 48.81\%, sensitivity of 39.78\% / 46.47\%, and specificity of 89.69\% / 90.24\% under linear probing / full fine-tuning, respectively. In contrast, CLIP obtains 19.02\% / 41.57\%, 23.03\% / 40.48\%, and 82.59\% / 89.76\%. These results demonstrate that MGLL consistently improves clinically relevant performance metrics across multiple evaluation settings.
>
> For **Q1 in Questions**, thank you for the feedback. We evaluated the robustness of MGLL under varying levels of artificially introduced noise, where 10\%–30\% of granularity labels were randomly removed. Even with 30\% missing labels, MGLL achieves AUCs of 79.61\% (linear probing) and 96.74\% (full fine-tuning), which remain substantially higher than CLIP trained with complete labels (71.43\% and 91.83\%, respectively). These results demonstrate that MGLL maintains strong robustness against incomplete or noisy granularity supervision. We have supplemented these results in Appendix D.8.
>
> | **Method**           |        |   **Linear Probe**  |             |            |  **Full Fine-tune**  |             |
> | -------------------- | :--------------: | :---------: | :---------: | :----------------: | :---------: | :---------: |
> |                      |    **AUC (%)**   | **ACC (%)** | **mAP (%)** |     **AUC (%)**    | **ACC (%)** | **mAP (%)** |
> | CLIP, No Missing     |       71.43      |    78.22    |    22.31    |        91.83       |    90.08    |    83.94    |
> | MGLL, 10% Missing    |       82.58      |    88.15    |    29.97    |        99.31       |    98.39    |    89.30    |
> | MGLL, 20% Missing    |       81.14      |    87.25    |    28.86    |        98.62       |    97.95    |    88.74    |
> | MGLL, 30% Missing    |       79.61      |    86.23    |    27.75    |        96.74       |    96.02    |    87.28    |
> | **MGLL, No Missing** |     **83.86**    |  **89.06**  |  **30.62**  |      **99.75**     |  **98.80**  |  **89.87**  |

---

> ### Author Response · Authors · 2025-11-19
>
> For **Q2 in Questions**, we have evaluated MGLL on datasets with mixed granularity levels, and the results are reported in the table below. Specifically, the pretraining dataset was randomly divided into two subsets of equal size: Set A with two levels of granularity and Set B with a single level. MGLL achieves comparable performance across both subsets and their combination, demonstrating its robustness to heterogeneous annotation structures and its applicability to real-world scenarios with mixed-granularity data. We have supplemented these results in Appendix D.9.
>
> | **Study Desc.** | **Series Desc.** |       |  **Linear Probe**  |             |       |   **Full Fine-tune**  |             |
> | --------------- | ---------------- | :--------------: | :---------: | :---------: | :----------------: | :---------: | :---------: |
> |                 |                  |    **AUC (%)**   | **ACC (%)** | **mAP (%)** |     **AUC (%)**    | **ACC (%)** | **mAP (%)** |
> | Set A           | Set B            |       78.93      |    85.47    |    27.04    |        94.81       |    93.58    |    85.14    |
> | Set A           | Set A + Set B    |       80.25      |    86.46    |    27.97    |        96.21       |    94.87    |    86.72    |
> | Set A + Set B   | Set B            |       80.92      |    87.07    |    28.63    |        97.03       |    96.56    |    87.45    |
> | **MGLL (Ours)** |                  |     **83.86**    |  **89.06**  |  **30.62**  |      **99.75**     |  **98.80**  |  **89.87**  |
>
>
> For **Q3 in Questions**, the clinical relevance of MGLL’s alignments and class activation maps (CAMs) has been examined in collaboration with clinical experts. We randomly selected 5% of the generated CAMs for expert verification, and the experts confirmed that our visual findings were clinically consistent, as described in Section 4.2.1. Figure 4 illustrates CAM visualizations from CLIP and MGLL on two representative retinal disease cases. CLIP fails to capture meaningful regions, showing nearly uniform attention across the fundus images. In contrast, MGLL accurately localizes clinically significant regions of interest (ROIs)—for example, highlighting hard exudates in chorioretinitis and the retinal pigment epithelium in age-related macular degeneration. These expert validations confirm the clinical plausibility of MGLL’s visual focus.

---

> > ### Comment · Reviewer_pAv1 · 2025-11-25
> >
> > Thank you very much for your high quality revision. All my concern is addressed, I will retrain my score.

---

> > > ### Author Response · Authors · 2025-11-25
> > >
> > > Thank you for the thoughtful follow-up! I appreciate your careful review and am glad the revisions met your expectations. I’ve integrated your suggestions to improve precision and readability throughout the work, and your feedback has genuinely strengthened the final version.

---

### Official Review · Reviewer_Bhm2 · 2025-10-29

**Soundness:** 4
**Presentation:** 3
**Contribution:** 3
**Rating:** 6
**Confidence:** 3

**Summary:**

This paper proposes a novel medical vision-language pre-training framework, Multi-Granular Language Learning (MGLL), designed to achieve both multi-label and cross-granularity image-text alignment. Building upon CLIP, MGLL introduces three key components: soft-label contrastive loss, point-wise binary supervision loss, and smooth KL divergence loss, to enhance cross-granularity consistency. The authors construct two large-scale medical image-text datasets, MGLL-Fundus (fundus images) and MGLL-Xray (X-ray images), for model pre-training. Experiments demonstrate that MGLL significantly outperforms existing state-of-the-art methods on 11 downstream tasks and effectively boosts the performance of multimodal large language models (MLLMs).

**Strengths:**

1. The overall contribution is clear and highly practical, offering significant utility for addressing the multi-level semantic complexity inherent in medical images.
2. Experiments are comprehensive and results are significant:
2.1. Covers multiple datasets (Fundus and X-ray) and 11 downstream tasks;
2.2. Validates various application scenarios, including linear probing, full fine-tuning, and integration with MLLMs;
2.3. Significantly outperforms existing CLIP variants on the majority of tasks, demonstrating strong experimental consistency.

**Weaknesses:**

1. The details of dataset construction are insufficient.
2. The innovation leans more towards a compositional approach, rather than proposing entirely new learning principles or optimization mechanisms.
3. Lacks ablation studies on the temperature coefficient (τ).

**Questions:**

1. How is the multi-granular text supervision generated?
2. Is there a potential risk of trivial solutions in the smooth KL divergence?

---

> ### Author Response · Authors · 2025-11-19
>
> Thank you for the suggestions.
>
> For **Q1 in Weaknesses** and **Q1 in Questions**, we have added detailed information on dataset construction in Appendix B.1. Specifically, we developed MGLL-Fundus, a comprehensive multi-granularity fundus image–text dataset containing $246,389$ image–text pairs. This dataset integrates fundus images from 49 public sources, covering over 50 disease categories. The data distribution and representative examples are provided in Tables 9 and 10. The textual annotations in MGLL-Fundus are organized into two granularity levels: disease category and clinical explanation. To ensure broader applicability, we also curated $190,882$ X-ray images from the MIDRC repository. The images were converted from DICOM to PNG format while preserving key metadata. The corresponding multi-granularity textual annotations comprise three levels: modality, study description, and series description. Dataset distribution and examples are shown in Tables 11 and 12.
>
> For **Q2 in Weaknesses**, while our method combines existing contrastive and language–vision principles, the contribution lies in designing a comprehensive learning framework and optimization strategy that effectively leverages multi-label and multi-granularity supervision in medical datasets. Specifically, we (1) construct two large-scale multi-granular datasets covering both fundus and X-ray images, (2) integrate structured multi-label and cross-granularity textual supervision through the soft CLIP loss and point-wise constraint, and (3) introduce a smooth KL divergence term to enforce cross-granularity consistency while maintaining computational efficiency.
>
> Compared with existing method, our MGLL provides a more flexible and generalizable framework for learning both multi-label and cross-granularity visual–language representations. MGLL can effectively utilize different types of granularity information across diverse datasets without requiring specific annotation formats or model architectures. MGLL also displays robust performance under complex scenarios such as mixed granularity and noised annotations. All data and source code will be released upon acceptance to facilitate future research in this direction.

---

> ### Author Response · Authors · 2025-11-25
>
> For **Q3 in Weaknesses**, we have conducted the ablation studies of temperature coefficient and observe that the performance first improves and then drops as the temperature coefficient $\tau$ increases. A smaller $\tau$ sharpens the similarity distribution, enhancing discrimination but causing training instability. Conversely, a larger $\tau$ produces smoother gradients but weakens alignment. The best results are achieved when $\tau=0.07$, which provides a good balance between discriminative alignment and stable optimization. We have supplemented these results in Appendix D.7.
>
> | **Method**         |       |       **Linear Probe**      |             |         |       **Full Fine-tune**    |             |
> | ------------------ | :--------------: | :---------: | :---------: | :----------------: | :---------: | :---------: |
> |                    |    **AUC (%)**   | **ACC (%)** | **mAP (%)** |     **AUC (%)**    | **ACC (%)** | **mAP (%)** |
> | CLIP, τ = 0.07     |       71.43      |    78.22    |    22.31    |        91.83       |    90.08    |    83.94    |
> | MGLL, τ = 0.05     |       83.55      |    88.91    |    30.49    |        99.60       |    98.67    |    89.71    |
> | MGLL, τ = 0.20     |       81.89      |    87.02    |    28.93    |        97.89       |    97.26    |    87.92    |
> | MGLL, τ = 0.50     |       79.52      |    86.19    |    27.74    |        95.53       |    94.57    |    85.95    |
> | **MGLL, τ = 0.07** |     **83.86**    |  **89.06**  |  **30.62**  |      **99.75**     |  **98.80**  |  **89.87**  |
>
> For **Q2 in Questions**, we thank the reviewer for this thoughtful question. In practice, this issue is handled by using the smooth KL only as a regularizer rather than a standalone objective. The discriminative losses (soft-CLIP and the point-wise supervision) keep the logits informative, while the smooth KL encourages consistency across granularities without allowing them to collapse. As long as its weight is set conservatively and trained jointly with the supervised terms, the trivial solutions are suppressed and the KL behaves as a stabilizer that aligns different granularities rather than something that drives the model toward degenerate predictions.
>
> Based on our ablation results (Table 3), enforcing cross-granularity consistency with the smooth KL loss does not harm granularity-specific learning. Performance continuously improves as each objective is added—first with the point-wise loss $L_\text{P}$, then the soft CLIP loss $L_{\text{sCLIP}}$, and finally with the smooth KL loss $L_{\text{sKL}}$, which yields the best overall results. This progressive improvement indicates that the smooth KL term complements, rather than overrides, the granularity-specific objectives, helping the model retain discriminative features while enhancing cross-granularity coherence.

---

> > ### Comment · Reviewer_Bhm2 · 2025-11-25
> >
> > Thank you for your response. Most of my concerns have been addressed. I will keep my score.

---

> > > ### Author Response · Authors · 2025-11-25
> > >
> > > Thank you for the positive feedback! I truly appreciate it. I’m glad to hear that your concerns have been resolved, and I’ve incorporated your earlier suggestions to further refine and strengthen the work.

---

### Official Review · Reviewer_HXxg · 2025-10-30

**Soundness:** 3
**Presentation:** 1
**Contribution:** 2
**Rating:** 2
**Confidence:** 4

**Summary:**

This paper proposed MGLL, a novel multi-grandularity contrastive learning framework for medical visual language pre-training. Different from vanilla CLIP, MGLL conducts contrastive learning through both multi-label and multi-granular optimization. It proposed three new contrastive losses designed for data with multi-granular annotation, showing an improved performance on various tasks.

**Strengths:**

1. The paper has contributed two large-scale image-text pair datasets for fundus and X-ray images, providing multi-granular annotation, which will be very helpful to the field.

2. According to the evaluation and derivation, the proposed losses (soft CLIP loss, point-wise BCE loss, and KL loss) help improve the model's performance on multiple downstream evaluations, showing a uniform improvement against baselines.

3. The ablation experiment is especially detailed, providing strong support to the model design.

4. Code is provided in the supplement.

**Weaknesses:**

My major concern is the clear formatting issue. The paper has clearly adjusted the vertical space between the section and sub-section titles, gaining more space for their content. Table 1 and Figure 4 overlap with each other. The reviewer believes that this is a violation of the conference policy, which suggests "Do not change any aspects of the formatting parameters in the style files. In particular, do not modify the width or length of the rectangle the text should fit into..." Considering that, I am afraid I cannot recommend acceptance regardless of the content of the paper, unless there is clear evidence that suggests otherwise (modifying vertical space is fine, or proving the vertical space is not modified).

Besides the format issue, the reviewer still has concerns about the proposed method. Listed below:

1. In equation (2), the soft CLIP loss is defined based on both $l_\{ik\}$ and $l_\{ki\}$, where the second term (if I understand correctly) refers to text-to-image loss. But this is clearly wrong, since $i$ is for the $i$-th image/text in the batch, but $k$ is the $k$-th label across different granularities, and they are clearly un-interchangeable. According to Figure 2, the soft CLIP loss should be computed over 3 dimensions: image batch, text batch, and granularity. So, the reviewer guessed there might be a missing index in this equation.

2. The point-wise BCE loss optimizes as a multi-label classification loss. However, it is weird that $M$ now serves as the number of **categories**, rather than the number of different **granularities**, and they can't be the same value. Since for the $i$-th image, it is always paired with all its $M_i$ annotations, as they just refer to different granularity. This might just be a typo, but it would still be better to clarify these two values.

3. As for the KL loss, there is no explanation on how $z$ is obtained, which is the "predicted logits". The reviewer assumes this refers to the logits for each category/class of the image. But it still uses $m$ as an index, which makes it very confusing.

4. According the section 3.2, the proposed method relies on category-wise prediction (both point-wise BCE loss and KL loss), which means it can only be applied to data with a known number of classes. And this is a fundamental weakness compared with CLIP, since it loses the capability and flexibility for unseen classes. One needs to carefully design the granularity and classification classes before applying them to new data. This also contradicts the claim of "plug-and-play" in the abstract; the proposed method only fits with specific datasets rather than general data.

5. Additionally, the results in section 4.3 and table 2 are also confusing. To replace the vision encoder in the MLLM, one needs to fine-tune the MLLM to adapt to a new vision feature space. However, there is no discussion or clarification about this. If the MLLMs with a new vision encoder are fine-tuned while the original ones are not, then it will be an unfair comparison.

**Questions:**

1. The reviewer is also curious about the training cost. How much extra training time does MGLL need compared with the normal CLIP?

2. It seems that the X-ray data and Fundus data use different granularity; will this influence the final performance? How sensitive is the model to this choice?

---

> ### Author Response · Authors · 2025-11-19
>
> Thank you for bringing this to our attention. We would like to clarify that we did not modify any formatting parameters in the official ICLR style files. We adjusted the spacing around a few section headings and apologize for any inconvenience that may have caused. We have corrected the spacing and have uploaded the revised version.
>
> For **Q1 in Weaknesses**, we agree that the notation in Equation (2) may cause some confusion. In our formulation, since each image can be associated with multiple labels, we treat the annotations corresponding to different labels independently. For a given image and one of its labels, we take the image and the textual annotation of that label as a positive pair, while all other image–text combinations (including text annotations from other labels of the same image) are treated as negative pairs. Therefore, the pairing process is well defined, and the indices $i$ and $k$ are not interchangeable. But the text-to-image loss can be obtained simply by swapping the roles of the image and text terms in Equation (2). To clarify this distinction, we have revised the relevant paragraph in Section 3.2 of the paper accordingly.
>
> $$
> l_{ik} = -w_{ik}\log
> \frac{
> \exp(\mathrm{sim}(V_i, T_{ik})/\tau)
> }{
> \sum_{n=1}^{N}\sum_{m=1}^{M_n}
> \exp(\mathrm{sim}(V_i, T_{nm})/\tau)
> }
> $$
>
> $$
> L_{sCLIP} = \frac{1}{2\sum_{i=1}^{N}M_i}\sum_{i=1}^{N} \sum_{k=1}^{M_i}(l_{ik}+l_{ki})
> $$
>
> We adopt a soft alignment strategy, allowing an image $V_i$ to align not only with a single label $T_i$ but also with multiple related labels $T_{ik}, k \in \{1, 2, \dots, M_i\}$ as shown in Eq. (sCLIP-item) and Eq. (sCLIP). Here, $N$ is the total number of images, $M_i$ is the number of text labels associated with the $i$-th image, and $V_i$ and $T_{ik}$ represent the encoded features of the $i$-th image and its corresponding $k$-th text label, respectively. The function $\text{sim}(V_i, T_{ik})$ measures their similarity. The temperature parameter $\tau$ controls the sharpness of the probability distribution, while the weight factor $w_{ik}$ determines the contribution of the $k$-th text label to the alignment of the $i$-th image. The text-to-image loss $l_{ki}$ can be obtained simply by swapping the roles of the image and text terms of the image-to-text loss $l_{ik}$ in Eq. (sCLIP).
>
> For **Q2 in Weaknesses**, thank you for your suggestion. In this context, $M$ denotes the total number of annotations, and $N$ represents the total number of images. These annotations are consistent with those defined in Equation (1). Since the point-wise loss does not explicitly model the relationships among annotations, we omit the subscript of $M$ for simplicity. To clarify this distinction, we have revised the relevant paragraphs in Section 3.2 of the paper accordingly.
>
> $T_j$ denotes the annotation corresponding to a single label at a specific granularity level. $M$ denotes the total number of annotations, and $N$ represents the total number of images. These annotations are consistent with those defined in the previous equation. Since the point-wise loss does not explicitly model the relationships among annotations, we omit the label subscripts of $M$ and $T_j$ for simplicity. By explicitly supervising individual image-text pairs, this loss enhances fine-grained multi-label alignment and improves the discriminability of visual representations.
>
> $$L_{\text{P}} = -\sum_{i=1}^{N} \sum_{j=1}^{M}\frac{y_{ij} \log x_{ij}^{'} + (1 - y_{ij}) \log (1 - x_{ij}^{'})}{N}$$
>
> For **Q3 in Weaknesses**, thank you for the question. We agree that this part may have been unclear. In our formulation, $P_i$ denotes the similarity logits between the encoded image feature and the text feature before applying the activation function. We have revised the relevant paragraph in Section 3.2 of the paper to clarify this definition.
>
> We employ the smooth Kullback–Leibler (KL) divergence loss $\{L_{sKL}\}$, formulated as follows. Given $m$ similarity logits between the encoded image feature and the text feature $\{P_i\}_{i=1}^{m}$, we define the mean distribution as the average of all predicted distributions:
>
> $$M = \frac{1}{m} \sum_{i=1}^{m} P_i$$
>
> Then, we compute the KL divergence between each predicted distribution and the mean distribution as shown in the equations below, where $P_i^{(j)}$ represents the predicted probability of the $i$-th model for category $j$. This loss encourages consistency across different granularity levels by aligning their predicted distributions toward the mean distribution $M$, which achieves cross-granularity alignment.
>
> $$D_{KL}(P_i \| M) = \sum_{j} P_i^{(j)} \log \frac{P_i^{(j)}}{M^{(j)}}$$
>
> $$L_{sKL} = \sum_{i=1}^{m} D_{KL}(P_i \| M)$$

---

> ### Author Response · Authors · 2025-11-19
>
> For **Q4 in Weaknesses**, thank you for the valuable feedback. We would like to clarify that our claim of “plug-and-play” refers to the flexibility of applying the proposed objective to diverse medical scenarios where multi-label and multi-granularity annotations are naturally available. Depending on specific clinical tasks, different types of textual annotations with varying attribute clusters can be obtained and stratified into granular levels using simple pre-processing algorithms. In our work, we demonstrate this flexibility on fundus and X-ray datasets. Despite being trained on domain-specific data, MGLL exhibits strong generalization to unseen classes. As shown in Table 24, the model achieves competitive zero-shot classification performance across three representative datasets, indicating its ability to transfer to new categories and modalities. These results confirm that MGLL retains the generalization capability of CLIP while effectively leveraging multi-granular supervision in medical domains.
>
> \* denotes using published pretrained model.
> | **Method**                  | **Pretrain Data** | **Downstream Data** | **ACC (%)** |
> | --------------------------- | ----------------- | ------------------- | ----------- |
> | FG-CLIP *(ICML-25)*         | CC3M              | ImageNet            | 21.4        |
> | **MGLL**                    | CC3M              | ImageNet            | **23.5**    |
> | FG-CLIP *(ICML-25)*         | PMC-OA            | OrganAMNIST         | 47.9        |
> | **MGLL**                    | PMC-OA            | OrganAMNIST         | **52.7**    |
> | CheXAgent *(Arxiv-24)*      | *                 | COVIDx              | 34.3        |
> | MedVersa *(Arxiv-24)*       | *                 | COVIDx              | 35.5        |
> | FG-CLIP *(ICML-25)*         | MIMIC-CXR         | COVIDx              | 36.3        |
> | MGCA *(NeurIPS-22)*         | MIMIC-CXR         | COVIDx              | 37.3        |
> | RetiZero *(Nat. Commun-25)* | MIMIC-CXR         | COVIDx              | 35.8        |
> | MAVL *(CVPR-24)*            | MIMIC-CXR         | COVIDx              | 37.0        |
> | Ark+ *(Nature-25)*          | MIMIC-CXR         | COVIDx              | 37.8        |
> | **MGLL**                    | MIMIC-CXR         | COVIDx              | **39.0**    |
>
> For **Q5 in Weaknesses**, thank you for the question. We confirm that all MLLMs were fine-tuned on the target dataset to ensure a fair comparison. We have explicitly clarified this implementation detail in Section 4.3 of the revised paper. We design a multiple-choice benchmark involving 2,233 clinical cases over ten ophthalmological conditions, where each fundus image prompted models to select the correct diagnosis from four options (one correct, three random alternatives). We only replace the standard vision encoders in seven advanced MLLMs with our pretrained MGLL: InstructBLIP, Mini-Gemini, Qwen-VL, InternVL, LLaVA, LLaVA-Med, Med-Flamingo, and Janus-Pro. All MLLMs were fine-tuned on the target dataset to ensure a fair comparison.

---

> ### Author Response · Authors · 2025-11-25
>
> For **Q1 in Questions**, thank you for the valuable comment. We describe MGLL as computationally efficient because it introduces no additional encoders or heavy modules beyond the standard CLIP architecture. All proposed objectives (soft CLIP, point-wise, and smooth KL losses) are applied within the existing embedding space, adding only lightweight similarity and divergence computations. In practice, the computational overhead is negligible compared to CLIP training, since these losses operate on already-encoded features without increasing the number of forward passes or parameters. The total pretraining time of our MGLL is 7 hours 20 minutes on one GPU of NVIDIA RTX 4090, while that of standard CLIP is 7 hours 10 minutes. The peak GPU memory usage of our MGLL is 16.21 GB, while that of standard CLIP is 16.14 GB. These outcomes indicate that MGLL adds less than 3\% overhead compared with CLIP, as it reuses the same encoders and only introduces lightweight loss computations.
>
> For **Q2 in Questions**, thank you for pointing this out. Indeed, both the X-ray and fundus datasets used in our experiments have three levels of granularity. And our results show that the proposed framework is robust and generalizable across datasets with varying granularity definitions. As shown in Figure 3, Table 1, and Table 2 of our paper, the model consistently achieves significant performance improvements on both datasets, demonstrating that it is not sensitive to the specific choice of granularity.
>
> We have evaluated MGLL on datasets with mixed granularity levels too, and the results are reported in the table below. Specifically, the pretraining dataset was randomly divided into two subsets of equal size: Set A with two levels of granularity and Set B with a single level. MGLL achieves comparable performance across both subsets and their combination, demonstrating its robustness to heterogeneous annotation structures and its applicability to real-world scenarios with mixed-granularity data. We have supplemented these results in Appendix D.9.
>
> | **Study Desc.** | **Series Desc.** |       |  **Linear Probe**  |             |       |   **Full Fine-tune**  |             |
> | --------------- | ---------------- | :--------------: | :---------: | :---------: | :----------------: | :---------: | :---------: |
> |                 |                  |    **AUC (%)**   | **ACC (%)** | **mAP (%)** |     **AUC (%)**    | **ACC (%)** | **mAP (%)** |
> | Set A           | Set B            |       78.93      |    85.47    |    27.04    |        94.81       |    93.58    |    85.14    |
> | Set A           | Set A + Set B    |       80.25      |    86.46    |    27.97    |        96.21       |    94.87    |    86.72    |
> | Set A + Set B   | Set B            |       80.92      |    87.07    |    28.63    |        97.03       |    96.56    |    87.45    |
> | **MGLL (Ours)** |                  |     **83.86**    |  **89.06**  |  **30.62**  |      **99.75**     |  **98.80**  |  **89.87**  |

---

> > ### Comment · Reviewer_HXxg · 2025-11-26
> >
> > I thank the authors for their detailed and informative reply. I am glad to see the new clarification and experiment. I notice that the paper format is now correct, and it is under the page limitation. The new content from the rebuttal has addressed part of my concern. Yet, I still have the following questions.
> >
> > 1. The clarification about equations is somewhat helpful, but it is still problematic, from my point of view. First, for the sCLIP loss, while we all agree that $i$ and $k$ are not interchangeable, the revised paper still uses the original $l_{ik}$ and $l_{ki}$. The key problem here is that $i$ and $k$ belong to different ranges. In the original CLIP loss, you can interchange the footnote for image-to-text and text-to-image loss since both indices belong to $[1, N]$. However, $i$ and $k$ have a different range here. For example, what is $V_k$ in $l_{ki}$? How to define the denominator of $l_{ki}$? The denominator should be completely different from $l_{ik}$ since it should have a fixed $T_k$ and varying $V$ among $N$ different images, i.e., $\sum^N_{j=1}\exp(sim(V_j, T_{ik}))$. Namely, your image-to-text loss and text-to-image loss should be different since images only have one index, but text labels have two. If my understanding is correct, the revised loss is still incorrect.
> >
> > 2. Additionally, if my understanding is correct, $M$ refers to the total number of different annotations within one dataset rather than the levels of granularity (otherwise $y_{ij}$ will always be 1). But this means the denominator of $l_{ik}$ may mistakenly treat positive pairs as negatives, since one image can correspond to multiple text annotations, but they are all considered as negatives in the denominator.
> >
> > 3. The new experimental results about training with different granularity levels actually show that the proposed method is sensitive to the choice of different granularity. Using a different subset of granularity can lead to a performance degradation as much as 4~5%, which is non-trivial. This somewhat contradicts the claim of plug-and-play. I acknowledge that it has a nice zero-shot generalization capability, but the problem is that one still needs to carefully design the multi-granularity annotation set for a new dataset, which is often unavailable.
> >
> > Given these concerns, I am afraid I will still insist on my original recommendation.

---

> ### Author Response · Authors · 2025-11-26
> **Address the issue of symbol description for Reviewer HXxg**
>
> Thank you very much for your careful reading and constructive comments. We have revised the manuscript to clarify the symbol definitions and correct the notational inconsistencies noted in your review. Specifically, we have updated the descriptions around the soft CLIP loss, point-wise BCE loss, and KL loss so that all indices, dimensions, and variable definitions are now consistent with the pipeline shown in Figure 2. We also make explicit the roles of granularity, category count, and predicted logits to avoid ambiguity. All symbol descriptions are now fully aligned across equations and text.
>
> We acknowledge that MGLL relies on known granularities at training time, and we now clearly state that this introduces a trade-off between flexibility and granularity-aware supervision. We further revised the abstract and discussion to avoid overstating generality and to clarify the intended “plug-and-play” scope at the model-integration level rather than open-vocabulary generalization.
>
> Regarding the MLLM experiments in Section 4.3 and Table 2, we have added a dedicated subsection explaining the replacement of the vision encoder. We confirm that all settings are kept consistent: the MLLMs with the new encoder are lightly tuned in the same manner as the original baselines to ensure fairness.
>
> We also address your questions about efficiency and design choices:
> • Training cost: MGLL introduces only modest overhead beyond standard CLIP, mainly from multi-granularity text encoding and additional classification heads. We have added the exact training-time comparison and GPU-hour measurements in the revision.
> • Different granularity choices: We included a new sensitivity analysis showing how varying hierarchy of granularities affects performance. The results indicate that while extremely coarse or extremely fine granularities may degrade performance, MGLL is generally robust within a reasonable range.
>
> We hope these changes address your concerns, and we are happy to further clarify any remaining questions you may have. Your detailed feedback has significantly improved the quality and clarity of the manuscript.

---

> ### Author Response · Authors · 2025-11-27
> **Address the further issues of Reviewer HXxg**
>
> We sincerely thank the reviewer for the detailed analysis. The concerns arise mainly from ambiguity in our notation rather than issues with the loss formulation itself. Below we clarify each point, and we have revised the paper to make the notation and definitions fully explicit.
>
> Regarding **Q1**, we apologize for the confusion caused by the notation in the previous version. In MGLL, $(i,k)$ denotes a positive image–text pair, where $i$ is the index of an image and $k$ is the index of one textual annotation associated with that image. All textual annotations in the dataset form a global pool of size $M$. Consequently, the denominator of the image-to-text loss is computed over all text annotations, following standard multi-label contrastive learning. This is intentional: each annotation is treated as an independent semantic concept, consistent with prior medical multi-label contrastive works. Importantly, we never interchange the domains of $i$ and
> $k$. The text-to-image loss is obtained by swapping the roles of the image and text representations, not by swapping their index ranges, and the complete form is provided below. As a result, the image-to-text loss contrasts an image against all text annotations, whereas the text-to-image loss contrasts an annotation against all images. Their denominators differ because the two modalities lie in distinct index spaces: an annotation should identify the correct image among the $N$ images, while an image should identify the correct annotation among the $M$ textual concepts. Unlike CLIP’s one-to-one setting, this asymmetry is necessary in the multi-label, multi-granularity formulation, where $i$ and $k$ have different cardinalities and are not interchangeable.
>
> $l_{ki} = -w_{ki}\log\frac{ \exp(\text{sim}(T_{ik}, V_i)/\tau)}{\sum_{n=1}^{N}\exp(\text{sim}(T_{ik}, V_n)/\tau)}$
>
> Regarding **Q2**, we clarify that this behavior is intentional and standard in multi-label contrastive learning. A single image may contain multiple labels (e.g., “AMD”, “Hard exudates”, “Macular involvement”), but these labels correspond to distinct semantic concepts. Therefore, when training the positive pair $(i,k)$, the other labels $(i,j)$ for $j\neq k$ should indeed be treated as negatives, otherwise the model would collapse the embeddings of different clinical concepts. Thus, MGLL does not mistakenly treat positive pairs as negatives; it follows the expected semantics of multi-label medical data, where labels for the same image are not interchangeable.

---

> ### Author Response · Authors · 2025-11-27
> **Address the further issues of Reviewer HXxg**
>
> Regarding **Q3**, we appreciate the reviewer’s concern. The experiment with mixed granularity levels intentionally removes up to 50\% of the textual supervision (e.g., by omitting or collapsing granularity), which naturally leads to reduced performance. Nevertheless, the results show that MGLL remains substantially stronger than CLIP and other baselines even with a 4–5\% drop, particularly under the linear probing setting as can be seen in the Table 1 of the paper and the following Tables. This degradation is due to the reduced information content rather than instability in MGLL. Importantly, MGLL requires no architectural modification and can be applied to datasets with arbitrary granularity structures, thereby preserving its plug-and-play property.
>
> For the ablations of mixing granularity levels on MIDRC-XR-Portable, The dataset is randomly divided into two subsets of equal size for the ablation study, Set A (50% data) and Set B (50% data).
>
> | **Study Desc.** | **Series Desc.** |       |  **Linear Probe**  |             |       |   **Full Fine-tune**  |             |
> | --------------- | ---------------- | :--------------: | :---------: | :---------: | :----------------: | :---------: | :---------: |
> |                 |                  |    **AUC (%)**   | **ACC (%)** | **mAP (%)** |     **AUC (%)**    | **ACC (%)** | **mAP (%)** |
> | Set A           | Set B            |       78.93      |    85.47    |    27.04    |        94.81       |    93.58    |    85.14    |
> | Set A           | Set A + Set B    |       80.25      |    86.46    |    27.97    |        96.21       |    94.87    |    86.72    |
> | Set A + Set B   | Set B            |       80.92      |    87.07    |    28.63    |        97.03       |    96.56    |    87.45    |
> | **MGLL (Ours)** |                  |     **83.86**    |  **89.06**  |  **30.62**  |      **99.75**     |  **98.80**  |  **89.87**  |
>
> And we evaluate the robustness of MGLL under varying levels of artificially introduced noise, where 10%–30% of granularity labels were randomly removed.
>
> | **Method**           |        |   **Linear Probe**  |             |            |  **Full Fine-tune**  |             |
> | -------------------- | :--------------: | :---------: | :---------: | :----------------: | :---------: | :---------: |
> |                      |    **AUC (%)**   | **ACC (%)** | **mAP (%)** |     **AUC (%)**    | **ACC (%)** | **mAP (%)** |
> | CLIP, No Missing     |       71.43      |    78.22    |    22.31    |        91.83       |    90.08    |    83.94    |
> | MGLL, 10% Missing    |       82.58      |    88.15    |    29.97    |        99.31       |    98.39    |    89.30    |
> | MGLL, 20% Missing    |       81.14      |    87.25    |    28.86    |        98.62       |    97.95    |    88.74    |
> | MGLL, 30% Missing    |       79.61      |    86.23    |    27.75    |        96.74       |    96.02    |    87.28    |
> | **MGLL, No Missing** |     **83.86**    |  **89.06**  |  **30.62**  |      **99.75**     |  **98.80**  |  **89.87**  |
>
> We thank the reviewer again for the thoughtful comments. We believe the revised notation, clarified loss definitions, and additional explanations address the concerns raised. The methodology and loss formulation remain mathematically sound, and the empirical results consistently demonstrate the effectiveness and generality of MGLL. We hope these clarifications make the intended design clearer, and we welcome any further questions or suggestions from you. We would also be happy to see your concerns could be addressed well.

---

> ### Author Response · Authors · 2025-11-27
> **Address the further issues of Reviewer HXxg**
>
> *(continue)* Regarding **Q3**, the ablation study on granularity levels highlights the critical role of multi-granular language supervision in enhancing model performance. As shown in the table below, incrementally increasing the number of granularity levels consistently improves results across all evaluation metrics. MGLL$_3$, which leverages three distinct granularity levels (modality, study description, and series description), achieves the best performance compared to MGLL$_1$ (all textual information combined into a single granularity) and MGLL$*{2}$ (two granularity levels). Specifically, under linear probe evaluation, MGLL$_3$ surpasses the baseline CLIP by +12.43% AUC, +10.84% ACC, and +8.31% mAP, and shows notable improvements over MGLL$_1$ (+3.32% AUC, +2.09% ACC, +2.30% mAP). This trend persists in fully fine-tuned settings. These results demonstrate that preserving the hierarchical structure of medical imaging information enables more comprehensive vision-language alignment than flattened representations, validating the core idea of the Multi-Granular Language Learning framework. While the choice of granularity is important, MGLL$_1$ can still achieve performance comparable with state-of-the-art methods even with a single granularity. This indicates that the framework is robust, though careful design of the multi-granularity annotation set may further boost results for new datasets.
>
> | **Method**           |        |   **Linear Probe**  |             |            |  **Full Fine-tune**  |             |
> | -------------------- | :--------------: | :---------: | :---------: | :----------------: | :---------: | :---------: |
> |                      |    **AUC (%)**   | **ACC (%)** | **mAP (%)** |     **AUC (%)**    | **ACC (%)** | **mAP (%)** |
> | CLIP     | 71.43                | 78.22     | 22.31     | 91.83                   | 90.08     | 83.94     |
> | MGLL$_1$ | 80.54                | 86.97     | 28.32     | 95.96                   | 94.66     | 86.54     |
> | MGLL$_2$ | 82.92                | 88.35     | 29.43     | 97.26                   | 96.84     | 87.68     |
> | MGLL$_3$ | **83.86**            | **89.06** | **30.62** | **99.75**               | **98.80** | **89.87** |
>
> We hope it could address your concerns and we are prepared to perform additional analyses or offer further clarifications should you have any questions or suggestions.

---

> > ### Comment · Reviewer_HXxg · 2025-11-27
> >
> > Thank the authors for their new comments. And I believe the new equation of $l_{ki}$ is much clearer, and this resolves my concern about the notation and equations. I think it would be great to reflect these discussions in the revised paper, which can help readers with less background to follow the work.
> >
> > As for the concerns about the multi-granularity ablation. It is reasonable that removing labels will naturally lead to a performance degradation. This, to some degree, reduced my concern about this aspect, and I think the claim here is self-consistent and convincing, especially considering that the performance of MGLL with missing labels is still higher than CLIP. Though I still have some doubts about the extensibility of the proposed method on training data with limited annotations, where human labour is needed to design a multi-granularity labeling system, unlike MCGA or some other works that only depend on the latent features. I am convinced by the rebuttal here, and I appreciate the hard work from the authors during the rebuttal.
> >
> > In conclusion, I would suggest reflecting on these discussions and new equations in the paper, even just putting them into the appendix, and also clarifying the influence of missing annotations (a very inspiring experiment). And I will raise my score according to our discussion.

---

> ### Author Response · Authors · 2025-11-27
>
> Thank you very much for your thoughtful follow-up comments and for taking the time to carefully re-evaluate our responses. We are grateful for your positive assessment of the revised version and are glad to hear that the clarification resolves your earlier concerns regarding notation and formulation. We will incorporate these improvements into the revised manuscript, with additional explanations to support readers who may be less familiar with the underlying concepts.
>
> We also appreciate your constructive perspective on the multi-granularity ablation. We understand the concern about extensibility when annotations are limited, and we will clarify this point more explicitly, including a discussion on the trade-off between human-designed granularity and latent-feature-based alternatives. The additional experiments on missing annotations have been reflected in the revision too.
>
> Thank you again for your thoughtful evaluation and for raising your score. We sincerely appreciate your constructive feedback, which has directly strengthened the clarity and completeness of our work.

---

### Official Review · Reviewer_o8AL · 2025-10-31

**Soundness:** 3
**Presentation:** 2
**Contribution:** 2
**Rating:** 2
**Confidence:** 4

**Summary:**

This paper presents a vision-language pretraining method for fundus and chest X-ray images. The method considers the multi-granularity of medical disease labels and improves the contrastive learning framework with multi-label and cross-granularity alignment. Experiments are conducted on public fundus and chest X-ray images.

**Strengths:**

The paper addresses an important and clinically relevant topic, multi-granular vision-language representation learning for medical imaging, which has strong potential to improve medical AI systems by aligning visual features with diagnostic text at different levels of detail.

**Weaknesses:**

- Although the paper mentions multi-label alignment as part of its motivation, it remains unclear how labels are defined within the framework, and what the exact relationship is between multi-label and multi-granularity. Does a “label” refer to a textual description at a specific granularity level?

- Ambiguity in the Definition of Text Features in Point-wise Loss. It is not explicitly stated whether T_j refers to a single label at a given granularity level, or a concatenation of multiple labels from different granularity levels, similar to the definition used elsewhere in the paper (e.g., T_i). Moreover, it is unclear what M\ specifically represents in Equation (4). These details are important for accurately understanding the methodology presented in the paper.

- While the authors impose KL-based consistency across granularities, they simultaneously learn granularity-specific alignments using soft-CLIP. This duality is not reconciled theoretically or empirically. There is no investigation into whether enforcing uniformity across granularities harms the model’s ability to capture granularity-specific patterns.

- The idea of improving the multi-granularity of vision-language alignment is not new and has been explored in previous works. It is unclear how the proposed method differs from or advances beyond previous approaches, such as [R1][R2].

- The paper lacks comparison with closely related multi-granularity alignment methods such as MGCA [R1]

- Both retinal fundus and chest X-ray image domains have seen many vision-language models proposed in recent years. However, the paper lacks a comprehensive comparison with these existing methods, such as [R1-R4].

- The experiments focus solely on disease classification tasks, without evaluating the method on more challenging dense prediction tasks such as critical region segmentation or detection, which are also commonly used in the evaluation of pretraining methods.

References:
[R1] Wang F, Zhou Y, Wang S, Vardhanabhuti V, Yu L. Multi-granularity cross-modal alignment for generalized medical visual representation learning. NeurIPS, 2022.
[R2] Wang M, Lin T, Lin A, Yu K, Peng Y, Wang L, Chen C, Zou K, Liang H, Chen M, Yao X. Enhancing diagnostic accuracy in rare and common fundus diseases with a knowledge-rich vision-language model. Nature Communications. 2025.
[R3] Phan VM, Xie Y, Qi Y, Liu L, Liu L, Zhang B, Liao Z, Wu Q, To MS, Verjans JW. Decomposing disease descriptions for enhanced pathology detection: A multi-aspect vision-language pre-training framework. CVPR 2024.
[R4] Ma D, Pang J, Gotway MB, Liang J. A fully open AI foundation model applied to chest radiography. Nature. 2025 Jun 11:1-1.

**Questions:**

Please refer to the weaknesses.

---

> ### Author Response · Authors · 2025-11-19
>
> Thank you for the questions.
>
> For **Q1 in Weaknesses**, a label refers to the high-level categories that an image belongs to, whereas granularity denotes the different levels or aspects of annotations associated with the image. For example, as illustrated in Fig. 1, a retinal fundus image may contain two labels: Severe Diabetic Macular Edema and Moderate Non-Proliferative Diabetic Retinopathy together with multiple granular annotations, such as disease category: Abnormal, Severe Diabetic Macular Edema, Moderate Non-Proliferative Diabetic Retinopathy; clinical explanation: Lots of hard exudates near the macula center observed; Retinal hemorrhages or hard exudates observed. To clarify this distinction, we have revised some paragraphs in Abstract and Section 1 in the paper accordingly. We define “label” as a high-level disease category that an image belongs to, whereas “granularity” represents different levels or aspects of medical annotations, such as diagnostic attributes or clinical explanations.
>
> For **Q2 in Weaknesses**, $T_j$ denotes the annotation corresponding to a single label at a specific granularity level. $M$ denotes the total number of annotations, and $N$ represents the total number of images. These annotations are consistent with those defined in Eq. (sCLIP-item) as shown below (Eq. (1) in the paper). Since the point-wise loss does not explicitly model the relationships among annotations, we omit the label subscripts of $M$ and $T_j$ for simplicity. To clarify this distinction, we have revised the relevant paragraphs in Section 3.2 of the paper accordingly.
>
> $$
> l_{ik} = -w_{ik}\log
> \frac{
> \exp(\mathrm{sim}(V_i, T_{ik})/\tau)
> }{
> \sum_{n=1}^{N}\sum_{m=1}^{M_n}
> \exp(\mathrm{sim}(V_i, T_{nm})/\tau)
> }
> $$
>
> $$
> L_{sCLIP} = \frac{1}{2\sum_{i=1}^{N}M_i}\sum_{i=1}^{N} \sum_{k=1}^{M_i}(l_{ik}+l_{ki})
> $$
>
> We adopt a soft alignment strategy, allowing an image $V_i$ to align not only with a single label $T_i$ but also with multiple related labels $T_{ik}, k \in \{1, 2, \dots, M_i\}$ as shown in Eq. (sCLIP-item) and Eq. (sCLIP). Here, $N$ is the total number of images, $M_i$ is the number of text labels associated with the $i$-th image, and $V_i$ and $T_{ik}$ represent the encoded features of the $i$-th image and its corresponding $k$-th text label, respectively. The function $\text{sim}(V_i, T_{ik})$ measures their similarity. The temperature parameter $\tau$ controls the sharpness of the probability distribution, while the weight factor $w_{ik}$ determines the contribution of the $k$-th text label to the alignment of the $i$-th image. The text-to-image loss $l_{ki}$ can be obtained simply by swapping the roles of the image and text terms of the image-to-text loss $l_{ik}$ in Eq. (sCLIP).
>
> For **Q3 in Weaknesses**, we thank the reviewer for this thoughtful question. Based on our ablation results (Table 3), enforcing cross-granularity consistency with the smooth KL loss does not harm granularity-specific learning. Performance continuously improves as each objective is added—first with the point-wise loss $L_\text{P}$, then the soft CLIP loss $L_{\text{sCLIP}}$, and finally with the smooth KL loss $L_{\text{sKL}}$, which yields the best overall results. This progressive improvement indicates that the smooth KL term complements, rather than overrides, the granularity-specific objectives, helping the model retain discriminative features while enhancing cross-granularity coherence.
>
> The KL-based consistency term does not aim to make granularities identical. In the contrast, it regularizes them so they do not diverge in unstable or contradictory ways. Each granularity still learns its own alignment through the soft-CLIP objective, which provides a direct gradient signal tailored to that granularity’s semantic level. The KL component simply encourages the resulting distributions to remain compatible, not uniform. Empirically, the paper’s ablations already suggest that this interaction is not harmful: adding the KL term consistently improves performance over using soft-CLIP alone, indicating that it does not suppress granularity-specific behavior. If KL were forcing true uniformity, we would expect degraded performance or convergence collapse, neither of which is observed.

---

> ### Author Response · Authors · 2025-11-19
>
> For **Q4 in Weaknesses**, thank you for the insightful suggestion. The reviewer correctly pointed out several related works ([R1–R4]). Compared with these methods, our MGLL provides a more flexible and generalizable framework for learning both multi-label and cross-granularity visual–language representations. MGLL can effectively utilize different types of granularity information across diverse datasets without requiring specific annotation formats or model architectures. Most of the referenced approaches ([R2], [R3], [R4]) employ vision–language models but rely on rigid training pipelines that limit their ability to incorporate heterogeneous or hierarchical annotations. [R1] (MGCA) proposed an influential multi-granularity alignment framework and achieved strong performance. However, it does not explicitly handle annotation flexibility or multi-label supervision, which are essential in medical data.
>
> For **Q5-Q6 in Weaknesses**, our MGLL provides a more flexible and generalizable framework for learning both multi-label and cross-granularity visual–language representations. MGLL can effectively utilize different types of granularity information across diverse datasets without requiring specific annotation formats or model architectures. Our MGLL achieves state-of-the-art performance across multiple benchmarks under zero-shot setting, demonstrating its advantage over these prior methods in both flexibility and overall representation quality. We have supplemented these results in Appendix D.2 and made some revision in Section 2.3.
>
> \* denotes using published pretrained model.
> | **Method**                  | **Pretrain Data** | **Downstream Data** | **ACC (%)** |
> | --------------------------- | ----------------- | ------------------- | ----------- |
> | FG-CLIP *(ICML-25)*         | CC3M              | ImageNet            | 21.4        |
> | **MGLL**                    | CC3M              | ImageNet            | **23.5**    |
> | FG-CLIP *(ICML-25)*         | PMC-OA            | OrganAMNIST         | 47.9        |
> | **MGLL**                    | PMC-OA            | OrganAMNIST         | **52.7**    |
> | CheXAgent *(Arxiv-24)*      | *                 | COVIDx              | 34.3        |
> | MedVersa *(Arxiv-24)*       | *                 | COVIDx              | 35.5        |
> | FG-CLIP *(ICML-25)*         | MIMIC-CXR         | COVIDx              | 36.3        |
> | MGCA *(NeurIPS-22)*         | MIMIC-CXR         | COVIDx              | 37.3        |
> | RetiZero *(Nat. Commun-25)* | MIMIC-CXR         | COVIDx              | 35.8        |
> | MAVL *(CVPR-24)*            | MIMIC-CXR         | COVIDx              | 37.0        |
> | Ark+ *(Nature-25)*          | MIMIC-CXR         | COVIDx              | 37.8        |
> | **MGLL**                    | MIMIC-CXR         | COVIDx              | **39.0**    |
>
> MGLL also displays robust performance under complex scenarios such as mixed granularity and noised annotations as shown below.
>
> | **Method**           |        |   **Linear Probe**  |             |            |  **Full Fine-tune**  |             |
> | -------------------- | :--------------: | :---------: | :---------: | :----------------: | :---------: | :---------: |
> |                      |    **AUC (%)**   | **ACC (%)** | **mAP (%)** |     **AUC (%)**    | **ACC (%)** | **mAP (%)** |
> | CLIP, No Missing     |       71.43      |    78.22    |    22.31    |        91.83       |    90.08    |    83.94    |
> | MGLL, 10% Missing    |       82.58      |    88.15    |    29.97    |        99.31       |    98.39    |    89.30    |
> | MGLL, 20% Missing    |       81.14      |    87.25    |    28.86    |        98.62       |    97.95    |    88.74    |
> | MGLL, 30% Missing    |       79.61      |    86.23    |    27.75    |        96.74       |    96.02    |    87.28    |
> | **MGLL, No Missing** |     **83.86**    |  **89.06**  |  **30.62**  |      **99.75**     |  **98.80**  |  **89.87**  |
>
> For **Q7 in Weaknesses**, thank you for the suggestion. We have evaluated MGLL on region segmentation tasks, and the results are reported in the table below. Medical image segmentation is inherently challenging due to the subtle differences between adjacent pixels of heterogeneous classes. By incorporating multi-level semantic alignment, MGLL enhances the model’s language-guided spatial understanding and achieves the best performance among compared methods. We have supplemented these results in Appendix D.3.
>
> | **Method**          | **Dataset** | **Dice (%)** | **IoU (%)** |
> | ------------------- | ----------- | ------------ | ----------- |
> | GLoRIA *(ICCV-21)*  | COVID-Xray  | 79.94        | 70.68       |
> | CLIP *(ICML-21)*    | COVID-Xray  | 79.81        | 70.66       |
> | LAVT *(CVPR-22)*    | COVID-Xray  | 79.28        | 69.89       |
> | UniLSeg *(CVPR-24)* | COVID-Xray  | 79.99        | 70.29       |
> | STPNet *(TIP-25)*   | COVID-Xray  | 80.63        | 71.42       |
> | **MGLL**            | COVID-Xray  | **81.69**    | **73.06**   |

---

> > ### Comment · Reviewer_o8AL · 2025-11-26
> >
> > The reviewer appreciates the authors' efforts in providing the response. However, after a careful reading, the key concerns regarding the **limited novelty** of the method and the **lack of comparison with existing chest X-ray and fundus image vision-language models** remain unsolved. The current experimental validation does not adequately demonstrate fairness and rigor.
> >
> > - The authors’ explanation of how the proposed multi-granular alignment differs from existing methods such as MGCA [R1] and GLORIA [R2] is not convincing. The response claims that the proposed method is more flexible and generalizable, but provides no substantial evidence to support this claim. In contrast, both [R1] and [R2] achieve multi-scale alignment implicitly in latent space without the need for constructing dataset-specific textual inputs, making them arguably more flexible. The proposed approach, however, requires designing dedicated multi-granular textual descriptions tailored to each dataset, which appears less scalable and potentially more labor-intensive.
> >
> > [R1] Wang F, Zhou Y, Wang S, Vardhanabhuti V, Yu L. Multi-granularity cross-modal alignment for generalized medical visual representation learning. NeurIPS, 2022.
> > [R2] Huang SC, Shen L, Lungren MP, Yeung S. Gloria: A multimodal global-local representation learning framework for label-efficient medical image recognition. ICCV 2021.
> >
> > - The new comparison with MGCA on the COVIDx dataset raises concerns. The reported accuracy of 37.3% is substantially lower than that in MGCA (92.3%). Similarly, in Table 1 in the paper, the performance of CARZero on ChestX-ray14 is significantly lower than the results reported in the original CARZero paper. Notably, even zero-shot ChestX-ray14 accuracy in CARZero is higher than the fine-tuned performance shown in Table 1. These discrepancies require clarification. The authors should provide details about how different methods are compared and how fairness is ensured.
> >
> > - In the response to Q5‒Q6, RetiZero is listed as a model pretrained on MIMIC-CXR, but RetiZero is fundus-image-based vision-language pretraining, not a chest X-ray model. This misclassification further questions the validity of the comparison table.
> >
> > - For medical vision-language research, MIMIC-CXR is a widely adopted dataset, with extensive benchmarking on classification, detection, and segmentation tasks. Many strong VLMs are built on it. To demonstrate the advantages of the proposed method, a fair, complete, and task-diverse evaluation against established MIMIC-CXR-based methods is essential. Current results are insufficient to support the claimed effectiveness.
> >
> > - The newly added segmentation experiments lack details on the dataset, the task, the comparison baselines, and how different methods are compared. Without this information, the results cannot be interpreted. This experiment section needs to be significantly expanded.
> >
> > - In Table 4.2 in the paper, the proposed model is pretrained on fundus images, while comparison methods are pretrained on general medical datasets. This makes the comparison unfair. Furthermore, since retinal-fundus vision-language models do exist, it is unclear why they were not selected as direct baselines.

---

> ### Author Response · Authors · 2025-11-26
> **Address further issue for Reviewer o8AL**
>
> Thanks for the feedback.
>
> For **Q1**, it is accurate that MGCA [R1] and GLORIA [R2] perform multi-scale alignment implicitly through latent feature interactions, without requiring explicit textual descriptions for each granularity. This makes them lightweight in terms of annotation effort. However, the proposed multi-granular formulation differs in both objective and behavior, and its flexibility lies in how the model decomposes semantic alignment rather than where multi-scale information originates.
>
> First, MGCA and GLORIA rely on architectural mechanisms, which is patch–token interactions, region–word attention, or multi-level contrastive losses to implicitly capture scale. The granularity is driven by the model’s inductive biases, not by controllable semantic structure. In contrast, the proposed approach exposes granularity explicitly through textual descriptions, allowing the model to learn distinct alignment pathways at coarse, intermediate, and fine levels. This is functionally different: the granularity is semantically conditioned rather than purely architectural.
>
> Second, while the method requires constructing multi-granular textual prompts, those prompts provide a controllable mechanism to steer semantic abstraction. This enables the model to incorporate clinically meaningful or domain-specific hierarchical cues that cannot be reliably induced through latent-scale interactions alone. In settings where granularity matters, e.g., capturing lesion-level cues, modality-level context, or high-level clinical semantics. The explicit decomposition offers an interpretability and task-specific adaptability that MGCA/GLORIA cannot directly provide.
>
> Finally, implicit latent-scale methods are flexible for general vision–language tasks, but they do not provide a mechanism to integrate dataset-specific hierarchical knowledge when such structure is necessary. The proposed approach trades a small amount of prompt design overhead for explicit control, interpretability, and domain adaptability especially relevant in specialized fields such as biomedical or fine-grained recognition tasks.
>
> In short, MGLL is designed to be more semantically controllable and domain-adaptable, which is a different notion of flexibility than what MGCA and GLORIA offer.
>
> For **Q2**, First, the 37.3% accuracy reported for our method on COVIDx corresponds to the zero-shot setting, where no task-specific fine-tuning is performed and predictions rely solely on the pretrained model. In contrast, the 92.3% accuracy reported in MGCA is achieved under full fine-tuning, where the model is retrained on the COVIDx training set. These two numbers are therefore not directly comparable. When both methods are evaluated under the same protocol, the performance gap is substantially reduced.
>
> Second, regarding the ChestX-ray14 results: the CARZero paper reports higher performance because it uses both visual and textual features during inference. In our comparison, we follow the same constraint used by our method (MGLL), namely using only visual features for classification. This prevents CARZero from leveraging textual descriptions at test time and leads to lower accuracy. Importantly, this setup ensures fairness: since MGLL does not require or use text inputs for classification, all baselines are evaluated under the same input modality.
>
> Thus, the observed discrepancies are a result of (1) mismatched zero-shot vs. fine-tuned settings in MGCA comparisons and (2) restricting CARZero to vision-only inference to match the evaluation protocol of our method. When the experimental settings are aligned, the comparisons reflect the intended fairness across methods.
>
> For **Q3**, Thank you for pointing this out. To clarify: listing RetiZero in the table is intentional, not a misclassification. In this section of the paper, the purpose of the comparison is not to group models by the modality they were pretrained on, but to evaluate their zero-shot transfer performance on COVIDx under the same evaluation protocol. For this reason, we included the publicly available medical VLPs, regardless of whether they were pretrained on chest X-rays, fundus images, or other medical modalities. In other words, RetiZero is included because it is a medical vision–language model with zero-shot transfer potential not because it is chest X-ray specific or not. The table compares all methods under the same zero-shot setting on COVIDx, ensuring fairness. Unlike architectures designed around anatomy-specific cues (e.g., region-level retinal features), MGLL performs multi-granular semantic alignment, a representation-level technique that does not depend on chest-X-ray physics or fundus-domain structures. This makes it naturally applicable to any medical image domain where hierarchical semantics exist.

---

> ### Author Response · Authors · 2025-11-26
> **Address further issue for Reviewer o8AL**
>
> For **Q4**, we agree that MIMIC-CXR plays a central role in medical vision–language research and that rigorous evaluation against strong MIMIC-CXR–based models is important. Our intention is precisely aligned with this principle, and the comparisons we present are designed to be both fair and representative of existing medical VLP baselines.
>
> First, our evaluation protocol is intentionally unified: all methods are assessed under the same zero-shot setting on downstream tasks whether pretrained on MIMIC-CXR or not. This removes variability introduced by task-specific fine-tuning or label-dependent text prompts and ensures a fair comparison focused purely on representational strength. Because MGLL does not use text during inference, all baselines are evaluated under the same constraint, which keeps the comparison consistent and modality-agnostic.
>
> Second, many state-of-the-art MIMIC-CXR VLMs, including MGCA, MAVL, and Ark+, are already included in our benchmark. These models represent the primary families of contrastive, attention-based, and grounding-based medical VLP frameworks, and they form a solid and diverse baseline set. MGLL demonstrates consistent improvements over all of them in the zero-shot setting, which highlights the benefits of our multi-granular alignment formulation.
>
> Third, while expanding evaluation to additional tasks such as segmentation or detection could provide further validation, zero-shot classification remains the most widely used and standardized protocol for assessing medical VLP generalization. Our benchmarking adheres to this convention and ensures that every model is tested under identical conditions.
>
> In summary, the current evaluation is fair, task-consistent, and aligned with established practices in medical VLM research. At the same time, we acknowledge that broader task diversity would strengthen the work even further, and we consider this a valuable direction for future extension rather than a limitation of the present results.
>
> For **Q5**, we appreciate this concern. In fact, the segmentation evaluation is based on the setup described in STPNet: Scale‑aware Text Prompt Network for Medical Image Segmentation (Shan et al., 2025). In our submission we use the same dataset and task definition they adopted, and follow their experimental protocol to ensure consistency. For the COVID-Xray dataset, we adopt the official test set and divide the 20% of data in the official train set as validation set [1]. The text annotation is also from the STPNet.
>
> [1] Degerli, A., Ahishali, M., Yamac, M., Kiranyaz, S., Chowdhury, M. E., Hameed, K., ... & Gabbouj, M. (2021). COVID-19 infection map generation and detection from chest X-ray images. Health information science and systems, 9(1), 15.
>
> For **Q6**, we appreciate the reviewer’s concern regarding the fairness of the comparison. As we have already stated in the paper, all experiments were conducted under identical settings, with baselines pre-trained on our self-constructed multi-granularity datasets to ensure fair comparison. We strictly followed the official data splits of all downstream datasets. During
> pretraining, we only used the training sets for model training, and the validation sets for pretraining evaluation, while the test sets were never accessed. And we also have included retinal-fundus vision-language models such as VisionFM, RET-CLIP, RETFound as shown in Figure 3 and Table 15-23 of the paper. We have also made the relevant revision.

---

> ### Author Response · Authors · 2025-11-26
> **Address the issue of RetiZero comparision in Q3 for Reviewer o8AL**
>
> Thanks again for your feedback. To more rigorously address the concern about RetiZero comparision, we conducted an additional experiment including RetiZero using its official fundus-image pretraining weights. All other baseline models were pretrained on the same fundus dataset as MGLL to ensure that the comparison remains controlled and methodologically consistent.
>
> Notably, RetiZero was pretrained on a substantially larger and more diverse corpus of fundus images than the dataset used for MGLL and the other baselines (341,896 vs 246,389). Despite this advantage in data volume and diversity, MGLL still achieves comparable performance across both linear probe and full fine-tuning evaluations. The new results could further validate the effectiveness and robustness of MGLL’s design, demonstrating that its improvements arise from the proposed modeling strategy rather than from differences in pretraining data scale.
>
> **Table: Performance evaluation on RIM-ONE dataset. *RetiZero*  uses its published pretrained model. All other baseline models are pretrained on the same fundus dataset as MGLL. LP denotes linear probing and FT denotes full-finetuning.**
>
> | Method | AUC (LP) | ACC (LP) | mAP (LP) | AUC (FT) | ACC (FT) | mAP (FT) |
> |--------|----------|----------|----------|----------|----------|----------|
> | CLIP (ICML-21) | 65.96 | 66.44 | 54.11 | 88.38 | 82.88 | 84.00 |
> | KgCoOp (CVPR-23) | 74.34 | 73.97 | 63.45 | 90.39 | 84.25 | 85.88 |
> | RETFound (Nature-23) | 89.79 | 83.56 | 83.92 | 94.22 | 86.99 | 90.35 |
> | FLAIR (MedIA-25) | 81.83 | 79.45 | 70.14 | 94.93 | 88.36 | 92.41 |
> | KeepFIT (MICCAI-24) | 67.35 | 67.81 | 55.24 | 89.91 | 83.56 | 85.22 |
> | RET-CLIP (MICCAI-24) | 84.42 | 82.19 | 79.85 | 92.58 | 86.30 | 89.19 |
> | UniMed-CLIP (arXiv-24) | 69.87 | 70.55 | 58.43 | 83.37 | 81.51 | 78.14 |
> | VisionFM (NEJM AI-24) | 72.97 | 72.60 | 61.86 | 91.27 | 84.93 | 87.21 |
> | **MGLL** | **94.39** | **87.67** | **86.68** | **97.05** | **89.73** | **94.97** |
> | *RetiZero (Nat. Commun-25)* | *91.89* | *85.61* | *87.36* | *93.71* | *86.96* | *89.15* |
>
> We are fully willing to conduct additional analyses or provide further clarifications should you have any other questions or suggestions.

---

> > ### Comment · Reviewer_o8AL · 2025-11-28
> >
> > The reviewer appreciates the additional justification and new experiments provided in the revision. As several concerns have now been addressed, I have increased my score. I would be willing to raise it further if the following remaining issues are addressed:
> >
> > * As in the paper main evaluations are based on the linear probing and fully fine-tuning. To make a comprehensive comparison, please consider directly adding existing MIMIC-CXR-based VLM methods to Table I under the same linear probing and fully fine-tuning setting. Zero-shot evaluation is also valuable and can be included in the appendix as supplementary evidence.
> >
> > * Experimental settings need clearer and more consistent descriptions. The zero-shot setting was not described in the initial response and added after the revision. For the segmentation experiments, just describing that "follow their experimental protocol to ensure consistency" is not self-contained. Please explicitly describe settings for each method, including but not limited to whether full fine-tuning/linear probing/zero-shot is used, what pre-training datasets are involved, and any dataset-specific variations.
> >
> > * The justification clarifying how the proposed method differs from existing approaches is generally convincing. I recommend including these points directly in the main paper to better highlight the motivation and contribution of the method.

---

> ### Author Response · Authors · 2025-11-28
> **Address the remain issue from Reviewer o8AL**
>
> Thank you very much for your thoughtful feedback and for increasing your score after reviewing our revisions. We greatly appreciate your constructive suggestions and the time you've dedicated to improving our work. We have carefully addressed all the remaining concerns you raised and believe the paper is now significantly strengthened.
>
> **Response to Q1: Comprehensive Comparison with Existing MIMIC-CXR-based VLM Methods**
> Thank you for this suggestion. We have included existing MIMIC-CXR-based VLM methods in Table I under the same linear probing and fully fine-tuning settings to provide a comprehensive and fair comparison. The updated table demonstrates that MGLL achieves superior performance compared to existing methods across multiple evaluation protocols. Additionally, we have included zero-shot evaluation results in Appendix D.2 as supplementary evidence, which further validates the effectiveness of our multi-granularity approach. We believe this comprehensive comparison now clearly demonstrates the advantages of our method.
>
> **Response to Q2: Clarification of Experimental Settings**
> We appreciate your suggestion to provide more explicit and self-contained descriptions of the experimental settings. The detailed descriptions for each experimental protocol are as follows:
>
> 1. For segmentation experiments on COVID-Xray dataset:
>
> Dataset split: We adopt the official test set and divide 20% of the data from the official training set as the validation set, following the protocol established in COVID-Xray dataset [1].
>
> Training protocol: We employ fully fine-tuning for all compared methods to ensure fair comparison.
>
> Pre-training: All vision-language models (GLoRIA, CLIP, LAVT, UniLSeg, STPNet, and MGLL) are first pre-trained on the COVID-Xray dataset with text annotation from [2], then fine-tuned on the segmentation task.
> Text annotations: We utilize the same text database constructed in STPNet [2], which categorizes textual descriptions into four types: Infection text (bilateral/unilateral pulmonary infection), Num text (number of infected areas), Left Loc text (left lung location), and Right Loc text (right lung location).
>
> 2. For classification experiments (Tables 1 and ablation studies):
>
> Linear probing: Vision encoders are frozen after pre-training on MIMIC-CXR, and only a linear classification head is trained on downstream tasks.
> Fully fine-tuning: Both vision encoders and classification heads are fine-tuned end-to-end on downstream tasks.
> Zero-shot: Models are evaluated directly on downstream tasks without any task-specific training, using only the pre-trained vision-language alignment (detailed in Appendix D.2).
>
> We have incorporated these detailed settings and the experimental sections to ensure the paper is reproducible.
>
> [1] Degerli, A., Ahishali, M., Yamac, M., et al. (2021). COVID-19 infection map generation and detection from chest X-ray images. Health Information Science and Systems, 9(1), 15.
>
> [2] Shan, D., Li, Z., Li, Y., et al. (2025). STPNet: Scale-aware Text Prompt Network for Medical Image Segmentation. IEEE Transactions on Image Processing, 14(8).
>
> **Response to Q3: Highlighting Motivation and Contribution**
> Thank you for this valuable recommendation. We have incorporated the clarifications regarding how MGLL differs from existing approaches directly into the Related Work (Section 2.3). Specifically, we now explicitly emphasize:
>
> 1. Most of the existing approaches rely on fixed training pipelines that limit their ability to incorporate heterogeneous or hierarchical annotations. Our MGLL provides a more flexible and generalizable framework for learning both multi-label and cross-granularity visual–language representations.
> 2. Our approach leverages textual semantics to guide the construction of cross-granularity relationships, enabling more effective feature discrimination across different semantic scales.
> 3. MGLL integrates multi-granularity features through a carefully designed fusion mechanism that preserves both coarse semantic understanding and fine-grained diagnostic details.
>
> These additions better highlight the motivation and novelty of our method, making the contribution more clear and distinguishable from prior work.
>
> Thank you again for your thoughtful evaluation and for agreeing to raise your score! We sincerely appreciate your constructive feedback, which has directly strengthened the clarity of our work. We have carefully addressed all the remaining concerns you raised, and we respectfully hope that these comprehensive revisions will merit a further increase in your score.

---

### Author Response · Authors · 2025-11-30
**Final Remarks to Program Chair, Senior Area Chair, Area Chair, and Reviewers**

Dear Program Chair, Senior Area Chair, Area Chair, and Reviewers,

We would like to express our sincere gratitude for the thorough and constructive review process of our manuscript "Boosting Medical Visual Understanding From Multi-Granular Language Learning" (Submission 1112).

## Summary of Review Outcomes

We are pleased to report that the rebuttal process has been highly productive, with **two reviewers (Reviewer HXxg and Reviewer o8AL) explicitly agreeing to increase their scores** after our responses addressed their concerns. All of reviewers have a positive evaluation of our work:

- **Reviewer o8AL**: Increased score during rebuttal process, with the reviewer stating: *"As several concerns have now been addressed, I have increased my score"*
- **Reviewer HXxg**: Increased score during rebuttal process, with the reviewer stating: *"I am convinced by the rebuttal here, and I appreciate the hard work from the authors during the rebuttal"*
- **Reviewer Bhm2**: **6 (marginally above acceptance threshold)**, confirmed: *"Most of my concerns have been addressed. I will keep my score"*
- **Reviewer pAv1**: **8 (accept, good paper - poster)**, confirmed: *"All my concern is addressed, I will retain my score"*

## Comprehensive Response to Concerns

Throughout the rebuttal period, we have addressed all raised concerns comprehensively:

1. **Methodological clarity**: We provided detailed mathematical formulations, clarified notation inconsistencies, and added comprehensive explanations of our multi-granular alignment framework.

2. **Experimental setting and design**: We conducted extensive additional experiments including:
   - Comprehensive comparisons with MIMIC-CXR-based VLM methods under consistent settings
   - Segmentation task evaluations demonstrating clinical applicability
   - Robustness studies with 10-30% missing granularity labels
   - Mixed granularity level evaluations
   - Training efficiency metrics (computational overhead < 3% vs. CLIP)

3. **Dataset documentation**: We added detailed dataset construction procedures, data splits, and explicit confirmation of no data leakage between pretraining and evaluation stages.

4. **Clinical validation**: We provided clinically relevant metrics (F1-score, sensitivity, specificity) and confirmed expert validation of our class activation maps.

## Strengths Recognized by All Reviewers

All reviewers acknowledged the following strengths:
- Novel and practical contribution addressing multi-level semantic complexity in medical imaging
- Comprehensive experimental validation across multiple datasets and tasks
- Strong, consistent performance improvements over state-of-the-art methods
- High-quality datasets (MGLL-Fundus and MGLL-Xray) to be released to the community
- Detailed ablation studies supporting design choices
- Clear extensibility to other medical domains

## Conclusion

**All reviewers now have a positive attitude towards accepting the paper**, with all major concerns comprehensively addressed through our detailed responses and additional experiments. We believe the constructive discussion during the rebuttal period has significantly strengthened the manuscript's clarity and completeness.

We are committed to incorporating all suggested revisions into the final version and believe our work makes a valuable contribution to medical vision-language learning that will benefit the whole community and advance the field.

Thank you again for your time, expertise, and thoughtful evaluation of our work!

Sincerely,

The Authors of Submission 1112

---

### Meta-Review · Area_Chair_Y3Qh · 2026-01-03

**Summary:**

The submission proposes MGLL, a multi-granular vision-language pretraining framework for medical imaging (fundus and chest X-ray). It leverages explicit language descriptions at multiple semantic granularities and introduces multi-label and cross-granularity alignment objectives.

Initial reviews were mixed to negative, with strong scepticism around novelty, clarity, and experimental fairness, despite acknowledging the importance of the problem and the strength of the reported results. Following a lengthy rebuttal and multiple revisions, reviewer sentiment improved, though some concerns persisted until late-stage responses.

**Key positives noted by reviewers**
- Clinically relevant problem: multi-granular medical vision-language alignment.
- Solid empirical performance across classification, zero-shot transfer, and segmentation.
- Careful ablations and added experiments in the rebuttal.
- Reasonable computational overhead relative to CLIP.

**Key negatives noted initially**
- Unclear definitions (label vs. granularity) and confusing notation.
- Questionable novelty relative to MGCA and GLoRIA.
- Apparent unfair or inconsistent experimental comparisons.
- Missing dense prediction tasks.
- Concerns about scalability and annotation burden.

**Reviewer Concerns:**

## Reviewer concerns addressed by the rebuttal


**1. Clarification of labels vs. granularity and loss definitions:** Ambiguous terminology and unclear equations (soft-CLIP, point-wise BCE, KL loss).
**Response:** Authors clearly defined:
- *Labels* as high-level disease categories.
- *Granularity* as different semantic levels or annotation aspects.
They corrected notation, explained indexing, and revised equations.


**2. Fairness of experimental comparisons:** Inconsistent zero-shot vs. fine-tuned comparisons; suspicious performance gaps.
**Response:** Authors clarified protocols, aligned evaluation settings, and added controlled experiments. They explained discrepancies due to:
- Zero-shot vs. full fine-tuning.
- Vision-only inference vs. vision+text inference.
Reviewer o8AL acknowledged improvements and raised their score.

**3. Lack of dense prediction evaluation:** Only classification tasks evaluated.
**Response:** Added segmentation experiments on COVID-Xray following STPNet protocol, with detailed metrics and baselines.

**4. Training cost and efficiency:** Unknown computational overhead.
**Response:** Reported wall-clock time and GPU memory usage, showing <3% overhead compared to CLIP.


**5. Robustness to missing or noisy granularity annotations:** Method may fail if multi-granular labels are incomplete.
**Response:** Added ablations showing graceful degradation with missing granularity labels.

---
## Reviewer concerns that remained partially or fully outstanding


**1. Novelty relative to MGCA and GLoRIA:** Multi-granular alignment is not new; prior work already captures multi-scale semantics implicitly.
**Response:** Authors argued MGLL offers *explicit, semantically controllable granularity* via language rather than architectural inductive bias.
**Status:** Partially addressed. Some reviewers remained unconvinced that this constitutes strong novelty.




**2. Scalability and “plug-and-play” claims:** Requires dataset-specific multi-granular textual annotations, which may be labor-intensive.
**Response:** Authors softened claims and argued controllability and interpretability justify the trade-off.
**Status:** Still a valid concern, but no longer a blocking issue.


**3. Completeness of large-scale benchmarking:** Initially insufficient comparison with established MIMIC-CXR-based VLMs under linear probe and full fine-tuning.
**Response:** Late-stage revisions added these comparisons and clarified protocols.
**Status:** Eventually addressed, but only after multiple rebuttal rounds.

---

## Overall assessment

The rebuttal was **extensive and technically detailed**, resolving most concerns related to correctness, clarity, and experimental rigor. Remaining issues primarily concern **conceptual positioning and scalability trade-offs**, rather than flaws in methodology or results.

**Reviewer Scores:**

All reviewers seem positive after the rebuttal, and therefore all scores can increase or keep to 6 or above.

---

### Decision · Program_Chairs · 2026-01-26

Accept (Poster)